# A physical mechanism of TANGO1-mediated bulky cargo export

Ishier Raote[1†]*, Morgan Chabanon[2,3†], Nikhil Walani[3], Marino Arroyo[3,4,5], Maria F Garcia-Parajo[2,6], Vivek Malhotra[1,6,7]*, Felix Campelo[2]*

[1]Centre for Genomic Regulation (CRG), The Barcelona Institute of Science and Technology, Barcelona, Spain; [2]ICFO-Institut de Ciencies Fotoniques, The Barcelona Institute of Science and Technology, Barcelona, Spain; [3]Universitat Politècnica de Catalunya-BarcelonaTech, Barcelona, Spain; [4]Institute for Bioengineering of Catalonia, The Barcelona Institute of Science and Technology, Barcelona, Spain; [5]Centre Internacional de Mètodes Numèrics en Enginyeria (CIMNE), Barcelona, Spain; [6]ICREA, Barcelona, Spain; [7]Universitat Pompeu Fabra (UPF), Barcelona, Spain

**Abstract** The endoplasmic reticulum (ER)-resident protein TANGO1 assembles into a ring around ER exit sites (ERES), and links procollagens in the ER lumen to COPII machinery, tethers, and ER-Golgi intermediate compartment (ERGIC) in the cytoplasm (Raote et al., 2018). Here, we present a theoretical approach to investigate the physical mechanisms of TANGO1 ring assembly and how COPII polymerization, membrane tension, and force facilitate the formation of a transport intermediate for procollagen export. Our results indicate that a TANGO1 ring, by acting as a linactant, stabilizes the open neck of a nascent COPII bud. Elongation of such a bud into a transport intermediate commensurate with bulky procollagens is then facilitated by two complementary mechanisms: (i) by relieving membrane tension, possibly by TANGO1-mediated fusion of retrograde ERGIC membranes and (ii) by force application. Altogether, our theoretical approach identifies key biophysical events in TANGO1-driven procollagen export.

**\*For correspondence:**
ishier.raote@crg.eu (IR);
vivek.malhotra@crg.eu (VM);
felix.campelo@icfo.eu (FC)

[†]These authors contributed equally to this work

## Introduction

Multicellularity requires not only the secretion of signaling proteins –such as neurotransmitters, cytokines, and hormones– to regulate cell-to-cell communication, but also of biomechanical matrices composed primarily of proteins such as collagens, which form the extracellular matrix (ECM) (*Kadler et al., 2007*; *Mouw et al., 2014*). These extracellular assemblies of collagens are necessary for tissue biogenesis and maintenance. Collagens, like all conventionally secreted proteins, contain a signal sequence that targets their entry into the endoplasmic reticulum (ER). After their glycosylation, procollagens fold and trimerize into a characteristic triple-helical, rigid structure, which, depending on the isoform, may extend several hundred nm in length (*McCaughey and Stephens, 2019*). These bulky procollagens are then exported from the ER at specialized export domains, termed ER exit sites (ERES), in a COPII-dependent manner. ERES are a fascinating subdomain of the ER, but a basic understanding of how these domains are created and segregated from the rest of the ER for the purpose of cargo export still remains a challenge. At the ERES, the formation of canonical COPII-coated carriers relies on the polymerization on the membrane surface of a large-scale protein structure: the protein coat. Polymerized coats, such as COPI, COPII, and clathrin coats, usually adopt spherical shapes –which bend the membrane underneath to form small vesicles of ~60–90 nm in diameter (*Faini et al., 2013*), although helical arrangements of COPII coats have been observed (*Zanetti et al., 2013*; *Ma and Goldberg, 2016*). Membrane bending by coats is promoted when the energy gain of coat polymerization is larger than the elastic energy of membrane deformation (*Derganc et al., 2013*; *Scheve et al., 2013*; *Kozlov et al., 2014*; *Saleem et al., 2015*;

*Day and Stachowiak, 2020*). However, how COPII components function specifically to export procollagen remains unclear. The discovery of TANGO1 as a key ERES-resident player has made the processes of procollagen export and the organization of ERES amenable to molecular analysis (*Bard et al., 2006*; *Saito et al., 2009*; *Wilson et al., 2011*).

In the lumen of the ER, the SH3-like domain of TANGO1 binds procollagen via HSP47 (*Saito et al., 2009*; *Ishikawa et al., 2016*; *Figure 1A*). On the cytoplasmic side, TANGO1 has a proline-rich domain (PRD) and two coiled-coil domains (CC1 and CC2) (*Figure 1A*). The PRD of TANGO1 interacts with the COPII components Sec23A and Sec16 (*Saito et al., 2009*; *Ma and Goldberg, 2016*; *Maeda et al., 2017*); the CC1 domain binds the NBAS/RINT1/ZW10 (NRZ) tethering complex to recruit ER-Golgi intermediate compartment (ERGIC) membranes and also drives self-association amongst TANGO1 proteins (*Santos et al., 2015*; *Raote et al., 2018*); and the CC2 domain oligomerizes with proteins of the TANGO1 family (such as cTAGE5) (*Saito et al., 2011*; *Maeda et al., 2016*). Both cytosolic and lumenal activities of TANGO1 are critical for its function. For instance, a recent report identified a disease-causing mutation in TANGO1 in a human family, which results in a substantial fraction of TANGO1 protein being truncated and lacking its cytosolic functions, leading to collagen export defects (*Lekszas et al., 2020*). Recently, we visualized procollagen export domains with high lateral spatial resolution using stimulated emission depletion (STED)

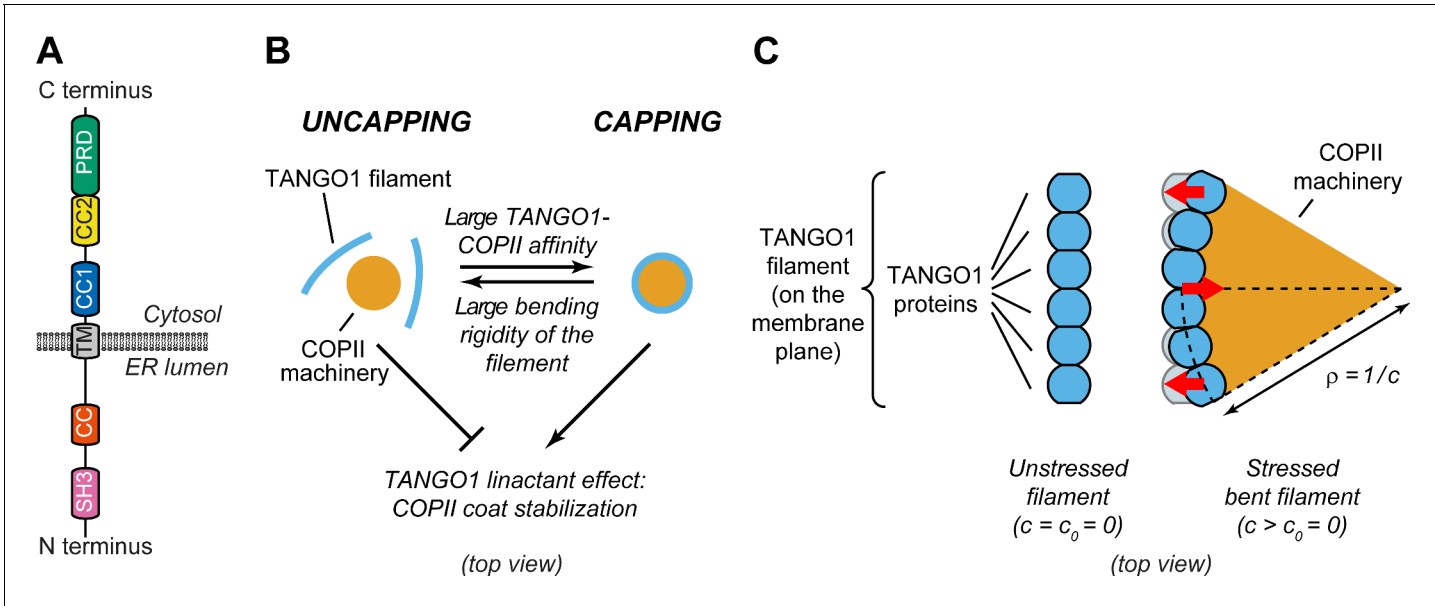

**Figure 1.** Physical model of TANGO1 ring formation. (**A**) Schematic representation of the domain structure and topology of TANGO1, indicating the SH3 domain, a lumenal coiled-coiled domain (CC), the one and a half transmembrane region (TM), the coiled-coiled 1 (CC1) and 2 (CC2) domains, and the PRD. (**B**) Schematic description of the TANGO1 ring formation model (top view). ERES consisting of COPII subunits assemble into in-plane circular lattices (orange), whereas proteins of the TANGO1 family assemble into filaments by lateral protein-protein interactions (light blue). The competition between the affinity of the TANGO1 filament to bind COPII subunits (promoting capping of peripheral COPII subunits) and the resistance of the filament to be bent (promoting uncapping) controls the capping-uncapping transition. Only when TANGO1 caps the COPII lattice, it acts as a linactant by stabilizing the peripheral COPII subunits. (**C**) Schematic representation of individual proteins that constitute a TANGO1 filament and of how filament bending is associated with elastic stress generation. Individual TANGO1 family proteins (blue shapes) bind each other in a way that is controlled by the structure of protein-protein binding interfaces, leading to formation of an unstressed filament of a certain preferred curvature, $c_0$ (left cartoon, showing the case where $c_0 = 0$). Filament bending can be caused by the capping of TANGO1 filament to peripheral COPII machinery (orange area), which generates a stressed bent filament of a certain radius of curvature, $\rho = 1/c$ (right cartoon). Such deviations from the preferred shape (shown in light blue) are associated with elastic stress generation (red arrows point to the direction of the generated bending torque, which correspond to the direction of recovery of the filament preferred shape).

The online version of this article includes the following figure supplement(s) for figure 1:

**Figure supplement 1.** Different contributions to the free energy profile of a transport intermediate as a function of its shape and TANGO1 capping.

nanoscopy in mammalian tissue cultured cells (*Raote et al., 2017*; *Raote et al., 2018*). These studies revealed that TANGO1 organizes at the ERES into ring-like structures, of ~200 nm in lumenal diameter, that corral COPII components. Moreover, two independent studies showed that TANGO1 rings are also present in *Drosophila melanogaster* (*Liu et al., 2017*; *Reynolds et al., 2019*).

To further extend these findings, we combined STED nanoscopy with genetic manipulations and established that TANGO1 rings are organized by (i) lateral self-interactions amongst TANGO1-like proteins, (ii) radial interactions with COPII subunits, and (iii) tethering of small ER-Golgi intermediate compartment (ERGIC) vesicles to assist in the formation of a procollagen-containing export intermediate (*Raote et al., 2018*). Overall, the data suggest a mechanism whereby TANGO1 assembles into a ring, which selectively gathers and organizes procollagen, remodels COPII budding machinery, and recruits ERGIC membranes for the formation of a procollagen-containing transport intermediate. However, the biophysical mechanisms governing these events and how they are regulated by TANGO1 remain unknown.

Here, we put forward a biophysical framework to investigate how TANGO1 rings assemble around polymerizing COPII-coated structures for procollagen export. This general theoretical approach allows us to address: (i) the physical mechanisms by which TANGO1 and its interactors can assemble into functional rings at ERES, forming a fence around COPII coat components and (ii) whether and how a TANGO1 fence can couple COPII polymerization, membrane tension, and force to create an export route for procollagens at the ERES. In particular, we implement this theory in the form of two complementary models: a computational dynamic model that provides us with a general picture of the process of TANGO1-assisted transport intermediate formation; and a simplified but more tractable analytical equilibrium model that allows us to get clearer physical insights.

Overall, our results support a model where TANGO1 temporally and spatially orchestrates the export of procollagen through biophysical interactions with procollagen, COPII components, ERGIC, and the ER membrane.

## Model development

In this section, we present the central hypotheses underlying our general modeling approach. These hypotheses are common to the two models that we subsequently describe: a dynamic computational model (Model A), and an equilibrium analytical model (Model B). We present here a qualitative overview of these models, highlighting their specificities, limitations and strengths. The detailed description of each model is provided in Appendix 1 and Appendix 2, respectively.

### Fundamental hypotheses

To assess and rationalize mechanisms of TANGO1 assembly into rings at ERES and their contribution to procollagen export, we formulate general hypotheses built on accumulated experimental data that will serve as the foundation for our two modeling approaches.

*Hypothesis 1: TANGO1 family of proteins and COPII coat proteins are described as two distinct membrane-bound species.* Given the complexity of the system and the lack of quantitative biochemical data regarding the interactions amongst the different TANGO1 and COPII components, we assume for our modeling purposes that the TANGO1 family of proteins (TANGO1, cTAGE5 and TANGO1-Short) can be effectively described as a single species, to which we will simply refer as TANGO1. Similarly, for simplicity, we describe all the subcomponents of the COPII inner and outer coats as a single species. For our purpose, a more detailed description would only add complexity and free parameters without providing relevant biophysical insights.

*Hypothesis 2: COPII subunits polymerize on the ER membrane forming a coat of specific curvature.* COPII coat assembly at the ERES is relatively well characterized (*Peotter et al., 2019*). Polymerization of COPII subunits into spherical lattices induces curvature on the underlying membrane, typically resulting in spherical carriers of 60–90 nm (*Faini et al., 2013*; *Aridor, 2018*; *Peotter et al., 2019*). However, for bulky cargoes such as procollagens, complementary mechanisms are required to prevent the premature completion of the small carriers and to enable the packaging of large molecules.

*Hypothesis 3: TANGO1 molecules have a propensity to assemble into rings as a result of protein-protein interactions between TANGO1-family proteins.* This hypothesis is based on the observations that: (i) TANGO1 is seen in ring-like filamentous assemblies of specific size by STED nanoscopy

(*Raote et al., 2017*; *Roux, 2018*); (ii) there is a direct 1:1 binding between TANGO1 and cTAGE5 CC2 domains (*Saito et al., 2011*); (iii) TANGO1-Short and cTAGE5 can form oligomers and oligomeric complexes together with Sec12 and TANGO1 (*Maeda et al., 2016*); (iv) TANGO1 and TANGO1-Short can directly homo-dimerize by their CC1 domains (*Raote et al., 2018*); and (v) super-resolution live lattice SIM imaging of TANGO1 in *D. melanogaster* larval salivary gland shows filament growth in ring formation (*Reynolds et al., 2019*). A physical consequence of this hypothesis is that deviations from the preferred TANGO1 ring curvature are associated to an elastic energetic cost: the ring is subject to internal strains and stresses and therefore resists bending away from its preferred curvature (*Figure 1B,C*).

*Hypothesis 4: TANGO1 has a binding affinity to the peripheral COPII coat subunits, helping stabilize the COPII domain boundary by effectively reducing its line energy.* This hypothesis is supported by the observations that proteins of the TANGO1 family bind to the COPII components Sec23, Sec16, and Sec12 (*Saito et al., 2009*; *Ma and Goldberg, 2016*; *Hutchings et al., 2018*; *Raote et al., 2018*). In physical terms, this hypothesis is equivalent to state that TANGO1 acts as a *linactant* by filling unsatisfied binding sites of the COPII coat edges and therefore effectively lowering its line tension (*Figure 1B*; *Trabelsi et al., 2008*, *Saleem et al., 2015*; *Glick, 2017*). Combined with *hypothesis 1*, another consequence of the TANGO1-COPII affinity is the mechanical coupling between the COPII coat edge and TANGO1.

Qualitatively, our generic model for TANGO1 ring assembly can be described as a competition between two different driving forces: the resistance to bending of TANGO1 filamentous assemblies (*hypothesis 3*) and the binding affinity of TANGO1 proteins for peripheral COPII subunits (*hypothesis 4*). In the case of low TANGO1 bending resistance and/or high TANGO1-COPII binding affinity, TANGO1 will easily assemble around COPII patches, forming a ring. In the following, we refer to this process as *capping* of the COPII patch by TANGO1, the analogue to 'wetting' in soft-matter physics (*Figure 1B,C*). As a result of capping, the linactant effect of TANGO1 on COPII-coated ERES reduces the line energy, thus limiting further growth of the COPII lattice and the size of the TANGO1 rings (*Figure 1B*). In contrast, in the case of high TANGO1 rigidity and/or low TANGO1-COPII affinity (for instance, in cells expressing mutants of TANGO1 with reduced or abrogated interaction to COPII proteins), capping of COPII coats by TANGO1 rings will be energetically unfavorable, preventing any linactant effect (*Figure 1B*).

## Model A: dynamic computational model

Here, we summarize our dynamic model of transport intermediate formation at ERES. The detailed, mathematical description of the model is presented in Appendix 1. The model extends the work of *Tozzi et al., 2019* to include two membrane-bound species representing TANGO1 and COPII, respectively.

The ER membrane is represented as a continuous elastic surface on which both protein species can diffuse, interact, and induce membrane curvature. To account for these coupled chemical-mechanical processes, we follow Onsager's variational formalism of dissipative dynamics (*Arroyo and Desimone, 2009*; *Arroyo et al., 2018*). The first step is to define the free energy of the system, which in our case is the sum of the following mechanical and chemical-related energetic contributions:

a. Membrane bending energy: the energetic cost to bend the membrane away from a preferred spontaneous curvature, known as Helfrich energy (*Helfrich, 1973*). Here, we assume that the spontaneous curvature is proportional to the COPII surface density, so that when the surface is saturated with COPII species (fully polymerized lattice), the membrane tends to adopt a given total curvature of $2/R_c$ (where $R_c$ is the radius of curvature of the polymerized COPII coat).

b. TANGO1 ring elastic energy: the energetic cost to bend a TANGO1 filament away from a preferred ring curvature.

c. Entropic free energy: the energetic cost for COPII and TANGO1 to mix or phase separate, here represented by a Flory-Huggins type of mixing free energy.

d. Species self-interaction energy: the energetic gain for COPII and TANGO1 to self-recruit or self-repel.

e.  Interfacial energy: the energetic cost for COPII and TANGO1 domains to have an interface, physically similar to a line energy.

f.  TANGO1-COPII affinity energy.

g.  TANGO1 affinity for COPII domain interfaces (linactant effect), effectively reducing COPII line energy.

Next, we identify the dissipative mechanisms such as those associated with protein diffusion on the membrane surface, and with membrane in-plane shear stress. In addition, the system exchanges energy in the form of external supply of species (fixed chemical potential at the boundaries) and mechanical power. Finally, membrane inextensibility is ensured with a Lagrange multiplier field that has the physical interpretation of the membrane tension.

With these definitions (see Appendix 1), we obtain the rate of change of the system at each time point by numerically minimizing the rate of free energy release, energy dissipation, and energy exchange by the system. To facilitate the computational scheme, we formulate the problem in axisymmetric conditions. Note that this is not a limitation of the model's physics itself but only of its numerical implementation.

The generality of Model A allows us to explore in a thermodynamically consistent manner a large range of possible dynamics of the system, including protein phase-separation, self-organization, spontaneous bud growth, and accessible stable states. However, the cost of this generality is a relative complexity that makes the physical understanding of the system more challenging. For this reason, in the next section, we describe a simplified but analytically tractable model.

## Model B: equilibrium analytical model

To develop a theoretical model of transport intermediate formation at an ERES, we extend and adapt the approach from *Saleem et al., 2015* on clathrin-coated vesicle formation to include the contributions of TANGO1 proteins in modulating COPII-dependent carrier formation. The detailed mathematical description is given in Appendix 2.

The main simplifying assumptions of this model are that (i) the system is at mechanical and thermodynamic equilibrium; (ii) the COPII coat is assumed to be much more rigid that the lipid bilayer, and therefore imposes a *fixed radius of curvature* leading to formation of spherical membranes of fixed radius $R_c$; and (iii) the buds are supposed axisymmetric. The second assumption allows us to limit the family of possible shapes to partial spheres connected to a flat surface, and greatly simplifies the theoretical treatment of the system. As shown below, this assumption is supported by the type of geometries obtained from Model A, which does not assume any a priori geometry.

We consider a continuous lipid membrane with a bending rigidity ($\kappa_b$), under a certain lateral membrane tension ($\sigma$). COPII assembly into a coat is driven by a binding chemical potential ($\mu_c^0$) and is opposed by the membrane bending energy. We consider that the major contribution to the binding chemical potential comes from coat-coat polymerization and not from coat-membrane binding (*Saleem et al., 2015*), so the binding energy $\mu_c^0$ represents the polymerization energy due to COPII-COPII binding, and will be referred to as the COPII polymerization energy in what follows. COPII units can therefore freely exchange with the rest of the membrane, which acts as the reservoir of COPII units at a fixed chemical potential. The edge of the COPII lattice has a line tension ($\lambda_0$) due to free COPII polymerization sites. TANGO1 protein interactions are assumed to favor the formation of a linear filament of preferred curvature ($c_0$) and bending rigidity ($\kappa_T$). Due to its affinity to COPII coat peripheral subunits, TANGO1 filaments can satisfy a fraction ($\omega$) of the edge length, effectively reducing the COPII coat line tension by $\Delta\lambda$. Finally, we also account for the mechanical work of an outward-directed force ($N$) which favors transport intermediate elongation.

The main advantage of this model is that the assumption on the carrier geometry allows us to analytically express the dependency of given quantities of interest on the model parameters, facilitating the physical interpretation of the system. For instance, writing the above mechanisms in terms of the free energy of the system, and taking a bare flat membrane as the reference state, the total free energy change of the system accounting for TANGO1 and COPII can be written as (see Appendix 2):

$$\Delta f_c = \sigma\left(\frac{A_m}{A_p} - 1\right) - \left(\mu_c^0 - 2\frac{\kappa_b}{R_c^2}\right)\frac{A_c}{A_p} + \left[\lambda_0 - \omega\Delta\lambda + \omega\frac{\kappa_T}{2}\left(\frac{1}{\rho} - c_0\right)^2\right]\frac{2\pi\rho}{A_p} - \frac{N\,z_{max}}{A_p}, \tag{1}$$

where $A_m$ is the membrane surface area, $A_c$ is the membrane surface area coated by COPII, $A_p$ is the surface area of the carrier projection onto the flat membrane, $\rho$ is the radius of the neck of the COPII coat, and $z_{max}$ is the height of the carrier (see *Appendix 2—figure 1*). For practical reasons, we define the dimensionless shape parameter $\eta = z_{max}/(2\,R_c)$, so that $\eta = 1$ for a single sphere of COPII spontaneous curvature. *Equation 1* is plotted in *Figure 1—figure supplement 1* for capped ($\omega = 1$) and uncapped ($\omega = 0$) carriers, highlighting the influence of COPII binding chemical potential on the energetically accessible shapes of the system.

## Results

### TANGO1 rings self-assemble to stabilize COPII shallow buds

We first make use of the generality of the computational dynamic model (Model A) to study the formation of TANGO1 assemblies at ERES. We start by investigating the ability of COPII complexes alone to generate spherical carriers. This corresponds to the control case where TANGO1 is treated as an inert species that only contributes to the entropic energy (no chemical nor mechanical interaction with COPII, the membrane, nor with itself). As shown in *Figure 2A*, after an initial nucleation stage (see Appendix 1), the COPII coat grows by recruiting more COPII species as it bends the membrane into a bud (*Figure 2A(i–iii)*, *Figure 2—video 1*). As a metric to follow the bud growth

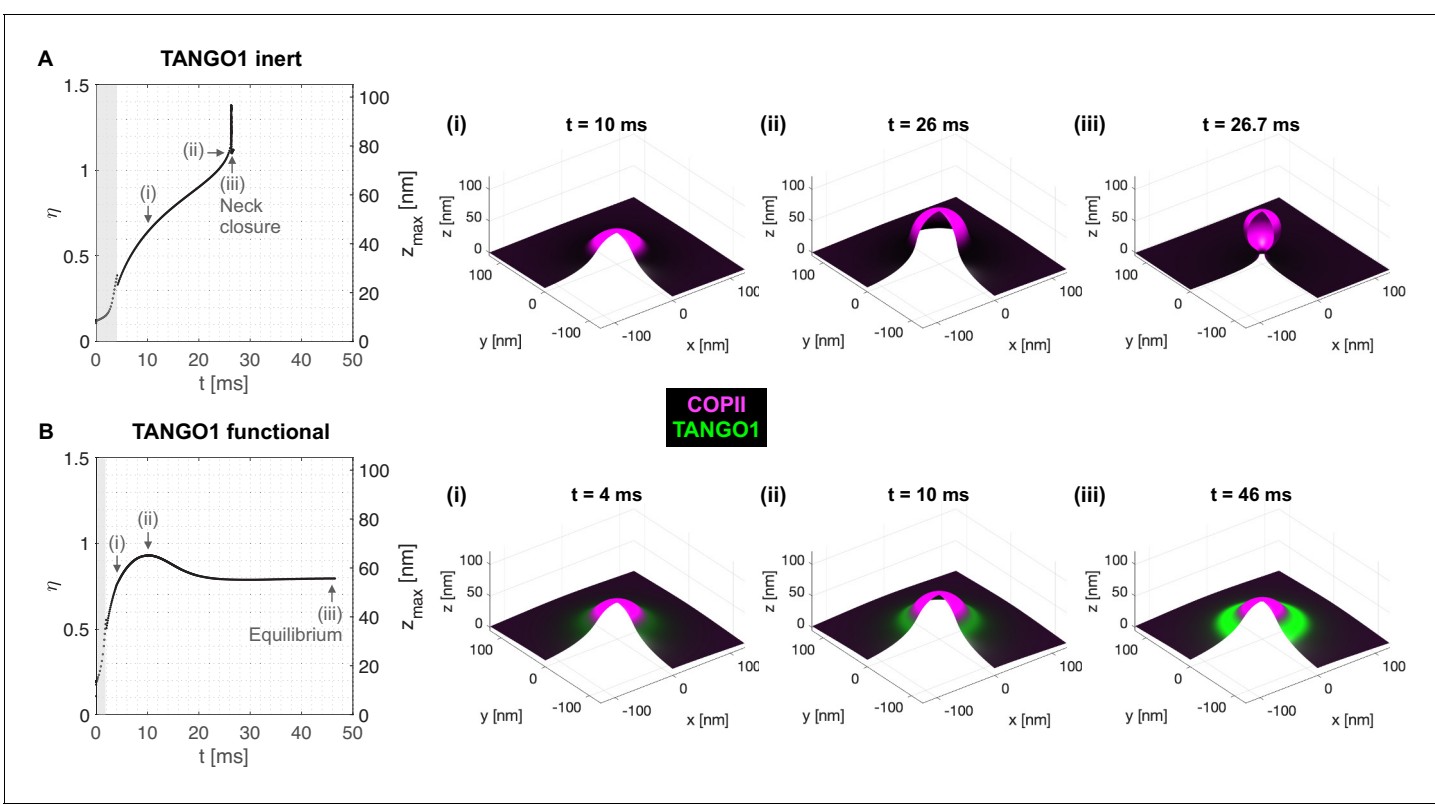

**Figure 2.** Computational dynamic model (Model A) recapitulates TANGO1 ring self-organization around COPII coats and highlights its role in stabilizing incomplete transporters. Shape factor ($\eta$) and height ($z_{max}$) of the transport intermediate as a function of time for (**A**) inert TANGO1 and (**B**) functional TANGO1. (i-iii) Snapshots of the buds at the times indicated in the left graphs. All results obtained for parameters as given in *Appendix 1—table 1*, with $\chi_c = -2k_BT$ and $\sigma = 0.006\ k_BT/nm^2$. Gray areas correspond to the coat nucleation stages (see Appendix 1 for details). See corresponding *Figure 2—video 1* and *Figure 2—video 2* for a dynamic representation of the simulations shown in (**A**) and (**B**), respectively.

The online version of this article includes the following video(s) for figure 2:

**Figure 2—video 1.** Video corresponding to *Figure 2A*.

https://elifesciences.org/articles/59426#fig2video1

**Figure 2—video 2.** Dynamic of COPII bud formation with functional TANGO1 leads to a stable shallow bud.

https://elifesciences.org/articles/59426#fig2video2

with time, we track the height of the bud ($z_{max}$) as a function of time. As seen in *Figure 2A*, for inert TANGO1, the shape parameter – which we have earlier defined as $\eta = z_{max}/(2\,R_c)$ – increases continuously until $\eta \simeq 1$ (*Figure 2(ii)*), before an abrupt event where $\eta$ jumps to 1.4 and then drops back to 1. This event corresponds to the neck closure (Appendix 1, *Figure 2A(iii)*), characterized by a snap-through instability (*Hassinger et al., 2017*).

In contrast, functional TANGO1 drastically modifies the dynamics and outcome of bud formation (see *Figure 2B* and *Figure 2—video 2*). First, the nucleation stage is shortened, suggesting that TANGO1 can facilitate the nucleation of the COPII coat in the presence of a small force perturbation at the membrane. Second, instead of growing to the point where the neck closes, we find that the shape parameter reaches a maximum around $\eta \simeq 0.9$, before slightly decreasing to a stable height around $\eta \simeq 0.8$. As seen from the corresponding snapshots (*Figure 2B(i–iii)*), this dynamic evolution is accompanied by the self-assembly of a high-density TANGO1 ring-like structure around the COPII coat that stabilizes the bud into a shallow, incomplete transport intermediate. These simulation results are in agreement with experimental observations of TANGO1 rings encircling COPII components in procollagen-rich membrane patches (*Raote et al., 2017*; *Raote et al., 2018*).

## Transient reduction of membrane tension in TANGO1-stabilized buds facilitates the formation of large transport intermediates

To test if the formation of a stable incomplete transporter is compatible with the hypothesis that TANGO1 facilitates the biogenesis of large transport intermediate for procollagen export, we applied transient membrane tension reductions to shallow buds stabilized by TANGO1 rings. Such scenario is motivated by our previous experimental data showing that (i) ERGIC53-containing vesicles are recruited to the ERES by TANGO1 via tethering complexes (*Nogueira et al., 2014*; *Santos et al., 2015*; *Raote et al., 2018*), presumably enabling their fusion to the ER membrane and (ii) cells depleted of the tethering factors recruited by TANGO1 to ERES showed a reduction of ~70–80% in collagen secretion (*Raote et al., 2018*).

To mimic the presence of procollagen molecules inside the growing carrier preventing the total closure of the neck, we arbitrarily set a minimum neck radius threshold of 7.5 nm (see Appendix 1 for details). Starting from a stable shallow bud of shape parameter, $\eta \simeq 0.8$, at a large membrane tension of $\sigma = 0.006\ k_BT/nm^2$ (*Figure 3D(i)*), we apply an ad hoc transient tension reduction as follows. First, the tension is reduced to $\sigma = 0.003\ k_BT/nm^2$, leading to bud re-growth. Second, when the bud height reaches $\eta = 1.5$ (just after the beginning of neck closure), the membrane tension is gradually set back to its original value of $\sigma = 0.006\ k_BT/nm^2$ over a 10 ms time ramp (*Figure 3A–C*). We find that the carrier reaches a new equilibrium state at $\eta \simeq 2.8$, corresponding to a 'key-hole' shape of about 195 nm height (*Figure 3D(ii)*). Next, continuing from this new equilibrium state, we repeat the transient tension reduction protocol, this time initiating the ramp to recover high tension for $\eta = 3.5$, just after a new neck is formed. This time, the bud continues growing in a pearled shape despite the recovery of high membrane tension (*Figure 3D(iii,iv)*, and *Figure 3—video 1*). Interestingly, the '2-pearl' shape corresponds to a height of about 300 nm, a typical length of trimerized procollagen I molecules (*McCaughey and Stephens, 2019*).

Altogether, our computational results support the scenario by which TANGO1 self-organizes around COPII coats and stabilizes shallow buds at physiological membrane tensions. This mechanism is enabled by TANGO1 affinity to COPII, TANGO1 filament bending rigidity, and the modulation of COPII coat line energy. The stability of shallow buds might facilitate procollagen recruitment and packaging by TANGO1 as well as the recruitment of ERGIC membranes to the ERES. We suggest that procollagen located at the neck of the growing carrier acts as a means to sterically prevent carrier scission. Transient membrane tension reduction, possibly mediated by fusing ERGIC membranes to the ER, allows the buds to grow from shallow to elongated pearled transport intermediates of sizes compatible with the encapsulation of procollagen molecules.

## Analytical model explains transition from incomplete buds to large transport intermediates as a ratchet-like mechanism

Although the generality of Model A allows us to explore ranges of possible dynamic behaviors of the TANGO1-COPII system, its computational nature and inherent complexity makes its complete and rigorous analysis prohibitively challenging. To overcome this difficulty and gain a deeper

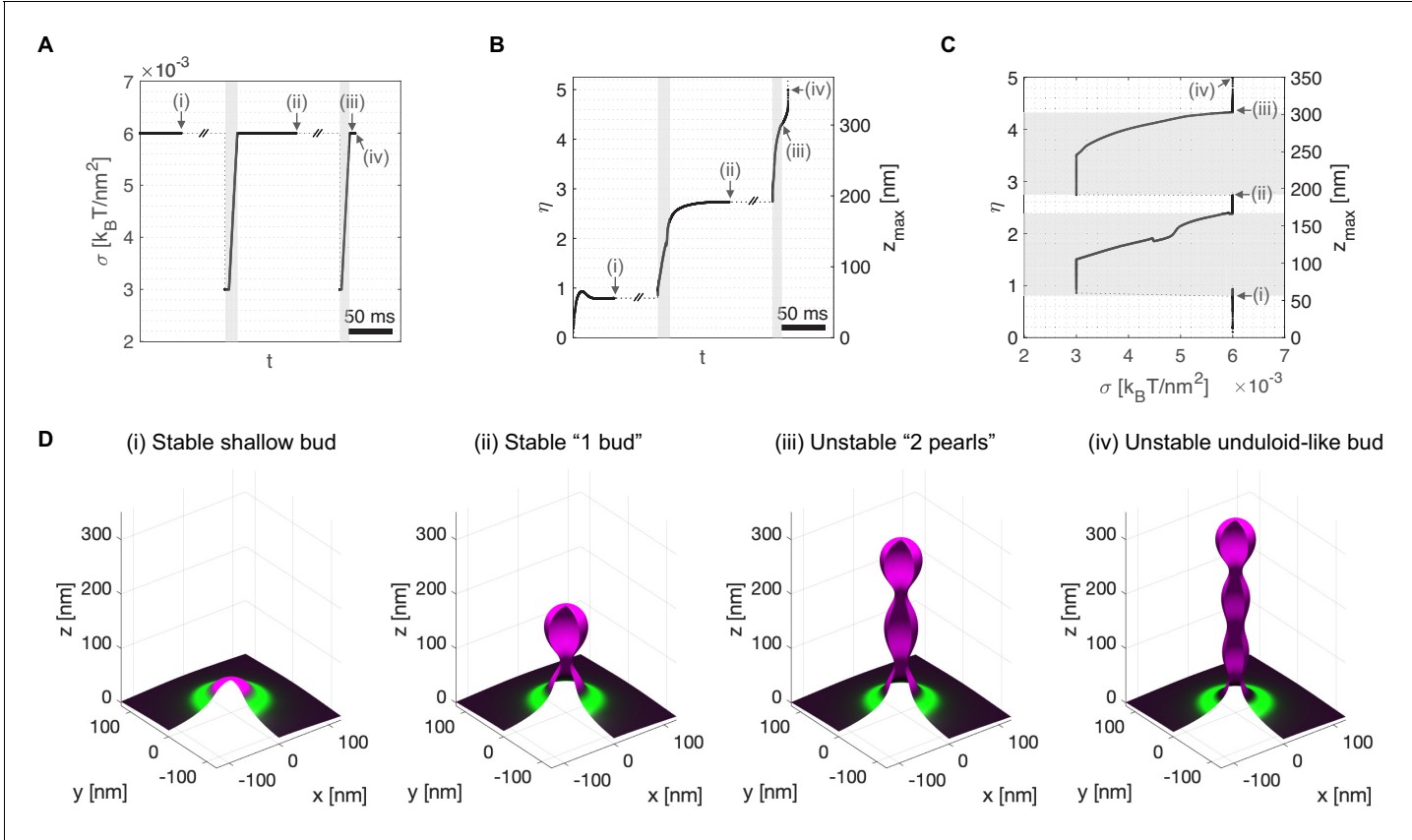

**Figure 3.** Transient membrane tension reduction in TANGO1-stabilized buds enables the formation of large procollagen-containing transport intermediates. A transient decrease of membrane tension mimicking ERGIC membrane recruitment by TANGO1 enables the sequential growth of COPII-TANGO1 transport intermediates. (**A**) Computational protocol for the applied membrane tension as a function of time. When an equilibrium state is reached, $\sigma$ is transiently reduced from $0.006 \ k_B T/nm^2$ to $0.003 \ k_B T/nm^2$ until the transport intermediate grows to either $\eta = 1.5$ or $\eta = 3.5$ Then $\sigma$ is progressively restored to $0.006 \ k_B T/nm^2$ following a linear ramp of ~10 ms. (**B**) Resulting shape parameter, $\eta$, as a function of time. Double-barred lines indicate that the system can remain at equilibrium for an arbitrarily long time. (**C**) Shape parameter as a function of membrane tension, $\sigma$. Gray regions in (**A–C**) correspond to low-tension regimes. (**D**) Three-dimensional rendering of the shape of the transport intermediate and protein surface distribution at the respective points indicated in panels (**A–C**). Parameters as in *Appendix 1—table 1*, with $\chi_c = -2k_B T$ (see Appendix 1 for details, and *Figure 3—video 1* for a dynamic representation of the simulation).

The online version of this article includes the following video for figure 3:

**Figure 3—video 1.** Transient membrane tension reduction in TANGO1 stabilized carriers enables the formation of large procollagen-containing transport intermediates.
https://elifesciences.org/articles/59426#fig3video1

physical insight, we developed an analytically tractable model that relies on two key simplifying assumptions: (i) given the shapes obtained with Model A (*Figures 2* and *3*), we constrain our analysis to carrier geometries that can be approximated by a stack of spheres connected to a flat membrane patch; and (ii) we focus on the local equilibrium states of the system. The details of this equilibrium analytic model (Model B) are summarized in the Model Development section and detailed in Appendix 2.

We start by examining the effect of membrane tension on the accessible equilibrium configurations of the system as a function of the two degrees of freedom of Model B: the transport intermediate shape characterized by the shape parameter ($\eta$), and the capping fraction ($\omega$). This geometric simplification allows to write the free energy change per unit area (*Equation 1*) as (see Appendix 2):

$$\Delta f_c = \begin{cases} \dfrac{\sigma\eta - \tilde{\mu}}{1-\eta} + \dfrac{\tilde{\lambda}(\omega)}{\sqrt{\eta(1-\eta)}} + \dfrac{\omega\tilde{\kappa}_T}{\eta(1-\eta)}\left(\dfrac{1}{\sqrt{\eta(1-\eta)}} - 4c_0 R_c\right), & \eta \leq 1/2 \\[4ex] 4\sigma\left[n + (\eta-n)^2\right] - 4\tilde{\mu}\eta + 4\tilde{\lambda}(\omega)\sqrt{(\eta-n)(1-\eta+n)} \\[1ex] \quad + 4\omega\tilde{\kappa}_T\left(\dfrac{1}{\sqrt{(\eta-n)(1-\eta+n)}} - 4c_0 R_c\right), & \eta \geq 1/2 \end{cases} \qquad (2)$$

where $\tilde{\mu} = \mu_c^0 - 2\kappa_b/R_c^2 + N/(2\pi R_c)$ is the effective chemical potential, which depends on the polymerization energy of the coat on the membrane, $\mu_c^0$, the bending rigidity of the membrane, $\kappa_b$, and the applied pulling/pushing force, $N$; $\tilde{\lambda}(\omega) = (\lambda_0 - \omega\Delta\lambda)/R_c + 4\omega\tilde{\kappa}_T(c_0 R_c)^2$, is the effective line tension of the coat; $\tilde{\kappa}_T = \kappa_T/8R_c^3$ is the renormalized bending rigidity of the TANGO1 filament; and $n$ is the number of fully-formed buds (see Appendix 2).

We plot in *Figure 4A* the free energy per unit area, $\Delta f_c$ (*Equation 2*), as a function of the shape parameter $\eta$, at two membrane tensions corresponding to experimentally measured value of the ER membrane, $\sigma_{ER} = 0.003 k_B T/nm^2$, and of Golgi membranes, $\sigma_{GC} = 0.0012 k_B T/nm^2$, respectively (*Upadhyaya and Sheetz, 2004*). In each case, the free energy per unit area for complete capping (TANGO1 rings forming around COPII patches, $\omega = 1$) and no capping (no TANGO1 rings around COPII patches, $\omega = 0$) are shown. The free energy per unit area of the transport intermediate, $\Delta f_c$, presents multiple local minima separated by energy barriers, indicating that global and locally stable shapes can coexist for a given set of parameters. At 'high' (ER) membrane tension, the *global minimum* of the free energy corresponds to a capped shallow bud ($\eta \simeq 0.35$), while *locally stable* shapes are also found for complete spherical buds connected to a capped shallow bud ($\eta \simeq 1.5$). In contrast, at 'low' (Golgi) membrane tension, the *global* free energy minimum is shifted to the capped complete spherical bud shape (pearled tube), while the incomplete bud state is now a *local* minimum. It should be noted that due to the geometric restriction of Model B, the energy barrier separating locally stable shapes tends to infinity for fully capped coats. However, our results indicate that the uncapping of the coat reduces the energy barrier, suggesting that a capping-uncapping transition might be necessary for a bud to grow. Alternatively, growth of a pearled structure from a shallow bud could also occur through unduloid-like shapes with a fully capped neck, which are not considered by model B, but are indeed observed with model A (see e.g. *Figure 3D*). The parameters controlling the capping-uncapping transition are studied in a later section.

To better visualize the effect of membrane tension on the shift in globally and locally stable shapes, we plot the optimal shape parameters, $\eta^*$, (*Figure 4B*) and energy barriers between stable incomplete buds ($\eta^* < 1$) and large transport intermediates ($\eta^* > 1$) (*Figure 4C*) as a function the membrane tension. The effect of a transient decrease in membrane tension from $\sigma_{ER}$ to $\sigma_{GC}$ on a capped stable shallow bud is represented by the green arrows in *Figure 4B*. Upon tension reduction, the initially globally-stable shallow bud becomes a local but not global minimum (*Figure 4A,B*). Concomitantly, the free energy barrier to transition from an incomplete bud to a large transport intermediate, $f_{1,2}$, is reduced, and the reciprocal free energy barrier to transition back from a large to shallow bud, $f_{2,1}$, is increased (*Figure 4C*). At low membrane tension, the growth of the metastable bud can be triggered by a small perturbation – such as thermal fluctuations or a mechanical perturbation in the system – and might be facilitated by transient uncapping (*Figure 4A*). By assuming Arrhenius kinetics, we can estimate an average transition time as $\tau_{1,2} = t_0 e^{F_{1,2}/k_B T}$, where $t_0 \sim 1\ ms$ is a characteristic time scale of membrane shape dynamics (*Campelo et al., 2017*); and $F_{1,2} \approx f_{1,2}\ \pi R_c^2$ is the overall free energy barrier of the shape transition. For the conditions detailed in *Figure 4C*, we obtain $F_{1,2}(\sigma_{ER}) \approx 33\ k_B T$ and $F_{1,2}(\sigma_{GC}) \approx 8\ k_B T$, and hence the transition time is decreases drastically upon tension reduction, from $\tau_{1,2}(\sigma_{ER}) \sim 10^9\ min$ to $\tau_{1,2}(\sigma_{GC}) \sim 0.1\ min$. Note that our equilibrium model can only predict the free energy barriers but not the actual dynamics of the transitions between metastable and stable states. Then, as the membrane tension recovers the 'high' ER value, the free energy barrier to transition back to a shallow bud remains relatively large ($f_{2,1}(\sigma_{ER}) > f_{1,2}(\sigma_{GC})$), preventing the carrier to recover its shallow, globally stable shape and hence possibly being kinetically trapped in a metastable configuration. Similar estimations as above, give that $F_{2,1}(\sigma_{ER}) \approx 25\ k_B T$, and $\tau_{2,1}(\sigma_{ER}) \sim 10^6\ min$. By increasing these energy barriers, TANGO1 capping

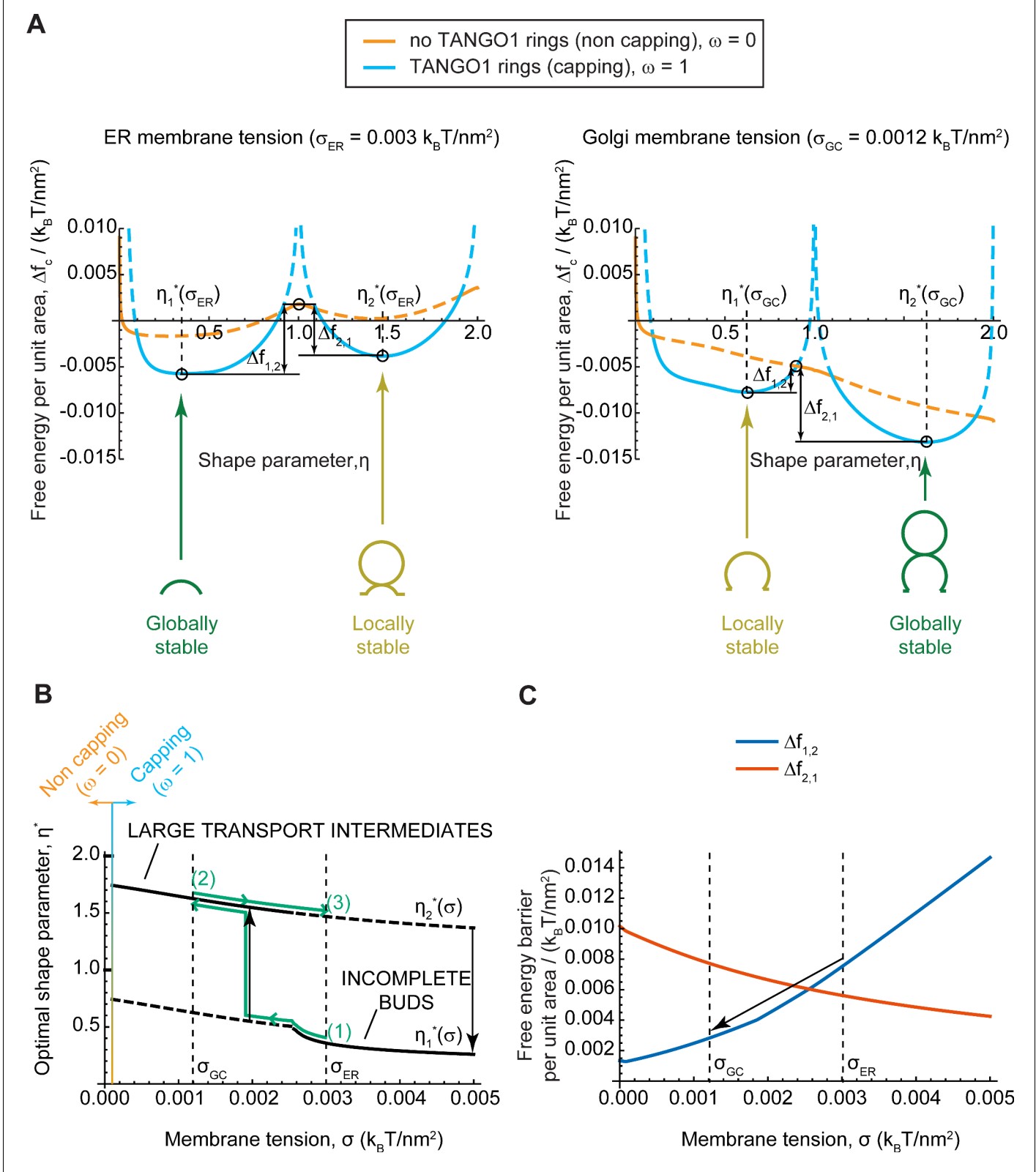

**Figure 4.** Equilibrium analytic model explains tension-mediated carrier growth in terms of energetically accessible configurations. (**A**) Free energy per unit area of the transport intermediate-TANGO1 system, $\Delta f_c$, as a function of the shape parameter, $\eta$, at typical ER membrane tension (left), and Golgi membrane tension (right), for fully capped ($\omega = 1$, sly blue curves), and non-capped ($\omega = 0$, orange curves) cases. Solid lines indicate the locally stable states (lower free energy). Schematics of the transport intermediate shapes for the optimal shape parameters, $\eta_{1,2}^*$, are represented at the global and

*Figure 4 continued on next page*

*Figure 4 continued*

local equilibrium states. (B) Optimal shape parameters, $\eta_{1,2}^*$, as a function of the membrane tension, $\sigma$. Globally stable shapes are indicated with solid lines, while locally stables shapes are shown by dashed lines. A transient reduction in the membrane tension (green arrows from point (1) to point (2)) can lead to the growth of the transport intermediate (black vertical arrow), whereas recovery of the tension to the initial value (green arrows from point (2) to point (3)) can keep the system in a kinetically arrested metastable configuration. The capping-uncapping transition is depicted by the orange-to-sky blue gradient line. (C) The free energy barriers separating the incomplete bud from the large intermediate morphologies, $\Delta f_{1,2}$ (blue line), and $\Delta f_{2,1}$ (vermillion line) as defined in (A), plotted as a function of the membrane tension, $\sigma$. The arrow illustrates how a decrease in membrane tension reduces the energy barrier for growth of the transport intermediate. All parameter values are reported in *Appendix 2—table 1*, with no applied force, $N = 0$. The online version of this article includes the following figure supplement(s) for figure 4:

**Figure supplement 1.** Equilibrium analytic model in no capping conditions.

can contribute to preventing large transport intermediates from shrinking back to shallow shapes during subsequent cycles of membrane tension reduction. Overall, the tension-mediated growth of large transport intermediates is effectively a ratchet-like mechanism, where the difference between the energy barriers guarantees an almost unidirectional evolution of the system leading to the transport intermediate growth.

## TANGO1 expands the parameter space of accessible stable incomplete carriers

The previous results were computed for a typical set of parameters of the TANGO1-COPII system (see *Appendix 2—table 1*). Next, we studied how robust was the stabilization of an incomplete carrier by a TANGO1 ring with respect to the COPII polymerization energy. To do so, we computed the local minima of the overall energy per unit area (*Equation 2*) for a range of values of the COPII polymerization energy, $\mu_c^0$, and the membrane tension, $\sigma$. The results are shown in *Figure 5A* in the form of a shape diagram for the optimal shape parameter, $\eta^*$. We find that at low COPII polymerization energy, the coat fails to bend the membrane into a bud, resulting in a flat membrane ($\eta^* = 0$). At high COPII polymerization energy, however, the system adopts a fully budded structure. Interestingly, we identify a range of intermediate COPII binding energies, ($0.0285\ k_BT/nm^2 < \mu_c^0 < 0.0315\ k_BT/nm^2$) at which a stable shallow bud ($\eta^* < 1$) can be brought to a large pearled shape ($\eta^* > 1$) by membrane tension reduction. Although this range of parameters might seem relatively narrow, similar narrow ranges have been shown to regulate clathrin-coated vesicle formation (*Saleem et al., 2015*; *Hassinger et al., 2017*).

As a comparison, we made use of the dynamic computational model (Model A) to compute the final carrier states and shape parameters for various values of the membrane tension, $\sigma$, and the COPII self-interaction coefficient, $\chi_c$. The resulting shape diagrams are shown in *Figure 5B* for both inert and functional TANGO1. Note that the COPII self-interaction parameter $\chi_c$ in Model A is conceptually equivalent to $\mu_c^0$ in the equilibrium Model B. In agreement with the results obtained with Model B (*Figure 5A*), we find that at low COPII self-interaction, the COPII coat either does not nucleate during the initial perturbation stage, or does nucleate but cannot provide enough bending energy to stabilize the membrane curvature against the membrane tension, thus resulting in a flat membrane ($\eta^* = 0$). At high COPII self-interaction, the nucleated coat successfully works against membrane tension to generate membrane curvature, generating a closed spherical bud. At intermediate COPII self-interaction, we find a transition region where the deformed membrane stabilizes as an open, shallow bud. Interestingly, the presence of functional TANGO1 dramatically widens the parameter space at which stable shallow buds are obtained. Importantly, the phase boundary at the transition from shallow to closed bud now spans a larger range of membrane tension and COPII self-interaction values (*Figure 5B,C*).

Overall, despite the different assumptions underlying the computational dynamic and equilibrium models, their qualitative agreement highlights the relevance of their common underlying physical mechanisms to TANGO1-mediated assembly of procollagen-containing transport intermediates.

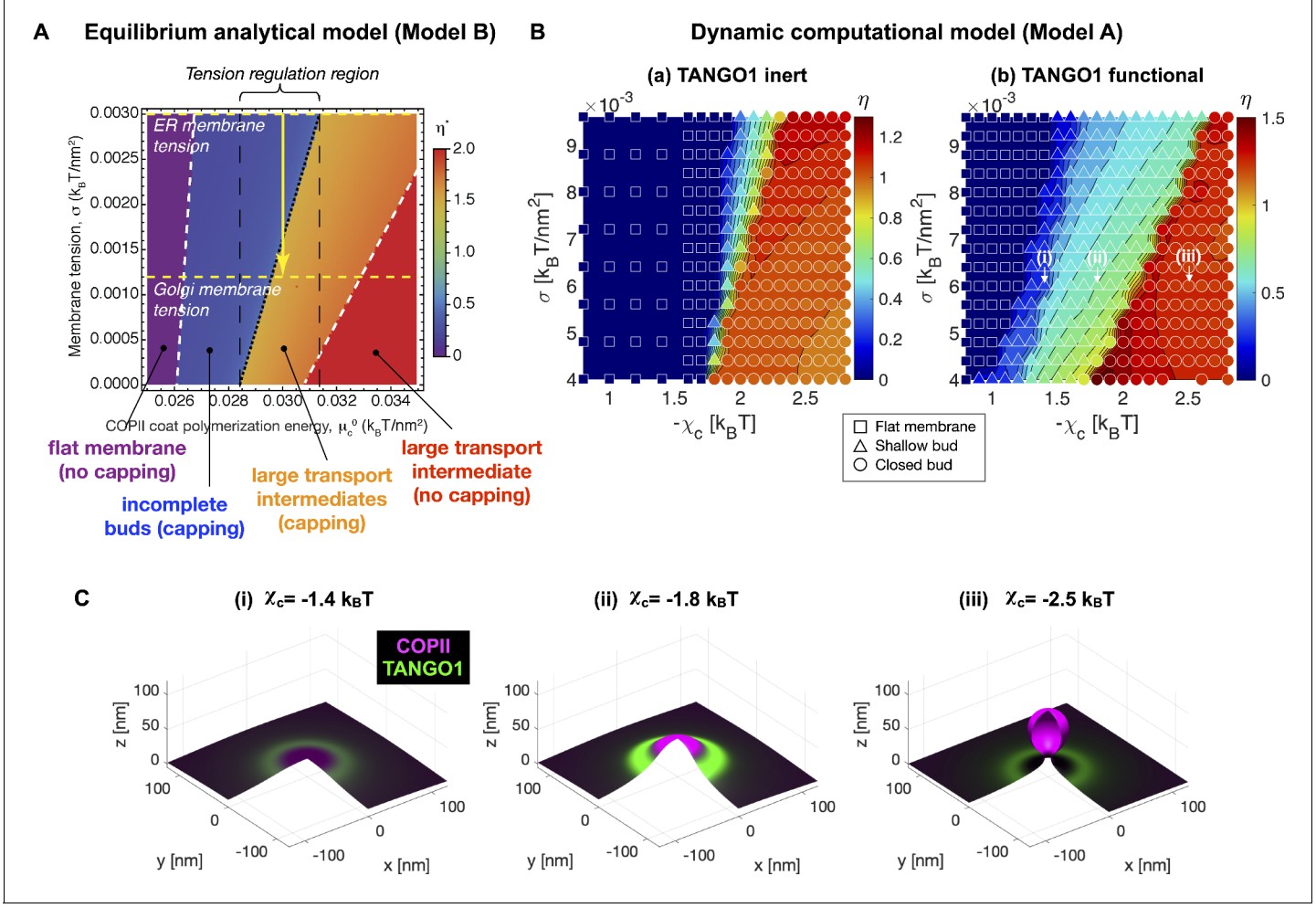

**Figure 5.** TANGO1 widens the parameter space at which membrane tension regulation can trigger the growth of a stable incomplete carrier. (**A**) Phase diagram obtained with Model B displaying the optimal shape parameter as a function of the COPII coat binding energy, $\mu_c^0$, and of the membrane tension, $\sigma$. Different shapes can be identified, from flat uncoated membranes, $\eta^* = 0$, to incomplete coated buds, $\eta^* < 1/2$, and to large transport intermediates, $\eta^* > 1$. Unless otherwise specified, parameters used for all the calculations are listed in *Appendix 2—table 1*. ER and Golgi membrane tensions are marked by yellow dashed lines. (**B**) Shape factor (color code) and final state (symbol) obtained with Model A as a function of membrane tension ($\sigma$) and COPII self-interaction ($\chi_c$). (i) Inert TANGO1 but functional COPII. (ii) Functional TANGO1 and COPII. Each symbol represents the final state of a dynamic simulation, while iso-contours and colors are interpolated from the values at each symbol. (**C**) Snapshots of final states obtained for functional TANGO1 at $\sigma = 0.006 \ k_B T/nm^2$ as indicated in panel (**B**). Note that (i) and (ii) reached local equilibrium states.

## TANGO1 properties control accessible membrane shapes through capping transition

As seen in *Figures 4* and *5*, the capping-uncapping transition of COPII coat by TANGO1 plays an important role in determining the transition between equilibrium-budded states. Therefore, we set out to examine the influence of TANGO1 model parameters on its capping ability.

We define $\eta^{tr}$ as the shape parameters at which the different capping-uncapping transitions occur, which corresponds to the solutions to the equation $\Delta f_c(\eta^{tr}, \omega = 0) = \Delta f_c(\eta^{tr}, \omega = 1)$. Similarly, it is also informative to define the neck opening radius at which the capping-uncapping transition occurs, $\rho^{tr}$, which can be analogously found as the stationary points of the free energy (*Equation 1*) with respect to the capping fraction ($\partial \Delta f_c / \partial \omega = 0$). From these equivalent definitions we obtain

$$\eta^{tr} = n + \frac{1}{2}\left(1 \pm \sqrt{1 - \frac{\xi^2}{R_c^2}(1 + c_0\xi)^2}\right),\tag{3}$$

$$\rho^{tr} = \frac{\xi}{1 + c_0\,\xi},\qquad(4)$$

for $c_0 > -1/\xi$. Here $\xi = \sqrt{\kappa_T/(2\Delta\lambda)} \approx 22nm$ is a TANGO1-related length-scale (see parameter values in *Appendix 2—table 1*). These analytical results highlight how capping of COPII components by TANGO1 is promoted by large values of the TANGO1-COPII interaction, $\Delta\lambda$, and prevented by large TANGO1 filament bending rigidities, $\kappa_T$ (*Figure 6—figure supplement 1A*). Additionally, if the bud opening radius $\rho$ is equal to the radius of curvature imposed by the spherical polymerization of COPII, $R_c$, we can define a critical value of the TANGO1 linactant strength below which there is no capping $\Delta\lambda^{tr} = \frac{\kappa_T}{2}\left(\frac{1}{R_c} - c_0\right)^2$. This is illustrated in *Figure 6A* where the value of the critical COPII boundary radius $\rho^{tr}$ (*Equation 4*) is plotted as a function of TANGO1 filament bending rigidity ($\kappa_T$) and linactant strength ($\Delta\lambda$). For $\rho > \rho^{tr}$, TANGO1 caps the COPII domains, whereas for $\rho < \rho^{tr}$, no capping nor ring formation occurs.

We then computed the optimal shape parameter and wetting fraction for a large range of values of TANGO1 bending rigidity, $\kappa_T$, linactant strength, $\Delta\lambda$, and spontaneous curvature, $c_0$. These are presented in the form of shape diagrams in *Figure 6B,C* and *Figure 6—figure supplement 2A*. We find that, although none of these properties control the transition between shallow buds ($\eta < 1/2$) and large intermediates ($\eta > 1$), they are key in determining the capping-uncapping transition. Remarkably, we find once again that TANGO1 capping significantly expands the regions where stable shallow buds are energetically favored. In addition, the presence of a TANGO1 ring capping COPII subunits induced the formation of shapes with larger neck radii (*Figure 6B,C*, and *Figure 6—figure supplement 1B*). These results provide further support to our hypothesis that by capping COPII coats, TANGO1 stabilizes the formation of open buds preventing therefore the formation of fully budded spherical carriers. Such a stable wide neck would facilitate procollagen packing into the nascent intermediate.

## Outward force facilitates the transition from TANGO1-stabilized buds to large transport intermediates

Finally, we examined how the application of a force directed toward the cytosolic side of the bud can facilitate the growth of a TANGO1-capped transport intermediate. The existence of such a force, although speculative in the context of procollagen export, could have several possible origins. For instance, molecular motors linking the membrane to the cytoskeleton could potentially produce such pulling forces. Another hypothetical source of force is linked to the folding of triple-helical procollagen in the ER lumen. TANGO1 recruits procollagen by binding its chaperone HSP47 (*Ishikawa et al., 2016*), which preferentially interacts with triple-helical procollagen (*Tasab et al., 2000*). Procollagen folds into a rigid triple-helix from the C terminus to the N terminus in a zipper-like manner (*Engel and Prockop, 1991*). Therefore, HSP47, and correspondingly TANGO1, binding sites are generated on the procollagen as it folds. We thus hypothesize that the interface between TANGO1/HSP47 and procollagen could act as a zipper helping in the elongation of triple helical rigid procollagen. The C termini of procollagens are often associated with the ER membrane (*Beck et al., 1996*). It is therefore possible that a ring of TANGO1 would constrain the procollagens in a bundle with the folded C termini oriented toward the growing tip of the bud. As procollagens fold, newly generated HSP47/TANGO1-binding sites will be more distant to the growing tip of the bud. The net result of this process would be procollagen applying a force directed toward the cytosol (and exerting an opposite reaction force on the TANGO1 ring). The isotropic TANGO1 ring structure would prevent the generation of any net torque and therefore of any tilt of the procollagen molecules.

Following the same protocol as above, we computed the optimal shape parameter as a function of the intensity of an outward point force for various COPII polymerization energies and membrane tensions. As shown in *Figure 7A,B*, we found that having a point force shifts the capping-uncapping and transport intermediate growth transitions toward lower COPII binding energies and higher membrane tensions. Interestingly, we can obtain an analytical estimate of the transition between incomplete shallow buds and large transport intermediates. To that end, we consider the situation

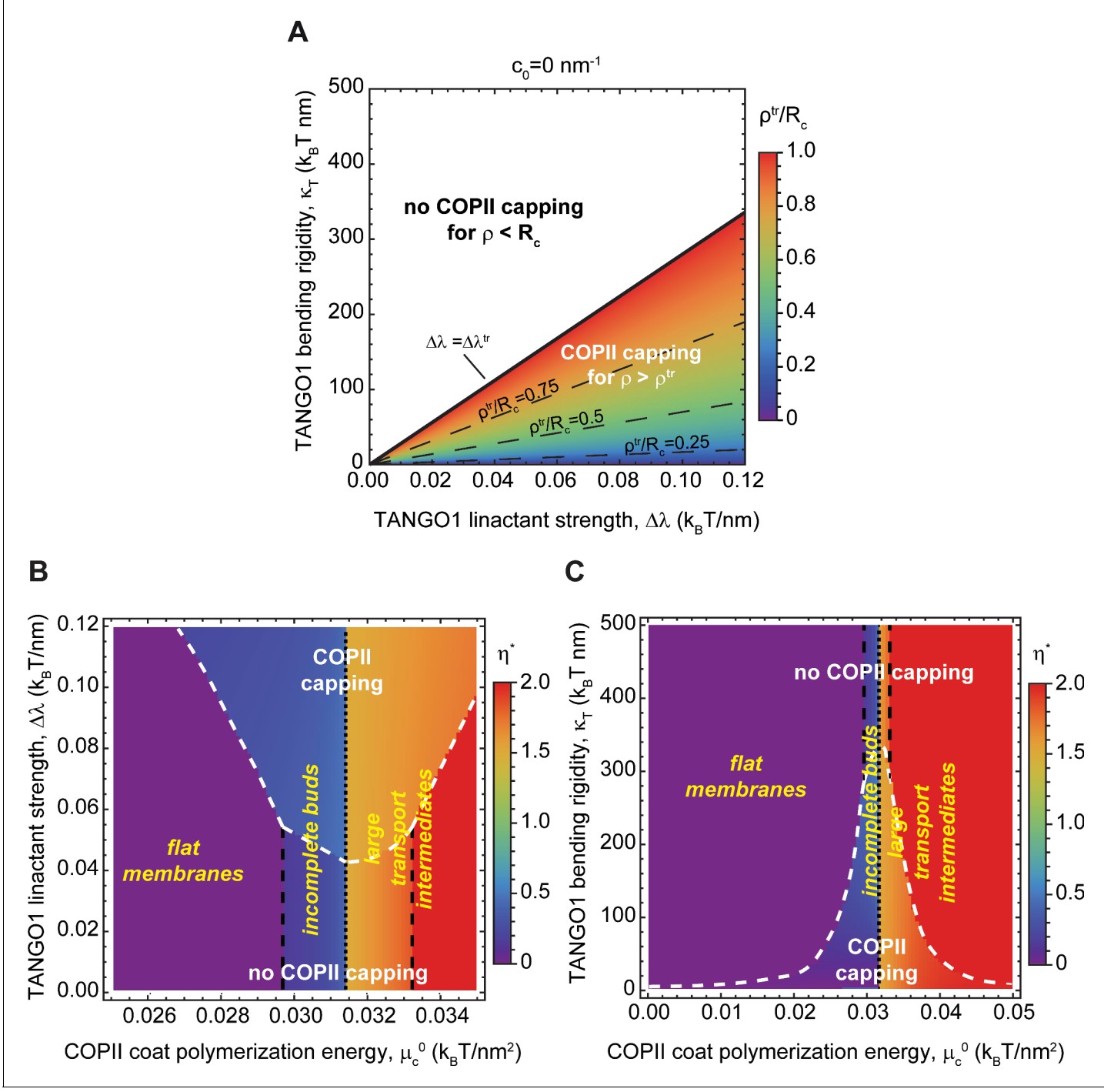

**Figure 6.** Capping is controlled by TANGO1 bending rigidity and linactant strength. (**A**) Capping-uncapping phase diagram where the membrane neck radius at the radius at the trasition, $\rho^{tr}$ (color-coded), given by *Equation 4*, is plotted against TANGO1 linactant strength, $\Delta\lambda$, and the bending rigidity, $\kappa_T$, for a filament of zero spontaneous curvature. Below the critical value of TANGO1 linactant strength, $\Delta\lambda^{tr}$, no capping occurs (no functional TANGO1 rings). Isolines for different values of $\rho^{tr}$ are indicated by the denoted dashed lines. (**B,C**) Shape diagrams indicating the shape of minimal free energy, represented by the optimal shape parameter, $\eta^*$ (color-coded), as a function of the COPII coat polymerization energy, $\mu_c^0$, and of the TANGO1 linactant strength, $\lambda$ (**B**), or the TANGO1 filament bending rigidity, $\kappa_T$ (**C**). Capping-uncapping transitions are marked by thick, dashed, white lines, whereas the different shape transitions are denoted by the different dashed lines.

The online version of this article includes the following figure supplement(s) for figure 6:

**Figure supplement 1.** Capping is controlled by TANGO1 bending rigidity and linactant strength.
**Figure supplement 2.** Capping is controlled by TANGO1 filament spontaneous curvature.

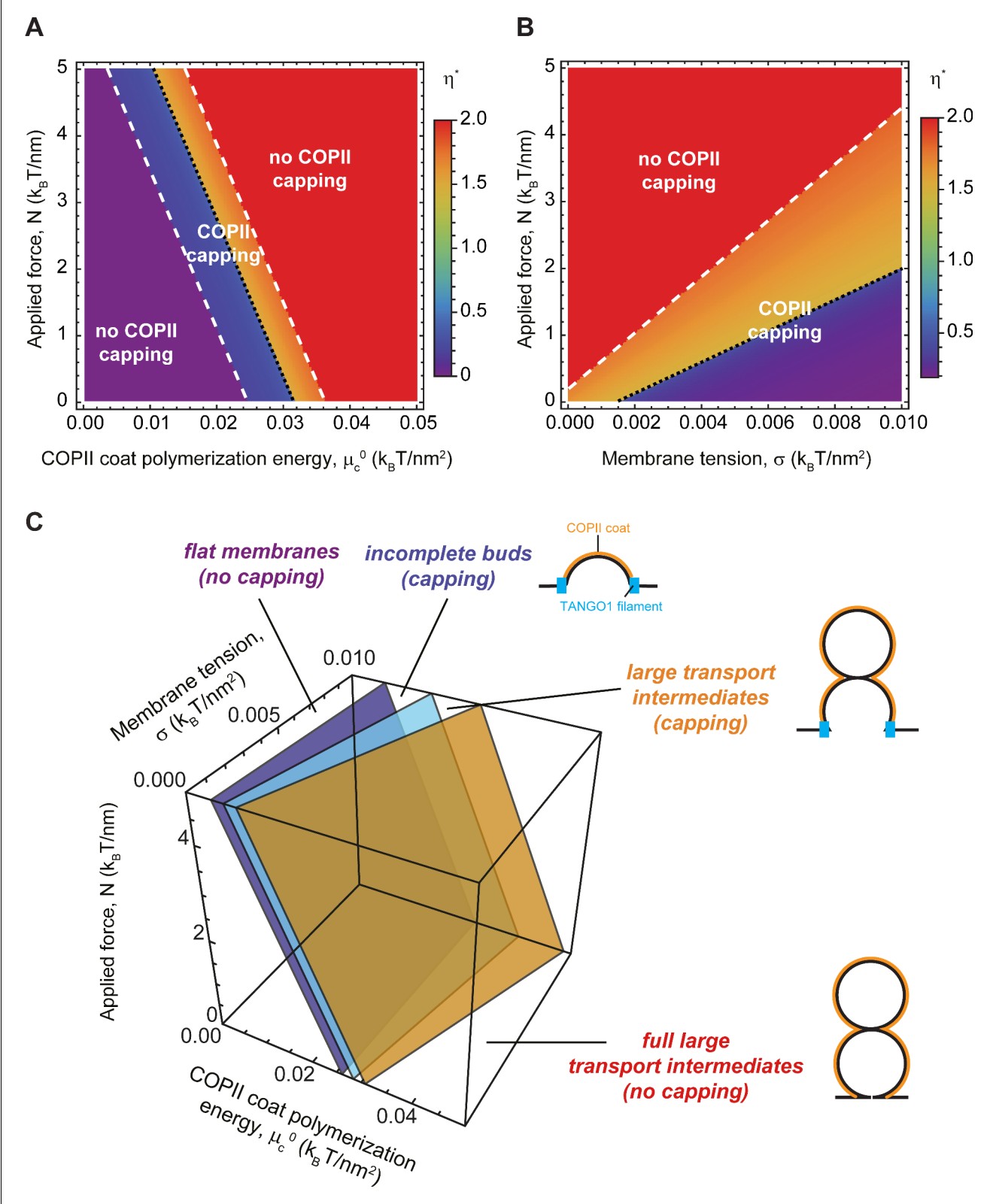

**Figure 7.** Outward force shifts capping transitions and facilitates bud formation. (A, B) Shape diagrams showing the optimal shape parameter, $\eta^*$ (color coded), as a function of the applied force, $N$, and the COPII coat polymerization energy, $\mu_c^0$, for $\sigma = \sigma_{ER} = 0.003 \; k_B T/nm^2$ (A); or the membrane tension, $\sigma$, for $\mu_c^0 = 0.03 \; k_B T/nm^2$ (B). (C) Three-dimensional shape diagram, indicating the transition zones between flat membranes ($\eta^* = 0$), incomplete buds ($\eta^* < 0$), and large transport intermediates ($\eta^* > 1$) – as given by *Equation 5* (light blue surface) and by numerical solution of model B (dark blue and

*Figure 7 continued on next page*

*Figure 7 continued*

orange surfaces) –, as a function of the COPII coat polymerization energy, $\mu_c^0$, membrane tension, $\sigma$, and applied force, $N$. Unless otherwise specified, the elastic parameters used for all the calculations shown in (**A–C**) are listed in *Appendix 2—table 1*.

where the free energy of an incomplete bud of shape parameter $\eta$, is equal to the free energy of a larger intermediate with an extra pearl, given by the shape parameter $\eta + 1$. *Equation 2* then gives

$$\mu_c^0 - 2\frac{\kappa_b}{R_c^2} + \frac{N}{2\pi R_c} - \sigma = 0, \tag{5}$$

which is independent of the structural details of the TANGO1 ring, as we have seen before (*Figure 6B,C*). The solution to this parametric equation is represented by the plane in light blue shown in *Figure 7C*. For completeness, in *Figure 7C* we also show the numerically found transition surfaces between flat membranes and shallow buds (dark blue surface, *Figure 7C*) and between large capped intermediates and full uncapped intermediates (orange surface, *Figure 7C*). Additionally, we rely on *Equation 5* to define the critical force $N^* = 2\pi R_c\left(\sigma - \mu_c^0 + 2\kappa_b/R_c^2\right)$ at which the transition to a large transport intermediate is triggered. For the experimental values reported in *Appendix 2—table 1* and $\mu_c^0 = 0.03 k_B T/nm^2$, we obtain $N^* \sim 0.34\frac{k_B T}{nm} = 1.4\,pN$, which, for comparison, is in the same order of magnitude of the force generated by molecular actin polymerization (*Footer et al., 2007*), and an order of magnitude smaller than typical force required to extract a membrane tethers (*Derényi et al., 2002*). Taken together, our results indicate that the formation of large transport intermediates can be greatly facilitated in the presence of TANGO1 by forces of physiological ranges oriented toward the cytosol. This mechanism could be complementary to the membrane tension regulation mechanism described above.

## Discussion

### A model for TANGO1-mediated procollagen export from the ER

Here we delineated a feasible biophysical mechanism of how TANGO1 contributes to the formation of procollagen-containing transport intermediates at the ER. In particular, we presented two complementary models to study this process. The first model (Model A) is a dynamic model that does not impose any particular geometry for the intermediate shape. We next complemented this dynamic model with a more simplified but analytically tractable equilibrium model (Model B), which assumes that COPII polymerizes into lattices of spherical geometry. Physically, these models can be understood in terms of a competition between different driving forces. Each of these forces can either prevent or promote the elongation of procollagen-containing transport intermediates. The results obtained with these two complementary models of large transport intermediate formation reinforce the notion that TANGO1 rings serve to modulate the formation of COPII carriers for the export of bulky cargoes.

We previously showed that TANGO1 forms rings at ERES surrounding COPII components (*Raote et al., 2017*). We also revealed interactions that are required for TANGO1 ring formation, which are also important to control TANGO1-mediated procollagen export from the ER (*Raote et al., 2018*). However, it still remained unclear how TANGO1 rings organize and coordinate the budding machinery for efficient procollagen-export. Here, we propose that TANGO1 rings form at ERES by assembling as a filamentous structure made of different components of the TANGO1 family of proteins (*Figure 1B,C*). Evidence for the existence of linear assemblies of transmembrane proteins has indeed been reported in the context of transmembrane actin-associated (TAN) lines that couple outer nuclear membrane components to actin cables (*Luxton et al., 2010*). The components of the TANGO1 filament have an affinity for COPII subunits (*Saito et al., 2009*), leading to a capping of the periphery of a COPII lattice (*Glick, 2017*; *Raote et al., 2018*). This effectively reduces the line energy of the COPII coat and therefore we propose that, by capping COPII lattices, TANGO1 filaments can act as linactants (*Figure 1B,C*). In the context of HIV gp41-mediated membrane fusion, linactant compounds, such as vitamin E, lower the interfacial line tension between different membrane domains to inhibit HIV fusion (*Yang et al., 2016*).

When the association of TANGO1 with COPII subunits was abrogated –either by expressing TANGO1-ΔPRD or by silencing the expression of SEC23A–, TANGO1 formed either smaller rings or long linear filamentous structures or planar clusters (*Raote et al., 2018*). In cells expressing TANGO1-ΔPRD, the interaction between one of the filament components, TANGO1, and the COPII subunits is abolished. However, a TANGO1 filament can still be formed because this mutant still interacts with other TANGO1 or cTAGE5 proteins (*Raote et al., 2018*). In this situation, the filament proteins cTAGE5 (*Saito et al., 2011*; *Saito et al., 2014*) and TANGO1-Short (*Maeda et al., 2016*) can still bind the COPII component Sec23A. Because the affinity to bind to the peripheral COPII sub-units is reduced, the filaments should be less line-active, and therefore less able to cap COPII com-ponents. Our theoretical results presented here predict that decreasing the TANGO1 filament linactant strength, $\lambda$, prevents capping of the peripheral COPII components and therefore results in a lower probability of having open carriers (*Figure 6*), which can lead to the observed defects in terms of ring structure and procollagen export.

Our results show that the formation of TANGO1 rings helps stabilize the COPII bud neck (*Figures 2* and *5*). This, we suggest, could allow for an efficient recruitment of procollagen molecules by TANGO1 to the exit site. It is not clear how long it takes for a procollagen molecule to fully fold into an export-competent triple helix after its translation into the ER, and it is possible that TANGO1 can act as a sensor of procollagen folding to couple it with the export machinery. However, in experi-ments where type I procollagen triple-helical stabilization and ER export were synchronized, fully folded procollagen required ~15 min to reach the early Golgi compartment (*McCaughey et al., 2019*). Since the half-lives of COPII components on the ER membrane are of the order of a few sec-onds (*Forster et al., 2006*), it seems likely that there is a mechanism in place to stabilize open nascent carriers, which helps to fully pack complex cargoes. Our theoretical results (*Figure 2*) sup-port the proposed mechanism by which TANGO1 rings arrest the growth of standard COPII carriers and stabilize open shallow buds for the efficient packaging of procollagens (*Figure 8A,B*).

Once a TANGO1 ring forms and stabilizes a shallow bud, the results of our model indicate that carrier expansion can proceed via at least two different, non-exclusive scenarios (*Figure 8C,D*): (i) local reduction of the membrane tension (*Figures 3–5*) and (ii) appearance of a directed force applied at the growing carrier and pointing toward the cytosol (*Figure 7*). Additionally, it is also pos-sible that carrier growth occurs by an increase in the polymerization ability of COPII coats (*Figure 5*). It is possible that TANGO1 can directly or indirectly control each of these mechanisms (*Ma and Goldberg, 2016*; *Raote et al., 2018*). Notably, the TANGO1 ring properties, such as the linactant strength or the filament bending rigidity, are not drivers for the incomplete bud to large transport intermediate transition (*Figure 6*) but rather, they enhance the regions of the parameter space where carrier shapes with wide-open necks are stabilized, as opposed to fully formed vesicles or unassembled coats (*Figure 5A,B*).

The results of our models indicate that a transient reduction of the ER membrane tension can induce a transition from a stable shallow bud encircled by a TANGO1 ring to a large transport inter-mediate, that is not reabsorbed upon recovery of the tension (*Figures 3* and *4*). It is worth noting that, despite the different modeling and implementation choices taken in these two approaches, the values of the membrane tension at which the predictions agree only differ by a factor two. This dif-ference could certainly be further reduced by undergoing a thorough analysis of the computational model's parameter space. However, given the computational cost of such study and the uncertainty of the experimental estimates for the ER and Golgi membrane tensions, seeking such a quantitative match between two qualitative models would only bring very limited insights. Finally, whether and how such tension regulation occurs at the level of the ERES still remains to be fully resolved. Never-theless, we suggest a possible way by which TANGO1 could act as the membrane tension regulator at the ERES. We propose that the fusion of ERGIC53-containing membranes tethered by the TANGO1 TEER domain (*Raote et al., 2018*) would be the trigger for carrier growth. In particular, the ER-specific SNARE protein Syntaxin18 and the SNARE regulator SLY1, both of which are involved in membrane fusion reactions at the ER, are also required for procollagen export in a TANGO1-dependent manner (*Nogueira et al., 2014*). Fusion of ERGIC membranes to the sites of procollagen export would lead to a local and transient reduction of the membrane tension (*Sens and Turner, 2006*). In this scenario, TANGO1 would act as a regulator of membrane tension homeostasis to control procollagen export at the ERES (*Figure 8C,E*). We have recently proposed that ERGIC membranes fuse directly or adjacently to the growing transport intermediate to allow for

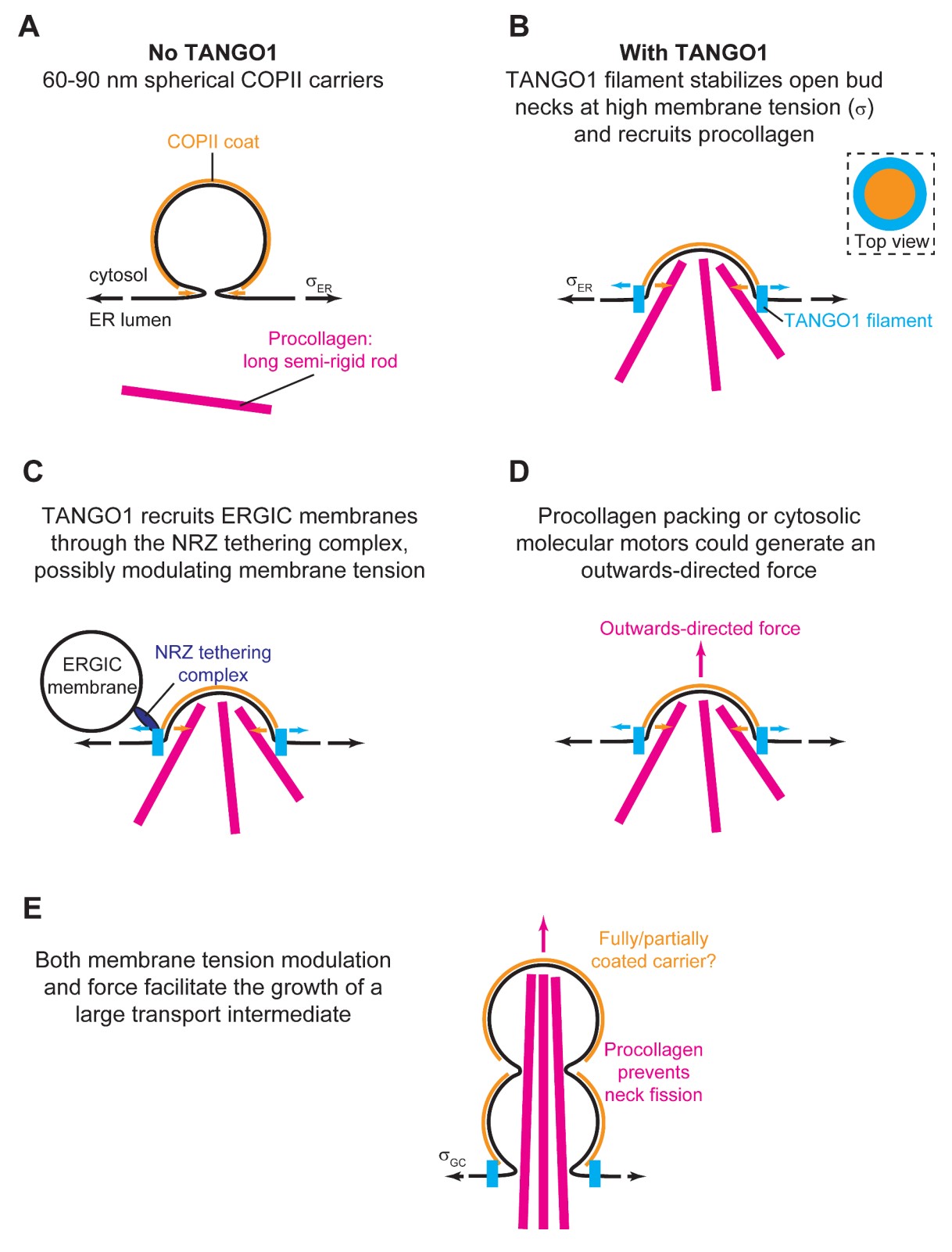

**Figure 8.** Schematic working model of TANGO-mediated bulky cargo export. (**A**) In absence of functional TANGO1, COPII-coated (orange) spherical vesicles assemble normally, generating spherical carriers of between 60–90 nm in size. In absence of TANGO1, procollagens (magenta) cannot be packed into such small carriers. (**B**) In presence of a functional TANGO1 (light blue), a filament forms capping the base of a growing COPII patch (see top view in the top right subpanel) and packages procollagens to the export sites. At the high membrane tension of the ER, our results showed that

*Figure 8 continued on next page*

*Figure 8 continued*

this TANGO1 fence stabilizes a shallow bud, which might allow the efficient packaging of export-competent procollagens. (C) The NRZ complex (dark blue), which is recruited to the procollagen export sites by the TANGO1 TEER domain, tethers ERGIC53-containing membranes. (D) A possible outwards-directed force (magenta arrow) facilitates the generation of a large transport intermediate. (E) A combination of membrane tension decrease (possibly mediated by fusion of ERGIC membranes), outwards-directed force, and the polymerization of COPII lattices, contribute to the growth of a large transport intermediate commensurate with long semi-rigid procollagen molecules. The actual shape of the transport intermediate and whether it is fully or only partially coated still remain unknown.

membrane addition and tension release; and showed that compartment mixing can be arrested by the TANGO1 ring serving as a diffusion barrier (*Raote et al., 2020*). In our models, we have not included the effects of a partial diffusion barrier at the base of the growing carrier. Although these effects could lead to changes in the dynamics of carrier growth as discussed below, we do not expect any major qualitative changes in our proposed mechanisms of carrier elongation.

From the free energy profile shown in *Figure 4A*, we also noticed that the system needs to overcome an energy barrier to reach the globally stable state. Hence, depending on the dynamics of the system, this can be kinetically trapped into a locally stable deep bud (yellow shape in *Figure 4A*, right panel). Although our analytical equilibrium model cannot account for the dynamics of such transitions, we can give some estimates. First, mechanical equilibration of the membrane shape ($\tau_{mech}$) can be theoretically estimated to be of the order of milliseconds (see e.g. *Campelo et al., 2017*; *Sens and Rao, 2013*), which is also in accordance to our own results from model A, where shape changes are found within few milliseconds (*Figure 3*). Second, the diffusive behavior of membrane tension has been measured at the plasma membrane of HeLa cells, with a diffusion coefficient of $D_\sigma = 0.024 \; \mu m^2/s$ (*Shi et al., 2018*), which gives a characteristic diffusion time of the order of $\tau_\sigma \sim 1 \; s$ (using characteristic length scales of $\sim 100 - 500 \; nm$). Since $\tau_\sigma \gg \tau_{mech}$, membrane shape equilibrates much faster than the recovery of the membrane tension. Moreover, our recent findings that TANGO1 can induce a diffusion barrier at the base of the growing bud (*Raote et al., 2020*), suggest that the dynamics of tension equilibration can be even slower.

In parallel, we also hypothesize a situation where TANGO1 rings help pushing procollagen molecules into the growing carrier and couple this pushing force to procollagen folding, through the chaperone HSP47 (*Figure 8D,E*). Because HSP47 chaperone assists in folding (and hence in rigidifying) procollagen, it is tempting to speculate that the trimerization of procollagen occurs concomitantly to the its export and that the physical interaction between TANGO1 rings and procollagen/HSP47 could serve as a means to couple procollagen folding to force production. Although the existence of this pushing force is largely speculative, it could, according to our model, promote formation of a large intermediate and hence TANGO1 could act as a sensor of procollagen folding to couple it with the export machinery.

Finally, the results of our dynamic model show that the shape of the elongated carrier is a pearled tubule (*Figure 3D* and *Figure 3—video 1*). The fission of the upstream pearls is prevented in our model by a steric hindrance of the procollagen molecules packed inside the tubule (*Figure 8E*). Since TANGO1 is the receptor molecule that binds procollagen and packs it to facilitate its export (*Saito et al., 2009*), TANGO1 is indirectly required to prevent the fission of upstream necks. The results of our dynamic model indicate that during the elongation of the structure (*Figure 3D* and *Figure 3—video 1*). We previously proposed that the large transport intermediate is converted into a tunnel connecting ER to the ERGIC/early Golgi cisternae (*Raote and Malhotra, 2019*). If so, an appealing possibility is that, at this stage, the role of a TANGO1 ring would be to prevent lipid mixing between these transiently connected organelles by acting as a diffusion barrier (*Raote et al., 2020*). Finally, after procollagen delivery to the acceptor compartment, fission of the neck between the first two pearls of the tubule would not be sterically prevented anymore and therefore the tunnel could be disconnected. Future work will be required to help elucidate the precise control of the timing between these proposed fusion and fission events.

What controls organelle size in the context of intracellular trafficking? There has been a lot of work on what set the size of organisms, the size of tissues in an organism, and the size of cells in a tissue (*Guertin and Sabatini, 2006*; *Marshall et al., 2012*). However, there has been less work toward elucidating what sets the size of organelles relative to the cell. Extensive cargo transfer while trafficking bulky cargoes such as collagens leads to large amounts of membrane being transferred

from one organelle to another. To maintain organelle homeostasis, loss of membrane from a compartment has to be concomitantly compensated by membrane acquisition from the biosynthetic pathway or by trafficking from other organelles; the arrival and departure of membrane at each compartment has to be efficiently balanced. How is this homeostatic balance controlled? Changes in membrane tension have been described to affect rates of exocytosis and endocytosis at the plasma membrane (*Apodaca, 2002*; *Kosmalska et al., 2015*; *Wu et al., 2017*) and of growth of membrane buds and tubes in general (*Derényi et al., 2002*; *Cuvelier et al., 2005*). Interestingly, a theoretical model has also established a crucial role for membrane tension in modulating the budding of clathrin-coated vesicles (*Hassinger et al., 2017*). Furthermore, it has been recently proposed that Atlastin-mediated maintenance of ER membrane tension is required for the efficient mobility of cargo proteins (*Niu et al., 2019*); and that the formation of intra-lumenal vesicles in endosomes is also regulated by membrane tension (*Mercier et al., 2020*). However, control of endomembrane trafficking by membrane tension still remains challenging to study experimentally and hence remains incompletely understood. We suggest that inter-organelle hubs, such as a TANGO1-scaffolded ERES-ERGIC, control the local tension homeostasis at specific membrane sub-domains and regulate the membrane flux between these organelles.

## Proposal of experimental approaches to test the model

We have proposed and analyzed a theory by which TANGO1 assembles into a ring at the ERES to facilitate cargo export. The results of the theoretical biophysical models presented are compatible with our current understanding of the biology of this complex process. Importantly, our analyses are meant to help design experiments to further test our hypotheses. We speculate that a TANGO1 ring functions as a hub to collect and concentrate procollagens, to stabilize open COPII lattices, and to recruit and fuse membranes thus alleviating the ERES tension while promoting the export of procollagens from the ER. Although many of these individual events are based on experimental evidence, we lack an understanding of how collagens are collected and then percolated into the growing intermediate, whether the binding of TANGO1 to the rims of growing COPII coats acts as a linactant, the involvement of tension, the mechanism of tension sensing, and finally how these events are coordinated to cause procollagen export. It is important to be able to ask these questions because they highlight the gaps in our understanding of the process of cargo export at the ER. We propose a set of experimental procedures to test our hypotheses. The new experimental data will undoubtedly improve our understanding of how cells engage to export cargoes based on their size, volume, and the overall cellular needs.

### Does TANGO1 form a linear or quasi-linear filament held together by lateral protein-protein interactions?

Although our *hypothesis 3* (see Model Development section) is based on a number of indirect experimental observations, to our knowledge, there is no direct evidence for the existence of TANGO1 filaments. A first step to address this question will be to resolve the stoichiometry of the TANGO1 family proteins within a TANGO1 ring. Controlled photobleaching (*Lee et al., 2012*) or DNA-PAINT (*Stein et al., 2019*) of the single-labeled, endogenously expressed proteins could allow the recording of the number and spatial positions of single fluorophores in individual TANGO1 rings. These results, after complete quantitative reconstruction of all the single molecule signals, should provide an absolute stoichiometry and ultra-resolved structure of TANGO1 organization in the ERES. Ultimately, in vitro reconstitution of TANGO1 ring formation in synthetic lipid bilayers by using recombinant proteins will be of paramount importance to experimentally observe the formation of TANGO1 filaments, assess the minimal components required for their formation, and eventually measure the elastic properties of a TANGO1 filament.

### Does the TANGO1-directed fusion of ERGIC membranes modulate the ERES membrane tension for procollagen export?

Future efforts in applying cutting-edge, super-resolution multicolor live-cell microscopy (*Bottanelli et al., 2016*; *Ito et al., 2018*; *Liu et al., 2018*; *Schroeder et al., 2019*) will help monitor the fusion of ERGIC membranes to the ER and couple these events to the formation of procollagen-containing transport intermediates. In addition, our hypothesis of TANGO1-mediated regulation of

membrane tension is based upon the premise that fusion of ERGIC-53-containing vesicles to the procollagen export sites locally decreases the membrane tension. A recently established fluorescent membrane tension sensor (*Colom et al., 2018*; *Goujon et al., 2019*) could provide a means to monitor such effects in relation to procollagen export.

### Is there an outwards-directed force contributing to transport intermediate elongation?

It has been shown that procollagen export from the ER does not require the presence of an intact microtubule network (*McCaughey et al., 2019*). However, the involvement of other force-producing agents, such as actin-myosin networks, remains unknown. The identification of additional physiological interactors of TANGO1 by proximity-dependent labeling assays, such as BioID (*Roux, 2018*), and the subsequent screening for candidates that can exert such forces could help identify possible molecular players involved in force-generation. However, it is important to stress that our model can explain the formation of large transport intermediates even in the absence of an applied force (*Figure 7*).

### What is the shape of the transport intermediate that shuttles collagens from the ER to the ERGIC/Golgi complex?

Our full dynamic model predicts that the TANGO1-mediated procollagen-containing transport intermediates are shaped as a pearled tubule (*Figure 3D*). That this prediction holds true in tissue cultured cells, or alternatively that the transport intermediates have a more cylindrical geometry, as proposed from in vitro data of COPII polymerization in the presence of PPP sequences, such as those in the TANGO1 PRD (*Ma and Goldberg, 2016*; *Hutchings et al., 2018*), will reveal how TANGO1 modulates COPII polymerization and coat flexibility. To this end, three-dimensional, multicolor super-resolution microscopy techniques, such as 3D single molecule localization microscopy (3D-SMLM) or 3D stimulated emission depletion (3D-STED) microscopy, could provide sufficient resolution to map the three-dimensional morphology of the transport intermediates. Recent efforts by using 3D-SMLM and correlative light and electron microscopy (CLEM) have revealed the existence of large procollagen-containing structures (*Gorur et al., 2017*; *Yuan et al., 2018*). However, a recent report suggested that those structures were directed for lysosomal degradation and not for trafficking to the Golgi complex (*Omari et al., 2018*). By contrast, direct transport of procollagen between the ER and the Golgi complex by a short-loop pathway in the absence of large vesicles has been recently proposed (*McCaughey et al., 2019*), opening to the possibility of a direct tunneling mechanism for trafficking proteins between compartments (*Raote and Malhotra, 2019*). Eventually, the use of modern electron microscopy techniques such as cryo-electron tomography (*Beck and Baumeister, 2016*) or focused ion beam-scanning electron microscopy (FIB-SEM) (*Nixon-Abell et al., 2016*) will help solve this issue on the morphology of the transport intermediates that shuttle procollagens from the ER to the Golgi complex.

## Concluding summary

In summary, we proposed a theoretical mechanical model that explains how TANGO1 molecules form functional rings at ERES, and how these TANGO1 rings assemble the machinery required to form a large transport intermediate commensurate with the size of procollagens. We envision that our hypotheses and the predictions of our model will guide new lines of experimental research to unravel mechanisms of COPII coats organization for the export of complex cargoes out of the ER.

## Acknowledgements

We thank Javier Diego Íñiguez, Iván López-Montero, and members of the Garcia-Parajo lab for valuable discussions. MF Garcia-Parajo and V Malhotra are Institució Catalana de Recerca i Estudis Avançats professors at ICFO-Institut de Ciencies Fotoniques and the Centre for Genomic Regulation (CRG), respectively. M Chabanon, MF Garcia-Parajo and F Campelo acknowledge support from the Government of Spain (FIS2015-63550-R, FIS2017-89560-R, BFU2015-73288-JIN, RYC-2017–22227, and PID2019-106232RB-I00/10.13039/501100011033; Severo Ochoa CEX2019-000910-S), Fundació Cellex, Fundació Mir-Puig, and Generalitat de Catalunya (CERCA, AGAUR), ERC Advanced Grant

NANO-MEMEC (GA 788546) and LaserLab 4 Europe (GA 654148). I Raote and V Malhotra acknowledge funding by grants from the Ministerio de Economía, Industria y Competitividad Plan Nacional (BFU2013-44188-P) and Consolider (CSD2009-00016); support of the Spanish Ministry of Economy and Competitiveness, through the Programmes 'Centro de Excelencia Severo Ochoa 2013–2017' (SEV-2012–0208) and Maria de Maeztu Units of Excellence in R and D (MDM-2015–0502); and support of the CERCA Programme/Generalitat de Catalunya. I Raote, MF Garcia-Parajo, V Malhotra., and F Campelo acknowledge initial support by a BIST Ignite Grant (eTANGO). I Raote acknowledges support from the Spanish Ministry of Science, Innovation and Universities (IJCI-2017–34751). This work reflects only the authors' views, and the EU Community is not liable for any use that may be made of the information contained therein. M Arroyo and N Walani acknowledge the support of the European Research Council (CoG-681434), and M Arroyo that of the Generalitat de Catalunya (2017-SGR-1278 and ICREA Academia prize for excellence in research) and of the Spanish Ministry of Economy and Competitiveness, through the Severo Ochoa Programme (CEX2018-000797- S).

## Additional information

### Competing interests

Felix Campelo: Reviewing editor, *eLife*. Vivek Malhotra: Senior editor, *eLife*. The other authors declare that no competing interests exist.

### Funding

| Funder | Grant reference number | Author |
|---|---|---|
| Government of Spain | "Severo Ochoa" Programme (CEX2019-000910-S) | Morgan Chabanon Maria F Garcia-Parajo Felix Campelo |
| Government of Spain | BFU2015-73288-JIN | Maria F Garcia-Parajo Felix Campelo |
| Government of Spain | FIS2015-63550-R | Maria F Garcia-Parajo Felix Campelo |
| Government of Spain | FIS2017-89560-R | Morgan Chabanon Maria F Garcia-Parajo Felix Campelo |
| Fundacio Privada Cellex | | Morgan Chabanon Maria F Garcia-Parajo Felix Campelo |
| Fundacio Privada Mir-Puig | | Morgan Chabanon Maria F Garcia-Parajo Felix Campelo |
| Generalitat de Catalunya | CERCA program | Ishier Raote Morgan Chabanon Maria F Garcia-Parajo Vivek Malhotra Felix Campelo |
| European Research Council | ERC Advanced Grant (GA 788546) | Morgan Chabanon Maria F Garcia-Parajo Felix Campelo |
| European Research Council | LaserLab 4 Europe GA 654148 | Morgan Chabanon Maria F Garcia-Parajo Felix Campelo |
| Spanish Government | BFU2013-44188-P | Ishier Raote Vivek Malhotra |
| Spanish Government | Consolider CSD2009-00016 | Ishier Raote Vivek Malhotra |
| Spanish Government | Severo Ochoa Program SEV-2012-0208 | Ishier Raote Vivek Malhotra |

| | | |
|---|---|---|
| Spanish Government | Maria de Maeztu MDM-2015-0502 | Vivek Malhotra |
| BIST | Ignite grant eTANGO | Ishier Raote<br>Maria F Garcia-Parajo<br>Vivek Malhotra<br>Felix Campelo |
| Spanish Ministry of Science and Innovation | IJCI-2017-34751 | Ishier Raote |
| Spanish Ministry of Science and Innovation | RYC-2017-22227 | Felix Campelo |
| European Research Council | CoG-681434 | Nikhil Walani<br>Marino Arroyo |
| Generalitat de Catalunya | 2017-SGR-1278 | Marino Arroyo |
| ICREA | ICREA academia | Marino Arroyo |
| Ministerio de Economía y Competitividad | Severo Ochoa Program CEX2018-000797- S | Marino Arroyo |
| State Research Agency | PID2019-106232RB-I00/10.13039/501100011033 | Morgan Chabanon<br>Felix Campelo |
| Ministerio de Economía y Competitividad | Severo Ochoa Program SEV-2012–0208 | Ishier Raote<br>Vivek Malhotra |
| María de Maeztu Unit of Excellence | MDM-2015–0502 | Ishier Raote<br>Vivek Malhotra |

The funders had no role in study design, data collection and interpretation, or the decision to submit the work for publication.

### Author contributions

Ishier Raote, Conceptualization, Formal analysis, Investigation, Writing - original draft, Writing - review and editing; Morgan Chabanon, Conceptualization, Software, Formal analysis, Investigation, Methodology, Writing - original draft, Writing - review and editing; Nikhil Walani, Software, Formal analysis, Investigation, Methodology, Writing - review and editing; Marino Arroyo, Conceptualization, Software, Supervision, Funding acquisition, Methodology, Project administration, Writing - review and editing; Maria F Garcia-Parajo, Conceptualization, Supervision, Funding acquisition, Methodology, Project administration, Writing - review and editing; Vivek Malhotra, Conceptualization, Supervision, Funding acquisition, Writing - original draft, Project administration, Writing - review and editing; Felix Campelo, Conceptualization, Formal analysis, Supervision, Funding acquisition, Investigation, Methodology, Writing - original draft, Project administration, Writing - review and editing

### Author ORCIDs

Ishier Raote [ID] https://orcid.org/0000-0002-5898-4896
Vivek Malhotra [ID] http://orcid.org/0000-0001-6198-7943
Felix Campelo [ID] https://orcid.org/0000-0002-0786-9548

### Decision letter and Author response

Decision letter https://doi.org/10.7554/eLife.59426.sa1
Author response https://doi.org/10.7554/eLife.59426.sa2

## Additional files

### Supplementary files

• Transparent reporting form

### Data availability

All data generated or analysed during this study are included in the manuscript and supporting files.

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

## Appendix 1

### Computational dynamic model of TANGO1-mediated bulky cargo export (Model A)

In this Appendix, we derive a dynamic model of a lipid bilayer whose spontaneous curvature is dictated by the diffusion and interaction of two membrane-bound species, and specialize it to COPII and TANGO1.

The proposed model extends the work of *Tozzi et al., 2019* to account for a second membrane-bound species and apply it to TANGO1-COPII complex assembly. We therefore focus our exposé on this novel aspect, and direct readers interested in the detailed underlying theory to *Tozzi et al., 2019*; *Arroyo et al., 2018*; *Rahimi and Arroyo, 2012*; *Torres-Sánchez et al., 2019*.

### Onsager's variational approach: energetics, dissipation, and power input

Our modeling approach is based on Onsager's variational formalism of dissipative dynamics (*Arroyo et al., 2018*; *Rahimi and Arroyo, 2012*). The fundamental principle consists in describing the time evolution of the system through a minimization process of energy released, energy dissipated, and energy exchanged by the system. In other words, if $\dot{\mathcal{F}}$ is the rate of change of free energy, $\mathcal{D}$ is the total dissipation potential, and $\mathcal{P}_{\text{ext}}$ is the power input, the rate of change of the system can be obtained by minimizing at each time point the Rayleighian functional

$$\mathcal{R} = \dot{\mathcal{F}} + \mathcal{D} + \mathcal{P}_{\text{ext}}. \tag{A1}$$

In this section, we define the energetic contributions to each of these quantities.

### Free energy

We consider a lipid bilayer as a material surface parametrized by $\mathbf{r}(\theta^\alpha, t)$, where $(\theta^1, \theta^2)$ are the Lagrangian surface coordinates, and $t$ is the time. The state variable associated with the mechanical energy of the system is $\mathbf{r}$, while the state variables associated with the chemical energy of the systems are the local area fractions of COPII and TANGO1, $\phi_c$ and $\phi_t$, respectively. Note that these latter are bounded and should satisfy $\phi_c > 0$, $\phi_t > 0$, and $0 \leq \phi_c + \phi_t \leq 1$.

### Mechanical energy

The bending energy of the membrane is described by the classical Helfrich model with spontaneous curvature (*Helfrich, 1973*; *Campelo et al., 2014*; *Chabanon et al., 2017*)

$$\mathcal{F}_{bend} = \int_\Gamma \frac{\kappa}{2}(J - C_c\phi_c)^2 dS, \tag{A2}$$

where $\kappa$ is the bending modulus, $J$ is the total curvature (twice the mean curvature, $H$) of the membrane, and the integration is performed over the entire membrane patch, $\Gamma$. Here we consider a local spontaneous curvature $C_c\phi_c$ resulting from the presence of COPII complexes. We assume a linear dependence of the spontaneous curvature on COPII local coverage, with $C_c = 2/R_c$ being the maximum spontaneous curvature induced by a full coverage of COPII ($\phi_c = 1$) with preferred radius of curvature $R_c$.

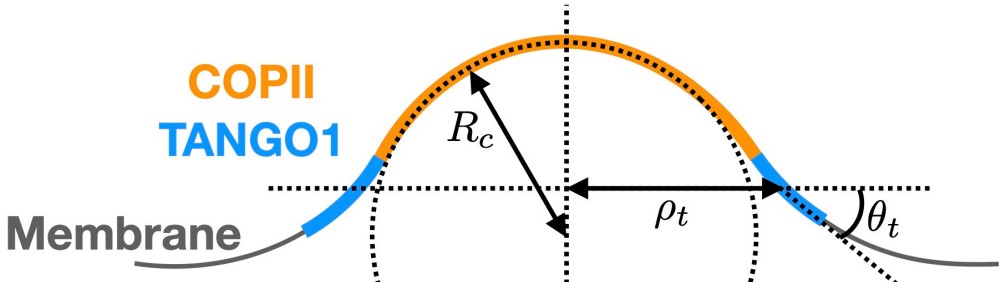

**Appendix 1—figure 1.** Schematic of a shallow COPII/TANGO1 bud and definition of the geometrical parameters favored by the proteins.

The total curvature $J$ is a function of the state variable $\mathbf{r}$. Briefly, from standard differential geometry (*Deserno, 2015*; *do Carmo, 2016*) we have that the tangent vectors at each point of the membrane are $\mathbf{g}_\alpha = \partial\mathbf{r}/\partial\theta^\alpha$. They define the natural basis of the tangent space, from which the covariant components of the metric tensor are obtained $g_{\alpha\beta} = \mathbf{g}_\alpha \cdot \mathbf{g}_\beta$. Additionally, the unit normal to the surface is $\mathbf{n} = (\mathbf{g}_1 \times \mathbf{g}_2)/\sqrt{g}$, where $g = \det(g_{\alpha\beta})$. From these definitions, one gets the components of the second fundamental form $k_{\alpha\beta} = \mathbf{n} \cdot \partial\mathbf{g}_\alpha/\partial\theta^\alpha$, whose invariants are the total curvature $J = \operatorname{tr}\mathbf{k} = k_{\alpha\beta}g^{\alpha\beta}$, and the Gaussian curvature $K = \det\mathbf{k} = k_{\alpha\beta}g^{\beta\gamma}$. Here $g^{\beta\gamma}$ are the components of the inverse of the metric tensor, obtained from $g_{\alpha\beta}g^{\beta\gamma} = \delta^\gamma_\alpha$. Note that for simplicity, we have neglected the contribution of the Gaussian curvature to the bending energy in *Equation (A2)*.

Based on our experimental observations (*Raote et al., 2018*), TANGO1 proteins are assumed to favor a ring-like conformation with a specific radius of curvature $\rho_t$ and a preferred angle with the plane of the ring $\theta_t$ (see *Appendix 1—figure 1*). As detailed later, we will restrict the model to axisymmetric shapes. Therefore for clarity, here we directly write an axisymmetric expression of the functional for the TANGO1 ring stiffness as

$$\mathcal{F}_{stif} = \int_\Gamma \left[ \frac{\kappa_\rho}{2}\left(\frac{1}{\rho} - \frac{1}{\rho_t}\right)^2 + \frac{\kappa_\theta}{2}(\theta - \theta_t)^2 \right]\phi_t dS, \tag{A3}$$

where $\kappa_\rho$ and $\kappa_\theta$ are, respectively, the stiffness coefficients associated with the ring curvature $1/\rho$ and angle with the membrane $\theta$.

## Chemical energy

We consider two distinct membrane-bound species representing COPII and TANGO1 proteins. They are described by continuous surface fractions $\phi_c$ and $\phi_t$, respectively. We consider the entropic mixing energy of the two proteins to be represented by a Flory–Huggins type energy such as

$$\mathcal{F}_{ent} = \int_\Gamma \frac{k_B T}{a_p}[\phi_c \ln\phi_c + \phi_t \ln\phi_t + (1 - \phi_c - \phi_t)\ln(1 - \phi_c - \phi_t)]dS, \tag{A4}$$

where $a_p$ is the molecular area of the proteins, assumed for simplicity to be identical for both proteins. The self-interaction and line energy of COPII proteins are

$$\mathcal{F}_c = \int_\Gamma \frac{\chi_c}{2a_p}\phi_c^2 + \frac{\Lambda_c}{2a_p}|\nabla\phi_c|^2 dS, \tag{A5}$$

where $\chi_c$ is negative for attractive interactions. The parameter $\Lambda_c$ ensures a length-scale associated with the interface of the COPII domain: the spatial gradient of $\phi_c$ is smoother for large values of $\Lambda_c$.

Similarly, we write the self-interaction and line energy of TANGO1 proteins as

$$\mathcal{F}_t = \int_\Gamma \frac{\chi_t}{2a_p}\phi_t^2 + \frac{\Lambda_t}{2a_p}|\nabla\phi_t|^2 dS. \tag{A6}$$

Finally, the interactions between COPII and TANGO1 proteins are

$$\mathcal{F}_{ct} = \int_{\Gamma} \frac{\chi_{ct}}{2a_p} \phi_c \phi_t + \frac{\Lambda_{ct}}{2a_p} \phi_t |\nabla \phi_c|^2 dS, \tag{A7}$$

where the first term represents the affinity between the two proteins and the second term represents the affinity of TANGO1 for the COPII coat boundary. Combining the second term of *Equation (A7)* with the second term of *Equation (A5)*, one can see that TANGO1 modulates the interfacial energy of COPII with an effective parameter $\Lambda_c + \Lambda_{ct}\phi_t$.

The total free energy is the sum of the protein and membrane contributions:

$$\mathcal{F} = \mathcal{F}_{bend} + \mathcal{F}_{stif} + \mathcal{F}_{ent} + \mathcal{F}_c + \mathcal{F}_t + \mathcal{F}_{ct}. \tag{A8}$$

## Dissipation mechanisms

Dissipation of mechanical energy occurs through in-plane shear stress of the lipid bilayer as it dynamically deforms under the action of a Lagrangian velocity of the membrane $\mathbf{V} = \partial \mathbf{r}/\partial t = \mathbf{v} + v_n\mathbf{n}$, where $\mathbf{v}$ and $v_n$ are its tangential and normal components respectively. This membrane 'flow' results in the time evolution metric tensor, whose time derivative in Lagrangian setting is the rate-of-deformation tensor of the surface (*Torres-Sánchez et al., 2019*)

$$\mathbf{d} = \frac{1}{2} \frac{\partial \mathbf{g}}{\partial t} = \frac{1}{2} \left( \nabla \mathbf{v} + \nabla \mathbf{v}^T \right) - v_n \mathbf{k}. \tag{A9}$$

The first term, involving the membrane tangential velocity $\mathbf{v}$ represents the contribution of the tangential flow to the membrane deformation. The last term, involving the normal velocity $v_n$, accounts for the shape change of the membrane. The rate of change of local area is $\mathrm{tr}(\mathbf{d}) = \nabla \cdot \mathbf{v} - v_n J$, which is zero for an inextensible membrane. We consider the lipid bilayer to be in a fluid phase that can be approximated by an interfacial viscous Newtonian fluid (*Arroyo and Desimone, 2009*). Neglecting intermonolayer friction, and assuming membrane inextensibility, the dissipation potential by in-plane shear stress takes the form

$$\mathcal{D}_{\mathrm{mech}} = \int_{\Gamma} \eta_m \mathbf{d} : \mathbf{d} \, dS, \tag{A10}$$

where $\eta_m$ is the in-plane viscosity of the lipid bilayer.

Dissipation of chemical energy occurs by protein diffusion along the membrane surface, described by the species diffusive velocities relative to the Lagrangian coordinates, $w_c$ and $w_t$ for COPII and TANGO1, respectively. Assuming for simplicity that the two proteins have the same molecular drag coefficient $\xi$ and surface area $a_p$, the chemical dissipative potential of the system is

$$\mathcal{D}_{\mathrm{chem}} = \int_{\Gamma} \frac{\xi}{2a_p} \left( \phi_c |\mathbf{w}_c|^2 + \phi_t |\mathbf{w}_t|^2 \right) dS. \tag{A11}$$

Given that the typical length scale of our system is well below the Saffman-Delbrück length scale (~1–10 μm), we can safely neglect dissipation arising from the friction between the membrane and the cytosol [*Arroyo and Desimone, 2009*]. The total dissipation potential of the system is $\mathcal{D} = \mathcal{D}_{\mathrm{mech}} + \mathcal{D}_{\mathrm{chem}}$.

### Power supplied

Mechanical power can only be supplied to our system through the boundary of the membrane patch $\partial \Gamma$ in the form of edge tractions and moments. Defining $\tau$ as the unit tangent vector along $\partial \Gamma$ so that $\nu = \tau \times \mathbf{n}$, the boundary tractions and moment power inputs are

$$\mathcal{P}_{\mathrm{mech}} = -\int_{\partial \Gamma} (F_\tau \mathbf{v} \cdot \tau + F_\nu \mathbf{v} \cdot \nu + F_n v_n) dl + \int_{\partial \Gamma} M\nu \cdot \dot{\mathbf{n}} dl, \tag{A12}$$

where $F_\tau$, $F_\nu$ and $F_n$ are the traction components at the boundary, $M$ is the bending moment per unit length, and $\dot{\mathbf{n}}$ is the material time derivative of the surface normal.

In this model, we assume that all proteins are membrane bound and provided at the boundary of the domain by a protein reservoir of fixed chemical potential. The chemical power supply is therefore written

$$\mathcal{P}_{\text{chem}} = \int_{\partial\Gamma} \left( \frac{\bar{\mu}_c^0}{a_p} \phi_c \mathbf{w}_c + \frac{\bar{\mu}_t^0}{a_p} \phi_t \mathbf{w}_t \right) \cdot \nu dl, \tag{A13}$$

where $\bar{\mu}_c^0$ and $\bar{\mu}_t^0$ are the fixed boundary chemical potentials of COPII and TANGO1, respectively. The total power supplied to the system is $\mathcal{P}_{\text{ext}} = \mathcal{P}_{\text{mech}} + \mathcal{P}_{\text{chem}}$.

## Governing dynamics

### Protein surface transport

Based on the definitions of the free energies, we define the energy density $W$ as $\mathcal{F} = \int_\Gamma W dS$. The chemical potentials of the two species can be written as $\bar{\mu}_i = a_p (W_{\phi_i} - \nabla \cdot W_{\nabla \phi_i})$, with $i = \{c, t\}$ (*Tozzi et al., 2019*). Here $W_{\phi_i}$ and $W_{\nabla \phi_i}$ are the partial derivatives of $W$ with respect to $\phi_i$ and $\nabla \phi_i$, respectively. The chemical potentials for each species are therefore

$$\bar{\mu}_c = -a_p \kappa C_c (J - C_c \phi_c) + k_B T \ln\left(\frac{\phi_c}{1 - \phi_c - \phi_t}\right) + \chi_c \phi_c - (\Lambda_c + \Lambda_{ct}\phi_t)\nabla^2\phi_c + \frac{\chi_{ct}}{2}\phi_t - \Lambda_{ct}\nabla\phi_t \cdot \nabla\phi_c, \tag{A14}$$

and

$$\bar{\mu}_t = \frac{a_p}{2}\left[\kappa_\rho\left(\frac{1}{\rho} - \frac{1}{\rho_t}\right) + \kappa_\theta(\theta - \theta_t)\right] + k_B T \ln\left(\frac{\phi_t}{1 - \phi_c - \phi_t}\right) + \chi_t \phi_t - \Lambda_t \nabla^2 \phi_t + \frac{\chi_{ct}}{2}\phi_c + \frac{\Lambda_{ct}}{2}|\nabla\phi_c|^2, \tag{A15}$$

respectively. The diffusive velocity of the species $i$ can be expressed as a function of the species chemical potential $\bar{\mu}_i$ by minimizing the Rayleighian with respect to $\mathbf{w}_i$, giving $\mathbf{w}_i = -\nabla\bar{\mu}_i/\xi$ (*Tozzi et al., 2019*). The strong form of the transport equations for the proteins on an incompressible surface is therefore

$$\xi\dot{\phi}_i - \nabla \cdot (\phi_i \nabla \bar{\mu}_i) = 0 \quad \text{with } i = \{c, t\}, \tag{A16}$$

with the diffusive flux for each species given by

$$-\phi_c \nabla\bar{\mu}_c = \left[a_p \kappa C_c \nabla J - \frac{\chi_{ct}}{2}\nabla\phi_t\right]\phi_c - \left[(\chi_c + a_p \kappa C_c^2 - \Lambda_{ct}\nabla^2\phi_t)\phi_c + k_B T \frac{1 - \phi_t}{1 - \phi_c - \phi_t}\right]\nabla\phi_c \tag{A17}$$
$$+ 2\Lambda_{ct}\nabla\phi_t\phi_c\nabla^2\phi_c + [(\Lambda_c + \Lambda_{ct}\phi_t)\phi_c]\nabla(\nabla^2\phi_c),$$

and

$$-\phi_t\nabla\bar{\mu}_t = -\left[a_p S_{\rho\theta} + \frac{\chi_{ct}}{2}\nabla\phi_c + \Lambda_{ct}\nabla\phi_c\nabla^2\phi_c\right]\phi_t - \left[\chi_t\phi_t + k_B T \frac{1 - \phi_c}{1 - \phi_c - \phi_t}\right]\nabla\phi_t + (\Lambda_t\phi_t)\nabla(\nabla^2\phi_t), \tag{A18}$$

respectively. Here, we defined $S_{\rho\theta} = \kappa_\rho(1/\rho - 1/\rho_t)\nabla(1/\rho) + \kappa_\theta(\theta - \theta_t)\nabla\theta$ and $\nabla^2$ is the surface Laplacian. These expressions highlight that COPII transport explicitly depends on the membrane curvature through the $\nabla J$ term, while TANGO1 transport explicitly depends on the ring radius and its angle with the membrane through $S_{\rho\theta}$.

### Membrane dynamics

To enforce local membrane incompressibility we introduce a Lagrange multiplier field $\sigma$ that can be interpreted as the membrane tension (*Rangamani et al., 2014*). Consequently, we aim at minimizing the Lagrangian functional

$$\mathcal{L} = \mathcal{R} + \int_\Gamma \sigma \operatorname{tr}(\mathbf{d})dS = \dot{\mathcal{F}} + \mathcal{D} + \mathcal{P}_{\text{ext}} + \int_\Gamma \sigma \operatorname{tr}(\mathbf{d})dS, \tag{A19}$$

where $\mathcal{R}$ is the Rayleighian defined in *Equation (A1)*. Following *Tozzi et al., 2019*, the dissipation rate of the free energy $\dot{\mathcal{F}}$ and the local area constraints can be expressed as functionals of the rate

variables $(\mathbf{w}, \mathbf{v}, v_n)$. The governing equations for the membrane mechanics are obtained by minimizing *Equation (A19)* with respect to $(\mathbf{w}, \mathbf{v}, v_n)$, and maximizing it with respect to $\sigma$. In the case of $W = W_{bend}$, this results in the well-known shape equation and incompressibility condition (*Zhongcan and Helfrich, 1989*; *Steigmann et al., 2003*). A full analysis of the governing equation for the total energy density of the COPII/TANGO1 system is out of the scope of this paper. For practical purposes, in what follows we proceed to a numerical minimization of *Equation (A19)*.

## Model implementation

### Axisymmetric parametrization and numerical scheme

Details of the model implementation for a single species can be found in *Tozzi et al., 2019*. Briefly, we formulate the model in axisymmetric coordinates so that each material point of the membrane is expressed in terms of the distance to the axis of symmetry $\rho(u, t)$ and of the axis of symmetry coordinate $z(u, t)$. Here, $u \in [0, 1]$ is the Lagrangian coordinate along the membrane arclength, and $t$ is time. The Dirichlet boundary conditions in the axisymmetric system take the form

$$\rho(0, t) = 0 ; \quad z'(0, t) = 0 ; \quad z(1, t) = 0 ; \quad z'(1, t) = 0. \tag{A20}$$

The fixed chemical potentials at the open boundary are ensured by

$$\phi_c(1, t) = \phi_c^0 ; \quad \phi_c'(1, t) = 0 ; \quad \phi_t(1, t) = \phi_t^0 ; \quad \phi_t'(1, t) = 0, \tag{A21}$$

where $\phi_c^0$ and $\phi_t^0$ are the imposed protein densities of COPII and TANGO1, respectively, at the open boundary, mimicking protein reservoirs far from the budding site.

To solve numerically the coupled chemo-mechanical system, we employ a staggered approach where at each time step, we first solve the protein density field for a given membrane shape, and then update the shape at fixed membrane density distribution. The state variables are discretized using B-splines with cubic B-spline basis functions. The chemical problem is solved with the finite element method using a backward Euler discretization in time of the protein transport equations *Equation (A16)*, and Newton's method to solve the resulting non-linear system. The mechanical problem is solved for a given distribution of proteins by minimizing the incremental Lagrangian from *Equation (A19)* with respect to space variables and maximizing with respect to the Lagrange multiplier.

### Simulation and analysis protocols

After a preliminary parameter analysis informed by the physics of the problem (see also the main text for a discussion on parameters), we chose the reference set of parameters given in *Appendix 1—table 1*. Except stated otherwise, all results are obtained for these parameter values. All computations are done on an initially flat membrane patch of 250 nm radius.

**Appendix 1—table 1.** Model parameters.

| Symbol | Parameter | Value |
|---|---|---|
| | *Material parameters* | |
| $\kappa$ | Membrane bending rigidity | 20 $k_B T$ |
| $R_c$ | Preferred radius of curvature of COPII | 35 nm |
| $C_c$ | COPII spontaneous curvature | $-2/R_c$ nm$^{-1}$ |
| $\rho_t$ | TANGO1 preferred radius of curvature | 45 nm |
| $\theta_t$ | TANGO1 preferred angle of curvature | $\pi/4$ |
| $\kappa_\rho$ | TANGO1 ring radius rigidity | 120 $k_B T$ |
| $\kappa_\theta$ | TANGO1 ring angle rigidity | $6.4 \times 10^{-4}$ $k_B T$ nm$^{-2}$ |
| $a_P$ | Characteristic area of a protein | 100 nm$^2$ |
| $\chi_c$ | COPII self-interaction coefficient | $-2$ $k_B T$ |

*Continued on next page*

*Appendix 1—table 1 continued*

| Symbol | Parameter | Value |
|---|---|---|
| $\chi_t$ | TANGO1 self-interaction coefficient | $-1.2\ k_BT$ |
| $\chi_{ct}$ | Affinity coefficient between COPII and TANGO 1 | $-0.4\ k_BT$ |
| $\Lambda_c$ | COPII interfacial coefficient | two $k_BT$ |
| $\Lambda_t$ | TANGO1 interfacial coefficient | $0.4\ k_BT$ |
| $\Lambda_{ct}$ | Coupling interfactial coefficient | $0.4\ k_BT$ |
| $\eta_m$ | Membrane viscosity | $5 \times 10^{-9}$ N s m$^{-1}$ |
| $\xi$ | Molecular drag coefficient ($= 2\pi\eta_m$) | $3.14 \times 10^{-8}$ N s m$^{-1}$ |
| | *Model constraints* | |
| $\phi_c^0$ | COPII protein surface fraction at the open boundary | 0.1 |
| $\phi_t^0$ | TANGO1 protein surface fraction at the open boundary | 0.02 |
| $\sigma$ | Membrane tension | 0.004–0.0096 $k_BT/nm^2$ |
| $N$ | Vertical axial force (applied for nucleation) | 0–0.56 $k_BT/nm$ |
| $\rho_{col}$ | Minimal neck radius in presence of procollagen molecules | 7.5 nm |

## Stable equilibrium

We assume that an equilibrium shape is reached if the maximum displacement between two time-steps of each material points is below $5 \times 10^{-3}$ nm for more than 10 time-steps in a row over a cumulative time larger than 1 ms.

## Shape parameter

To facilitate a quantitative comparison between the end-states of the system obtained for different sets of parameters, we use the shape parameter, which is essentially the maximum height of the bud normalized by the preferred diameter of curvature of COPII $\eta = z_{\max}/2R_c$ (see also main text).

## Coat nucleation

In our system, the flat membrane state is a locally (sometimes globally) stable equilibrium state. This means that a perturbation needs to be applied to initiate the nucleation of COPII coats. To ensure a reproducible perturbation protocol, we define a criteria for COPII coat nucleation such as the average surface density of COPII within a surface area $A_{\mathrm{nuc}} = \pi(25\ \mathrm{nm})^2$ around the axis of symmetry must satisfy $\int_{A_{\mathrm{nuc}}} \phi_c\ dS > 0.75\ A_{\mathrm{nuc}}$. Starting from a flat membrane patch with homogeneous distribution of species, a small upward point force of $N = 0.16\ k_BT/nm$ is applied at $\rho = 0$. The force induces membrane deformation and initiates the recruitment of COPII. If the force is sufficient to nucleate a COPII coat as defined above, the point force is set back to zero, and the system is free to evolve. Alternatively, if an equilibrium is reached but the nucleation criteria is not satisfied, we gradually increase $N$ until either a COPII coat is nucleated or until $N > 0.56\ k_BT/nm$, in which case we assume the flat membrane to be the stable state. The up-ward point force is implemented within the arclength parametrization by setting $F_n(0, t) = N$ in the power supply *Equation (A12)*.

## Neck closure

In the cases where the transport carrier closes, no equilibrium is reached. Consequently, we assume that if the neck radius goes below 5 nm, we reach the small length scale limit of the continuum modeling approach, and assume neck closure. Because the neck closure event happens at different times after the neck snap-through, in order to facilitate the comparison of the carrier height in a systematic manner in the equilibrium shape phase diagrams, we take $\eta$ at the minimum bud height after the neck snaps.

## Prevention of neck closure by procollagen molecules

In simulations where procollagen molecules prevent the total closure, an energy penalty $\int_\Gamma 10^{-3}\kappa/(\rho - \rho_{\mathrm{col}})^2 dS$ is imposed on the portion of the membrane $u \in [0.1; 0.8]$ using a hyperbolic tangent of the form $0.5[\tanh(100(u - 0.1)) - \tanh(100(u - 0.8))]$.

## Appendix 2

## Equilibrium analytical model of TANGO1-assisted transport intermediate formation (Model B)

Detailed description of the physical model of TANGO1-dependent transport intermediate formation

Here, we present the detailed description and derivation of the physical model of TANGO1-dependent transport intermediate formation presented in the main text. Our model builds on a previously presented mechanical model for clathrin-coated vesicle formation (*Saleem et al., 2015*), which includes the contributions from the coat polymerization on the membrane, the line tension of the polymerized coat, the membrane resistance to bending, and the membrane tension. We extended this model to allow for the growth of larger transport intermediates by incorporating (i) the effects of TANGO1 rings on COPII coats; and (ii) an outward-directed force (*Appendix 2—figure 1A*).

Analogously to the clathrin vesicle model by *Saleem et al., 2015*, we consider that the free energy per unit area of coat polymerization onto the membrane, $\mu_c$, has a bipartite contribution arising from the positive chemical potential of COPII binding to the membrane, $\mu_c^0$, and from the negative contribution of membrane deformation by bending, so $\mu_c = \mu_c^0 - 2\frac{\kappa_b}{R_c^2}$, where $\kappa_b$ is the bending rigidity of the lipid bilayer, and $R_c$ is the radius of curvature imposed by the spherically polymerized COPII coat. We define the COPII chemical potential, $\mu_c^0$, as the free energy gain of polymerization of a COPII unit area on the ER membrane, which includes the contributions of the interactions between the coat and the membrane ($\mu_{c-m}$) and between the coat subunits (actual polymerization interactions, $\mu_{c-c}$). For clathrin-coated vesicle formation, *Saleem et al., 2015* showed that the direct coat-coat interactions are the major contributors to the assembly of the coat. Based on this, we consider the simplified situation where the binding energy $\mu_c^0$ represents the polymerization energy due to COPII-COPII binding, and therefore the reservoir of COPII is the rest of the membrane (as is the case also for our dynamic model, see Appendix 1). An additional term associated to the possible elastic deformation of the COPII coat could be considered as $\mu_{coat,bend} = -\frac{1}{2}\kappa_{coat}\left(\frac{2}{R_c} - \frac{2}{R_{coat}}\right)^2$, where $\kappa_{coat}$ is the coat rigidity and $R_{coat}$ is the spontaneous radius of curvature of the coat (*Iglic et al., 2007*; *Boucrot et al., 2012*). However, we assume here, for simplicity, that the coat is considerably more rigid than the membrane, $\kappa_{coat} \gg \kappa_b$, so there is no coat deformation and $R_c = R_{coat}$. Alternatively, coat contribution to membrane bending has also been tackled by using a spontaneous curvature-based model (*Agrawal and Steigmann, 2009*; *Hassinger et al., 2017*). In our analytical model, we follow the approach of *Saleem et al., 2015*, which allows us to define the preferred spherical architecture of the polymerized coat. A spontaneous curvature-based approach was followed for our computational analysis of carrier shapes (Appendix 1). In summary, the free energy per unit area of the initially undeformed membrane due to COPII polymerization, $f_{coat}$, can be expressed as

$$f_{coat} = \frac{-\mu_c A_c}{A_p},$$ 
(B1)

where $A_c$ is the surface area of the membrane covered by the COPII coat, and $A_p$ is the projected area of the carrier, that is, the area of the initially undeformed membrane under the carrier (*Saleem et al., 2015*; *Appendix 2—figure 1B*). We express all the free energies per unit area of the initially flat membrane and consider that there is a densely packed assembly of TANGO1 rings, as have been experimentally observed in cells (*Raote et al., 2017*; *Raote et al., 2018*). We also consider that the amounts of TANGO1 and COPII units are non-limiting. We next consider a line energy for the coat subunits laying at the edge of the polymerizing structure. This line energy per unit area reads as

$$f_{line}^0 = \lambda_0 \frac{l}{A_p},$$ 
(B2)

where $\lambda_0$ is the line tension of the bare coat, and $l = 2\pi\rho$ is the length of the carrier edge, associated to the opening radius at the base of the carrier, $\rho$ (*Appendix 2—figure 1C*). Next, we consider the

contribution of the membrane tension, $\sigma$, to the free energy per unit area of the system, which reads as

$$f_{tension} = \sigma \frac{A_m}{A_p}, \tag{B3}$$

where $A_m$ is the surface area of the entire membrane after deformation (*Appendix 2—figure 1B*).

The following step is to expand on this model to include the contributions by which TANGO1 can modulate the formation of a transport intermediate. TANGO1 filaments are described by their length, $L_T$, and by their persistence length, $\xi_p = \frac{\kappa_T}{k_B T}$, –where $\kappa_T$ is the filament bending rigidity and $k_B T$ is the thermal energy, equal to the Boltzmann constant times the absolute temperature (*Doi and Edwards, 1986*)–, which describes how stiff the filament is. As long as the filament length is not much larger than the persistence length, the bending energy of the TANGO1 filament can be expressed as $F_{bend} = \frac{\kappa_T}{2} \int_{L_T} (c - c_0)^2 dl$, where $c$ and $c_0$ are the actual and spontaneous curvature of the filament, respectively, and the integral is performed over the entire filament length. We define positive spontaneous curvatures of the filament as those where the TANGO1-COPII interacting domains lie on the concave side of the filament, and negative when they lie on the convex side. For a TANGO1 filament of length $L_T$, that is bound to the circular boundary length of a COPII patch (of radius $\rho$), the filament bending energy per unit length can be written as $f_{bend} = \frac{\kappa_T}{2} (1/\rho - c_0)^2 \omega$, where we assumed that any existing filaments not adsorbed to the COPII patches adopt the preferred curvature, $c_0$, and where $\omega$ is the capping fraction: the fraction of COPII domain boundary length covered ('capped') by TANGO1 molecules. Hence, analogously to our discussion for the free energy of coat binding to the membrane, *Equation (B1)*, we can write the free energy per unit area of a TANGO1 filament as

$$f_{T,bend} = -\frac{\mu_T \, l}{A_p}, \tag{B4}$$

where $\mu_T = -f_{bend}$ includes the negative contribution of the filament bending energy. A positive contribution of the filament assembly chemical potential, $\mu_T^0$, is not considered here since we assume that the assembly chemical potential is independent of whether the filament is capping or not a COPII patch and hence the fraction of TANGO1 monomers forming a filament is independent of the capping fraction. Moreover, we want to stress that the bending energy penalty of the filament diverges when the bud approaches closure, meaning that either there is *uncapping* of the TANGO1 filament from the edge of the COPII coat at narrow necks or the shape transition of the carrier goes through intermediate shapes with a relatively large bud neck, such as Delaunay shapes (e.g. unduloids) (*Naito and Ou-Yang, 1997*). This second option is analyzed with our model A, detailed in Appendix 1. In addition, TANGO1 proteins have an affinity to bind COPII components, and hence adsorb to the boundary of the COPII domains by binding the most external subunits. We therefore consider an extra free energy term associated to this TANGO1-COPII interaction, which is proportional to the boundary length of the COPII domain capped by TANGO1, and hence reads as

$$f_{TANGO1-COPII} = -\Delta\lambda \, \omega \frac{l}{A_p}, \tag{B5}$$

where $\Delta\lambda$ is the interaction strength between TANGO1 and COPII. We can write together *Equations B2* and *B5* as

$$f_{line} = \lambda \frac{l}{A_p}, \tag{B6}$$

where $\lambda = \lambda_0 \left(1 - \frac{\Delta\lambda}{\lambda_0} \omega\right)$ can be understood as the effective line tension of the COPII coat, in which $\Delta\lambda$ is the reduction in the line tension due to TANGO1 capping, and hence is a measure of the linactant power of TANGO1.

Finally, the mechanical work performed by the outward-directed force, $N$, is also included in the free energy of the system, as

$$f_f = -\frac{N \, z_{max}}{A_p}, \tag{B7}$$

where $z_{max}$ is the length of the carrier (*Appendix 2—figure 1C*). At this stage, for the sake of simplicity, we disregard the effects of the growth-shrinkage dynamics of the polymerizing COPII lattice. Hence, the total free energy per unit area of the carrier, $f_c$, is the sum of all these contributions *Equations B1,3,4,6,7*,

$$f_c = f_{coat} + f_{line} + f_{tension} + f_{T,bend} + f_f. \tag{B8}$$

This free energy per unit area using a bare flat membrane as the reference state is presented in *Equation 1* in the main text.

## Geometry of the problem

Based on the proposed geometries for the growing carrier we can distinguish three geometries, depending on how complete the transport intermediate is: shallow buds, deep buds, and pearled intermediates (*Appendix 2—figure 1C*, panels (i) to (iii), respectively). This restricted family of shapes allows us to calculate as a function of the carrier morphology the geometric parameters that enter in *Equation 1*, namely, the area of the coat, $A_c$, the area of the membrane, $A_m$, the projected area, $A_p$, and the opening radius at the coat rim, $\rho$ (*Saleem et al., 2015*; *Appendix 2—figure 1B*). A convenient quantity to parametrize the shape of the carrier is the height of the carrier, $z_{max}$, which we will use in a dimensionless manner by normalizing it to the diameter of the spherical COPII bud, $\eta = z_{max} / 2R_c$.

### (i) Shallow bud

For a shallow bud (*Appendix 2—figure 1B (i)*), which corresponds to buds smaller than a hemisphere, we can write that $A_c = A_m = 2\pi R_c^2 (1 - \cos\theta)$, where $0 < \theta < \pi/2$ is the opening angle of the bud (see *Appendix 2—figure 1B (i)*). In addition, $A_p = \pi\rho^2 = \pi R_c^2 \sin^2\theta$; and $z_{max} = R_c(1 - \cos\theta)$. Expressing these quantities as a function of the shape parameter, $\eta$, we obtain

$$A_c = A_m = 4\pi R_c^2 \eta : \eta < \frac{1}{2}, \tag{B9}$$

$$A_p = 4\pi R_c^2 \eta \, (1 - \eta) : \eta < \frac{1}{2}, \tag{B10}$$

$$\rho = 2R_c \sqrt{\eta(1 - \eta)} : \eta < \frac{1}{2}. \tag{B11}$$

### (ii) Deep bud

For a deep bud (*Appendix 2—figure 1B (ii)*), which corresponds to buds larger than a hemisphere, we can write that $A_c = 2\pi R_c^2 (1 - \cos\theta)$, where $\pi/2 < \theta < \pi$. In addition, $A_m = \pi R_c^2 \left(1 + (1 - \cos\theta)^2\right)$; $A_p = \pi R_c^2$; and $z_{max} = R_c(1 - \cos\theta)$. Expressing these quantities as a function of the shape parameter, $\eta$, which in this case ranges between $\frac{1}{2} < \eta < 1$, we obtain

$$A_c = 4\pi R_c^2 \eta : \frac{1}{2} < \eta < 1, \tag{B12}$$

$$A_m = \pi R_c^2 \left(1 + 4\eta^2\right) : \frac{1}{2} < \eta < 1, \tag{B13}$$

$$A_p = \pi R_c^2 : \frac{1}{2} < \eta < 1, \tag{B14}$$

$$\rho = 2R_c\sqrt{\eta(1-\eta)} : \frac{1}{2} < \eta < 1. \tag{B15}$$

## (iii) Pearled intermediate

A pearled intermediate corresponds to carriers form by an incomplete bud with opening angle $0 < \theta < \pi$, connected via a narrow connection with $n$ complete buds (*Appendix 2—figure 1B (iii)*). Here, we can write that $A_c = 2\pi R_c^2 [2n + (1 - \cos\theta)]$, where $0 < \theta < \pi$ and $n \geq 1$. In addition, $A_m = \pi R_c^2 \left[4n + 1 + (1 - \cos\theta)^2\right]$; $A_p = \pi R_c^2$; and $z_{max} = R_c(2n + 1 - \cos\theta)$. Expressing these quantities as a function of the shape parameter, $\eta$, we obtain

$$A_c = 4\pi R_c^2 \eta : \eta > 1, \tag{B16}$$

$$A_m = \pi R_c^2 \left(1 + 4n + 4(\eta - n)^2\right) : \eta > 1, \tag{B17}$$

$$A_p = \pi R_c^2 : \eta > 1, \tag{B18}$$

$$\rho = 2R_c\sqrt{(\eta - n) - (\eta - n)^2} : \eta > 1. \tag{B19}$$

Putting together *Equations B9-19*, we get:

$$A_c = 4\pi R_c^2 \eta \tag{B20}$$

$$A_m = \begin{cases} 4\pi R_c^2 \eta, & \eta < 1/2 \\ \pi R_c^2 \left[1 + 4n + 4(\eta - n)^2\right], & \eta > 1/2 \end{cases} \tag{B21}$$

$$A_p = \begin{cases} 4\pi R_c^2 \eta(1 - \eta), & \eta < 1/2 \\ \pi R_c^2, & \eta > 1/2 \end{cases} \tag{B22}$$

$$\rho = 2R_c\sqrt{(\eta - n)(1 - \eta + n)}, \tag{B23}$$

$$z_{max} = 2R_c \eta. \tag{B24}$$

where $n = [\eta]$ is the number of complete pearls, the brackets denoting the integer part operator. This allows us to express *Equation 1* in the main text as

$$\Delta f_c = \frac{\sigma\eta - \tilde{\mu}}{1 - \eta} + \frac{\tilde{\lambda}(\omega)}{\sqrt{\eta(1 - \eta)}} - \frac{4\omega\,\tilde{\kappa}_T c_0 R_c}{\eta(1 - \eta)} + \frac{\omega\,\tilde{\kappa}_T}{[\eta(1 - \eta)]^{3/2}}, \qquad \eta < 1/2, \tag{B25}$$

$$\begin{aligned} \Delta f_c = 4\sigma\left[n + (\eta - n)^2\right] - 4\tilde{\mu}\eta + 4\tilde{\lambda}(\omega)\sqrt{(\eta - n)(1 - \eta + n)} \\ + 4\omega\tilde{\kappa}_T\left[\frac{1}{\sqrt{(\eta - n)(1 - \eta + n)}} - 4\,c_0 R_c\right], \qquad \eta > 1/2, \end{aligned} \tag{B26}$$

where $\tilde{\mu} = \mu_c^0 - 2\frac{\kappa_b}{R_c^2} + \frac{N}{2\pi R_c}$, $\tilde{\lambda}(\omega) = \frac{(\lambda_0 - \omega\,\Delta\lambda)}{R_c} + 4\omega\tilde{\kappa}_T(c_0 R_c)^2$, and $\tilde{\kappa}_T = \frac{\kappa_T}{8R_c^3}$ (*Equation 2* in the main text).

## Parameter estimation

The free energy per unit area, *Equation (1)* depends on a number of structural, biochemical, and mechanical parameters, which we can split in three groups: (i) membrane-associated parameters, (ii)

coat-associated parameters, and (iii) TANGO1-associated parameters (*Appendix 2—table 1*). As for the membrane-associated parameters, we have the lateral tension, $\sigma$, and the bending rigidity, $\kappa_b$.

Regarding membrane-associated parameters, we use the experimentally measured values of the standard membrane tension of the ER, $\sigma_{ER} = 0.003\ k_B T/nm^2$ (*Upadhyaya and Sheetz, 2004*); and of the membrane bending rigidity, $\kappa_b = 20\ k_B T$ (*Niggemann et al., 1995*; *Appendix 2—table 1*).

Regarding coat-associated parameters, we use the size of the standard spherical COPII vesicle, $R_c = 37.5\ nm$ (*Miller and Schekman, 2013*). The line tension and the binding free energy of the polymerizing COPII coat, $\lambda_0$ and $\mu_c^0$, respectively, have not been, to the best of our knowledge, experimentally measured. Nevertheless, for clathrin coats, which lead to the formation of vesicles of a size comparable to the standard COPII vesicles, these values have been recently measured, yielding a value of $\lambda_{clathrin} = 0.05\ pN$ for the line tension and of $\mu_{clathrin}^0 = 0.024 \pm 0.012\ k_B T/nm^2$ for the binding free energy (*Saleem et al., 2015*). We use these values as starting estimations for COPII coats, which we will then vary within a certain range (*Appendix 2—table 1*). It is also informative to estimate the binding energy of COPII polymerization per molecule. To obtain this energy in units of $k_B T$ for the polymerization of a single COPII unit, we use the characteristic size of the Sec13-31 heterotetramers, $\sim 30\ nm$ (*Stagg et al., 2008*; *Zanetti et al., 2013*), which gives a characteristic area of a COPII unit, $a_{COPII} \sim (30\ nm)^2$. This gives that $\mu_c^0\ a_{COPII} \sim 22\ k_B T$, very similar to the estimated lateral interaction energy per triskelion of the clathrin coat (*Saleem et al., 2015*).

Finally, regarding the TANGO1-associated parameters, which are associated to different protein-protein interactions, we have the bending rigidity of the TANGO1 filament, $\kappa_T$; the preferred curvature of the filament, $c_0$; and the TANGO1-COPII binding energy/linactant strength of TANGO1, $\Delta\lambda$. The elastic parameters of the TANGO1 filament, $\kappa_T$ and $c_0$, depend on the chemistry of the bonds between the different proteins within a TANGO1 filament. As we lack experimental data on the value of these parameters, we consider them within a wide range of reasonable values. Typical values of the bending rigidity of intracellular filaments formed by protein-protein interactions, such as intermediate filaments, are of the order of $\kappa_{IF} = 2000\ pN \cdot nm^2$ (*Fletcher and Mullins, 2010*), which we consider as an upper limit for the rigidity of a TANGO1 filament. In addition, by taking $\kappa_T = 0$, we can exploit our model to study the case where TANGO1 proteins do not form a cohesive filament by attractive lateral protein-protein interactions, but individual proteins can still bind COPII subunits and hence act as monomeric linactants. For our analytical analysis, we will start by taking a zero spontaneous curvature of the TANGO1 filament, $c_0 = 0$, and later study it within a range given by twice the radius of experimentally measured TANGO1 rings, $-0.02\ nm^{-1} < c_0 < 0.02\ nm^{-1}$. For the value of $\Delta\lambda$, we can make an upper limit estimate, by considering that the TANGO1-COPII binding energy should be lower than the corresponding binding energy between polymerizing COPII components, that is, $\Delta\lambda\ l_1 < \mu_c^0\ l_1 l_2$, where $l_1 \approx 16\ nm$ and $l_2 \approx 10\ nm$ are the lateral dimensions of the inner COPII coat components Sec23/24 (*Matsuoka et al., 2001*). Hence, our estimation gives that $\Delta\lambda < 0.24\ k_B T/nm$, and therefore we use as the initial value for our analysis half of the upper limit value, $\Delta\lambda = 0.12\ k_B T/nm$, which is 10-fold larger than the bare line tension of the coat (*Appendix 2—table 1*).

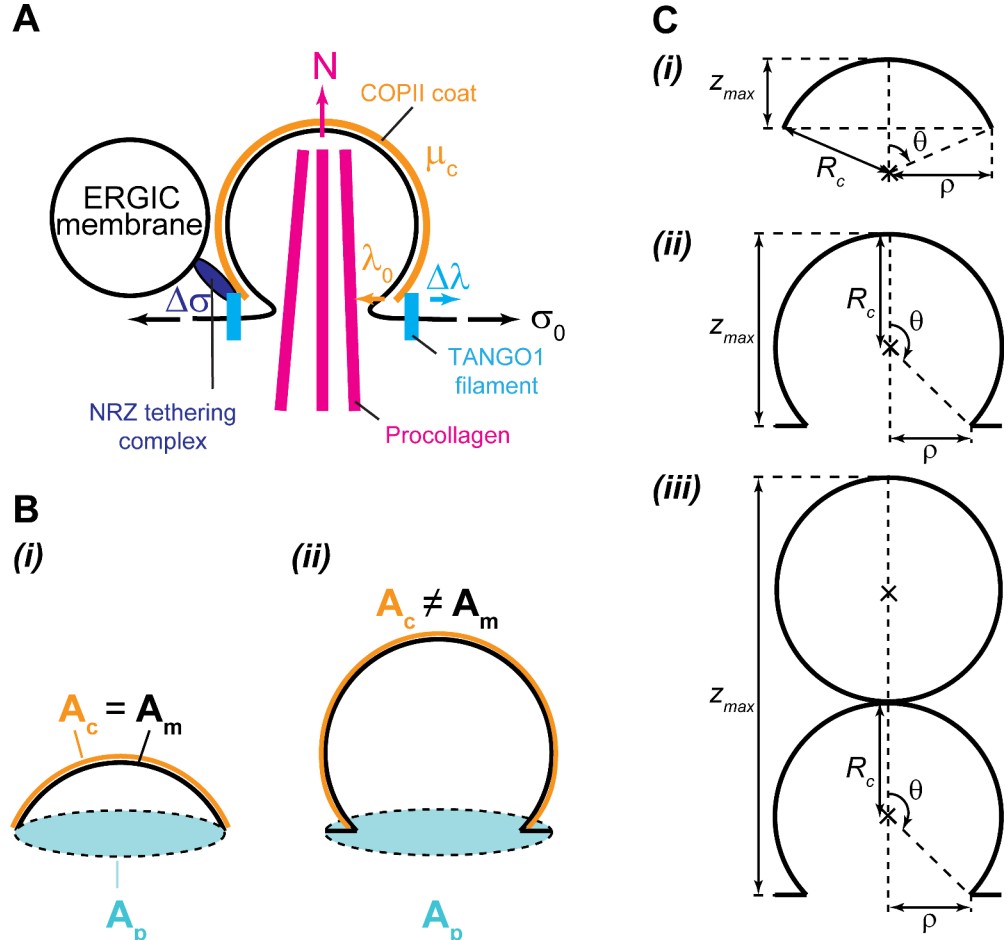

**Appendix 2—figure 1.** Geometry and physical forces in the transport intermediate generation model. (**A**) TANGO1 rings assembling on the ER membrane are depicted in light blue, accounting for a line tension reduction of the COPII coat, $\Delta\lambda$. The ER membrane is shown in black, associated with a tension, $\sigma_0$. The COPII coat polymerizing on the membrane is depicted in orange, and accounts for a coat binding free energy (or chemical potential), $\mu_c$, and a COPII coat line tension, $\lambda_0$. Packaged procollagen rods are shown in magenta, which can (but not necessarily) contribute with a pushing normal force, $N$, and sterically prevent membrane fission. Finally, ERGIC53-containing membranes tethered to the export site through the NRZ complex (dark blue) are shown, which can be a source of membrane tension reduction, $\Delta\sigma$. (**B**) Schematics of the different surface areas used in the model for both shallow (i) and deep (ii) buds: the projected area, $A_p$, shown in light blue; the membrane area, $A_m$, shown in black; and the coat area, $A_c$, shown in orange. (**C**) Scheme of the carrier geometry used for shallow buds (i), deep buds (ii); and pearled carriers (iii).

**Appendix 2—table 1.** Parameters used in the large transport intermediate formation model. The free energy *Equation (2)* depends on a number of different parameters, which are described in this table.

| Parameter | Description | Value | Notes | Reference |
|---|---|---|---|---|
| $\sigma$ | Membrane tension | 0.003 $k_BT/nm^2$ (ER); 0.0012 $k_BT/nm^2$ (Golgi membranes) | | *Upadhyaya and Sheetz, 2004* |
| $\kappa_b$ | Membrane bending rigidity | 20 $k_BT$ | | *Niggemann et al., 1995* |

*Continued on next page*

*Appendix 2—table 1 continued*

| Parameter | Description | Value | Notes | Reference |
|---|---|---|---|---|
| $R_c$ | Radius of curvature of the COPII coat | 37.5 nm | | *Miller and Schekman, 2013* |
| $\lambda_0$ | Bare coat line tension | 0.012 $k_BT$/nm | Not measured for COPII. Used the clathrin value as a reference | *Saleem et al., 2015* |
| $\mu_c^0$ | COPII coat polymerization energy | $0.024 \pm 0.012$ $k_BT$/nm$^2$ | Not measured for COPII. Used the clathrin values as a reference | *Saleem et al., 2015* |
| $\kappa_T$ | TANGO1 filament bending rigidity | 120 $k_BT$ nm | Not measured. Range based on standard filament rigidities (see text) | |
| $c_0$ | TANGO1 filament spontaneous curvature | $(-0.02, 0.02)$ nm$^{-1}$ | Not measured. Range based on observed TANGO1 ring sizes | *Raote et al., 2017* |
| $\lambda$ | TANGO1 linactant strength | 0.12 $k_BT$/nm | Not measured. Range based on protein-protein affinity (see text) | - |
| $N$ | Outwards-directed force | $0 - 5k_BT$/nm | Not measured. Range based on known intracellular forces | *Kovar and Pollard, 2004* (Actin); *Block et al., 2003* (Molecular motors) |

