## [Decision Letter]

Thank you for submitting your article "A physical mechanism of TANGO1-mediated bulky cargo export" for consideration by *eLife*. Your article has been reviewed by three peer reviewers, one of whom is a member of our Board of Reviewing Editors, and the evaluation has been overseen by Suzanne Pfeffer as the Senior Editor. The reviewers have opted to remain anonymous.

The reviewers have discussed the reviews with one another and the Reviewing Editor has drafted this decision to help you prepare a revised submission.

As the editors have judged that your manuscript is of interest, but as described below that additional work is required before it is published, we would like to draw your attention to changes in our revision policy that we have made in response to COVID-19 (https://elifesciences.org/articles/57162). First, because many researchers have temporarily lost access to the labs, we will give authors as much time as they need to submit revised manuscripts. We are also offering, if you choose, to post the manuscript to bioRxiv (if it is not already there) along with this decision letter and a formal designation that the manuscript is "in revision at *eLife*". Please let us know if you would like to pursue this option. (If your work is more suitable for medRxiv, you will need to post the preprint yourself, as the mechanisms for us to do so are still in development.)

The paper presents an elaborate theoretical analysis of TANGO1-mediated transport of large procollagen cargo from endoplasmic reticulum exit sites (ERES). The paper addresses a very exciting topic since the mechanisms of cellular transport of large cargo are still poorly understood. The central hypothesis of the paper is that TANGO1 binds to the boundary of a growing COPII bud, where it (a) acts as linactant and modulates COPII assembly; (b) tethers ERGIC53-containing membranes that subsequently fuse with ERES and promote the growth of the transport carrier; (c) binds to procollagen in ER lumen and directs it into the carrier. Compared to the previous submission, the paper has been, without any doubt, substantially improved. However, as the topic is complex, it is no surprise that there are still some open questions.

What became evident during the review process is a sense that what must seem clear to the authors was not clear to the reviewers, and the authors are strongly encouraged to respond with great care in the revised text to address each of the comments/concerns raised. Because of the extensive reviewer comments, we include them in full to guide you as you revise the manuscript.

Reviewer #1:

The paper presents an elaborate theoretical analysis of TANGO1-mediated transport of large procollagen cargo from endoplasmic reticulum exit sites (ERES). The paper addresses a very exciting topic since the mechanisms of cellular transport of large cargo are still poorly understood. The central hypothesis of the paper is that TANGO1 binds to the boundary of a growing COPII bud, where it (a) acts as linactant and modulates COPII assembly; (b) tethers ERGIC53-containing membranes that subsequently fuse with ERES and promote the growth of the transport carrier; (c) binds to procollagen in ER lumen and directs it into the carrier. The analysis presented in the paper comprises two distinct theoretical models:

A) within the first model, the transport carrier shape is approximated by a string of spherical pearls and parametrized by a shape parameter eta. The equilibrium configuration of the carrier is determined by analytical minimization of the system's free energy in a quasi-static approximation.

B) the second model is a dynamic computational model, where the shape of the transport carrier is constrained only by axial symmetry, and the time-evolution of the carrier configuration is determined by using Onsager's variational approach applied to a membrane with freely diffusing and interacting COPII and TANGO1 proteins. The solution is calculated numerically.

If compared to previous submissions, the paper has been, without any doubt, substantially improved. However, as the topic is complex, it is no surprise that there are still some open questions.

1) The paper could be more clear on why TANGO1 has to act as a linactant stabilizing a growing COPII coat. Is it because the linactant effect is needed to slow down the growing COPII bud and thus give the time that is needed to recruit procollagen? If so, this should be additionally emphasized in the paper and corroborated by the model results. As it stands now, model A focuses mostly on the quasi-equilibrium shapes of the carrier and not on kinetics (the kinetics of the growth is described only briefly in L706-L733). Within equilibrium mechanics "stabilizing" can have a slightly different meaning than "slowing down". Consequently, the reader could wonder if the linactant effect is needed at all? The role of TANGO1 could as well be just to recruit the procollagen. The carrier would grow by itself, driven by COPII polymerization (see also point 2). In this regime, the closure of COPII bud would be prevented simply by the presence of procollagen on the luminal side, as described in model B. Why is this option not possible? Within the same lines: are the authors aware of any data on the kinetics of procollagen recruitment?

2) The region in the parameter space, where the proposed model can describe the transport carrier growth, is relatively small, i.e., having other parameters fixed, it spans approximately within 0.026 kT/nm2 < mu_c_^0^ < 0.032 kT/nm2 and 0.001 < σ < 0.003 kT/nm2 (Figure 4A). This raises concerns that the proposed mechanism is not very robust, e.g., if one parameter from the Table 1 changes for ~10% the proposed mechanism could break down completely. E.g. if mu_c_^0^ is 0.034, the carrier could grow spontaneously, without the need for a linactant effect or tension regulation. I would encourage the authors to discuss this aspect of their model.

3) While the analysis carried out with model B (dynamic computational model) is very comprehensive and sophisticated (kudos to its authors!), it is not clear if it can be applied to TANGO1/COPII transport carrier growth as presented in this paper. How to compare the results of both models presented? Namely, while the first model assumes that the whole carrier is covered with COPII, and that COPII coat is surrounded by a TANGO1 ring, the second model in intrinsically continuous and its results do not show any phase separation (or capping/no-capping transitions). Also, it is not clear how to compare the (values of) parameters mu_c_^0^ and chi_c_, which are not defined in the same way. Even if these parameters were conceptually similar, can one be sure that the model B is being solved in the same parameter region as Model A (as we saw earlier, model A is very sensitive to the values of mu_c_^0^)? Unsurprisingly, Tozzi, Walani and Arroyo, 2019, show growth of protrusions that are very similar to the ones presented in Figure 7D, but with only one protein species and different parameter values. In addition, the time scales within both models are quite different, Model A assumes average budding transition times in the region of 0.1 min, while Model B shows that this time can be less than 100 ms.

Reviewer #2:

The authors present a theoretical study of how TANGO1 protein controls the transport of large cargo (procollagen) from the endoplasmic reticulum exit sites (ERES) to the Golgi apparatus. This is an interesting problem and has been dealt with experimentally from various angles by the authors (several published in *eLife*). This work builds on the 2018 *eLife* paper that revealed how TANGO1 organizes to corral COPII coats, and recruits ERGIC (ER-Golgi intermediate compartment) membranes for procollagen export. Here the authors develop an analytical model that considers COPII assembly into a spherical shell on the membrane and TANGO1 polymerization into a filament that caps the COPII lattice. They then study how various physical parameters (TANGO1 self-interaction, COPII self-interaction, TANGO-COPII interaction, membrane tension, external force) influence the resulting equilibrium membrane shape. In the end they discuss the dynamic growth of the protrusion.

The problem is interesting, and the Abstract seems exciting, however, in my opinion the data presented fails to support the main claims of the paper.

Main concerns:

i) The first claim by the authors is that by capping the COPII lattice TANGO1 prevents premature carrier scission and enables packing the bulky cargo into the carrier. This claim is repeated many times throughout the paper, without any data to support it, since the model does not consider carrier scission or cargo loading. This claim appears mainly speculative.

The authors do indeed show that TANGO1 capping stabilizes incomplete buds, but they do not show the relationship between this stabilization and cargo-loading or carrier scission. In fact, in their dynamic model they prevent carrier scission "by hand", by not allowing radii below certain size. They also do not present any experimental evidence that the procollagen transport is necessarily carried by incomplete buds or pearled tubes.

Hence the result that the capping stabilizes incomplete buds can also be interpreted in other ways, for instance as inhibitory to the carrier growth since the capping imposes a free energy barrier for the growth (Figure 2). In summary: there is no evidence in the paper that the capping indeed enables packing the bulky cargo.

ii) Related to the previous comment, the main assumption of the paper is that the procollagen transport is facilitated by incomplete COPII buds. However, the authors do not really present any evidence for that. For instance, if the carrier has a spherocylindrical shape (which has been reported for COPII in vitro) the whole conclusions on the role of capping would be different.

iii) The second main claim of the paper is that TANGO1 enables the carrier growth by reducing the membrane tension, by recruiting ERGIC. The authors model this effect "by hand", by decreasing the tension in their model and showing that the free energy minimum is shifted to long carrier shapes. In their dynamic model they impose sudden decrease and increase in tension and show that this enables and prevents the carrier growth, respectively. This result is nice, but not unexpected. The authors also do not really show that TANGO1 enables the carrier growth, but rather that the decrease of the membrane tension enables it, be it mediated by TANGO1 or otherwise. The role of membrane tension on the protrusion growth has been studied extensively before (see eg the body of work by Nassoy and Bassereau, for instance EPL 2005), and agrees with the findings presented here. In that sense the physical result that the decrease in the membrane tension enables the protrusion growth is not novel, although it has been put in a context of a different system.

iv) The paper feels unnecessarily long and repetitive, where the same claims are repeated many times.

Overall, this is a nice exploratory paper that makes bold claims that appear not to be supported by data, it is my impression that the authors somewhat try to fit the results into a preconceived mechanism of how TANGO1 enables procollagen transport, as opposed to the mechanism arising from their data. The physical result that the decrease in the membrane tension enables protrusion growth is not surprising (or novel). In my opinion the work does not present a substantial development with respect to the 2018 *eLife* paper by the same authors.

Reviewer #3:

In this manuscript, the authors develop a theoretical model to analyse a scenario for the formation of large membrane carrier able to export procollagen fibres from ERES. COPII coat components polymerise into a spherical cap, which is prevented from maturation and budding by TANGO1 proteins assembling into a ring at the coat edge. TANGO1 also promotes the fusion of ERGIC vesicles which lower membrane tension locally and allow for the longer COPII-covered carriers able to incorporate procollagen.

I appreciate some aspect of the model, primarily the analysis of the self-organisation of two types of membrane proteins into non-trivial structure. The fairly thorough study of the different possible range of behaviour, including a careful account of the relevance of different parameters, is valuable.

On the other hand, other aspects of the manuscript seem dubious to me and rely on what I believe to be incompatible assumptions. This is true for the tension regulation aspect, which is unfortunately a point that is strongly emphasise in the manuscript. Another weak point is the way the interaction with the procollagen fibres is treated.

Another problem I have with the manuscript is the constant mixing between rigorous results obtained from the model and speculations which are not supported by the model. Therefore, I think that this paper present interesting results regarding the self-organisation of coatomers and ring-forming proteins, which are certainly relevant for the situation under study, but that it requires important rewriting to put the result in their proper context and to separate facts from conjectures.

1) The deformation of membrane by aggregation of proteins inducing membrane curvature has been studied in great details, and the role of the different physical parameters in the process, including membrane tension, is well documented. The originality of the present model is the involvement of a second class of proteins that stabilise the coat (e.g. disfavor the formation of a full spherical coat) due to attractive interaction with coat proteins AND the fact that these proteins tend to form semi-flexible filaments. The problem is treated at thermodynamic equilibrium, through energy minimisation. As such, it assumes ergodicity; that the entire space of configuration can be explored, and cannot investigate situations where the system is kinetically trapped into a particular state. This limitation is insufficiently discuss in the text, especially in the Introduction. Furthermore, the possibility for kinetic trapping is often explicitly mentioned without a word of caution, which could confused an uninformed reader that this possibility is accounted for by the model. It is not, and this must be stated explicitly.

2) Thermodynamic equilibrium implies that proteins in the coat and in the ring can freely exchange with a reservoir, or that the time scales associated with this exchange are much faster than other time scales, such as the one for tension variation. The chemical potential of COPII monomer is assumed fixed, regardless of the coat size. This means that there exists a reservoir of monomer at fixed chemical potential with which the coat can freely exchange component. I see two possibilities. The reservoir is the flat membrane, which is itself in equilibrium with coat component in the cytosol, or the reservoir is the cytosol itself, which requires that coat component bind directly from the cytosol onto the growing coat. The chemical potential \mu_c_ is said to include binding energy, without specifying if it is the binding of one monomer to the membrane or the binding between two monomers. I assume it is the latter which suggests that the reservoir is indeed the rest of the membrane. This makes sense to me, but it should be clearly discussed, including the fact that this model only makes sense if coatomers can be freely exchange between the growing coat and the rest of the membrane. By the way, the binding energy \mu_c_^0^ is expressed as an energy per unit area. This is not optimal. It should be expressed as an energy, so it can be compared with the thermal energy kT.

3) The equilibrium treatment becomes particularly problematic when the situation involves diffusion barrier, which is the main argument put forward by the author in support of their claim that TANGO1 is a regulator of membrane tension, allowing to the elongation of membrane carrier. For this two happen, at least three conditions must be satisfied: i) lipids must be prevented from diffusing across the TANGO1 ring, ii) ERGIC vesicles must be allowed to fuse within the area delimited by the TANGO1 ring, and iii) coatomers must be able to freely exchange with their reservoir, i.e. to cross the diffusion barrier. It seems rather unlikely that condition 1 is strictly satisfied. the TANGO1 ring could slow down lipid diffusion, but probably not abolish it. This is indeed the conclusion of Raote et al., 2020, so the question of tension regulation becomes a competition between time scales; the time scale of tension equilibration vs. the time scale of coat formation. It is not discussed as such but it should.

There is an even bigger problem. There is a clear conflict between conditions (i) and (iii). It is not reasonable to claim that lipids are trapped but not coatomers. Furthermore, the fusion of ERGIC vesicles, even if one assumes that it occurs inside the TANGO1 ring, which seems very unlikely considering that this part of the membrane is crowded with COPII presumptuous, would locally and transiently reduce tension because membrane area of a completely different composition than the initial coat would be added to the region inside the ring. None of this is discussed. Probably because it is a difficult problem, but the way it is handled here is not at all satisfactory. The bottom line is that the claim of tension regulation is presumptuous and not acceptable in the current form.

4) As far as I can see, procollagen is entirely absent from the model equations. Nevertheless it is present in the Discussion in mostly two ways; as a way to sterically hinder neck closure or as responsible for a normal force that helps elongate the membrane carrier. The first aspect is entirely conjectural and is not included in any way in the model. It should be presented as such in the Discussion. The second is somewhat introduced phenomenologically in the model, but is not properly discussed. For polymerising procollagen fibres to push on the coat, it would need to exert an opposite force on something else. What would that be? The ER membrane opposite to the ERES presumably, but expect if the fiber is perfectly perpendicular to that membrane, the force would also result in a torque that would made the fibre tilt. It is said repeatedly in the text that this force is dispensable, but we are still lacking an argument to explain how the fibre would properly orientate along the carrier axis of symmetry to be properly incorporated. Since this is a model for how to build membrane carrier for bulky cargo, the interaction between the cargo and the carrier must be discussed more thoroughly.

5) Regarding the energy minimisation procedure. It is unclear why all energies are expressed in unit of A_p_ (Equations.1-4 etc…). A_p_ is the projected area of the carrier, which changes non-linearly with the number of coatomers as the coat grows. It is unclear to me why this area is important, except if one wishes to discuss steric interaction between buds. I would think that the proper thing to do is to compare the energy per coatomer for different coat geometry, rather than to minimise the energy divided by A_c_. As such, I don't think that the minimisation procedure is correct. For instance, looking at Figure 2B, we can see that the "free" energy of the double bud is smaller than that of the single bud so that the former is said to be "globally stable". What should be compared is the chemical potential (energy per COPII monomer) is both situations, which could very well favour single buds for the parameters of Figure 1B, in particular due to the favorable interaction between TANGO1 and COPII monomers.

Related to this, I could not figure out what is the difference between A_m_ and A_c_. A_c_ is supposed to be the coat area and A_m_ the membrane area, but in this simplified geometry using spherical caps, I imagine that the entire curved part of the membrane is covered by coat, so why is A_m_ not equal to A_c_?

6) TANGO1 is repeatedly described as a linactant, but it is not clear that the stabilisation of incomplete buds is due to the linactant effect. The TANGO1 ring is rigid and its caping of the coat edges involve an elastic energy penalty. Isn't this the dominant effect of coat stabilisation, rather than the linactant effect?

7) The section proposal of experimental approaches to test the model is not very useful. It does not really provide experimental test of the model based on the quantitative findings of the model. The section .…“ TANGO1 as a regulator of membrane tension homeostasis” in largely speculative and not supported by the model. It is rather presents hypothesis that are a starting point of the model, several of which are dubious (in particular the fact that tension is reduced while membrane composition is not changed, as discussed above). It should be presented as speculation.

[Editors' note: further revisions were suggested prior to acceptance, as described below.]

Thank you for submitting your article "A physical mechanism of TANGO1-mediated bulky cargo export" for consideration by *eLife*. Your article has been reviewed by three peer reviewers, one of whom is a member of our Board of Reviewing Editors, and the evaluation has been overseen by Suzanne Pfeffer as the Senior Editor.

The reviewers have discussed the reviews with one another and the Reviewing Editor has drafted this decision to help you prepare a revised submission.

Summary:

The paper entitled "A physical mechanism of TANGO1-mediated bulky cargo export" by Raote et al. has been re-submitted to *eLife* as a research advance to their prior research paper entitled "TANGO1 builds a machine for collagen export by recruiting and spatially organizing COPII, tethers and membranes" (Raote et al., 2018).

The paper addresses cellular transport of large cargo from endoplasmic reticulum exit sites (ERES), which is a very important open question in cell biology. Based on their previous experimental paper (Raote et al., 2018), the authors put forward a hypothesis that TANGO1 binds to the boundary of a growing COPII bud in the ERES, where it (a) acts as linactant and modulates COPII assembly; (b) tethers ERGIC53-containing membranes that subsequently fuse with ERES, decreases the ERES membrane tension and thus promote the growth of the transport carrier; (c) binds to procollagen in the ER lumen and directs it into the carrier. The present paper introduces an elaborate and interesting theoretical analysis showing that the TANGO1 hypothesis is compatible with the current understanding of physical mechanisms that guide membrane remodeling. Thus, the results present an essential and invaluable step towards a thorough understanding of cellular trafficking of large cargo.

The reviewers agree that the paper is suitable for publication in *eLife* after the authors address the following two concerns:

1) The narrative of the paper has been much improved and the complex modeling is now presented clearly and concisely. However, the reviewers judge that the link between the modeling and the biological situation remains rather speculative. The reviewers therefore suggest that the authors add a paragraph of open and honest discussion on the speculative link between the model and biology.

2) The authors should address the inconsistent use of high / low tension values in the two models. (0.006 / 0.003 kBT/nm2 in the continuous model, and 0.003 / 0.0012 kBT/nm2 in the equilibrium model).

---

## [Author Response]

Reviewer #1:The paper presents an elaborate theoretical analysis of TANGO1-mediated transport of large procollagen cargo from endoplasmic reticulum exit sites (ERES). The paper addresses a very exciting topic since the mechanisms of cellular transport of large cargo are still poorly understood. The central hypothesis of the paper is that TANGO1 binds to the boundary of a growing COPII bud, where it (a) acts as linactant and modulates COPII assembly; (b) tethers ERGIC53-containing membranes that subsequently fuse with ERES and promote the growth of the transport carrier; (c) binds to procollagen in ER lumen and directs it into the carrier. The analysis presented in the paper comprises two distinct theoretical models:A) within the first model, the transport carrier shape is approximated by a string of spherical pearls and parametrized by a shape parameter eta. The equilibrium configuration of the carrier is determined by analytical minimization of the system's free energy in a quasi-static approximation.B) the second model is a dynamic computational model, where the shape of the transport carrier is constrained only by axial symmetry, and the time-evolution of the carrier configuration is determined by using Onsager's variational approach applied to a membrane with freely diffusing and interacting COPII and TANGO1 proteins. The solution is calculated numerically.If compared to previous submissions, the paper has been, without any doubt, substantially improved. However, as the topic is complex, it is no surprise that there are still some open questions.1) The paper could be more clear on why TANGO1 has to act as a linactant stabilizing a growing COPII coat. Is it because the linactant effect is needed to slow down the growing COPII bud and thus give the time that is needed to recruit procollagen? If so, this should be additionally emphasized in the paper and corroborated by the model results. As it stands now, model A focuses mostly on the quasi-equilibrium shapes of the carrier and not on kinetics (the kinetics of the growth is described only briefly in L706-L733). Within equilibrium mechanics "stabilizing" can have a slightly different meaning than "slowing down". Consequently, the reader could wonder if the linactant effect is needed at all? The role of TANGO1 could as well be just to recruit the procollagen. The carrier would grow by itself, driven by COPII polymerization (see also point 2). In this regime, the closure of COPII bud would be prevented simply by the presence of procollagen on the luminal side, as described in model B. Why is this option not possible? Within the same lines: are the authors aware of any data on the kinetics of procollagen recruitment?

We apologize for not explaining this point clearly in the previous version of the manuscript. The results obtained by using our equilibrium model indicate that *capping* of the TANGO1 ring to growing COPII patches modulates the energetically stable shapes of the transport intermediates (see e.g. Figure 4A). The increase in the linactant strength of the filament (and/or the reduction of its bending rigidity) triggers the formation of TANGO1 ring capping COPII coats. Importantly, these changes also lead to expansion of the parameter space in which incomplete, open budded structures are predicted (see Figure 6B,C).

To better illustrate this point, we include a new figure panel, Figure 6—figure supplement 1B. In there, we show the plots of two cross-section lines of Figure 6B, corresponding to a situation where TANGO1 is not a linactant (∆𝜆 = 0) and where it is a linactant (∆𝜆 = 0.12 𝑘_*_𝑇/𝑛𝑚), respectively. In these cross-section plots, we can readily observe how TANGO1 linactant strength modulates the stable bud shapes and the capping state. First, it expands the range of values of the parameter 𝜇_0_^1^ where the stable shape is a shallow open bud with a finite sized neck radius, 𝜌^∗^ (see Figure 6—figure supplement 1B); and second, it increases the radius of the opening neck, 𝜌^∗^. In summary, TANGO1 as a linactant (in capping conditions) implies that *(i)* there is a larger range of parameters that lead to stable open buds; and *(ii)* the radius of the neck opening is larger.

In our dynamic model, the linactant effect of TANGO1 is included by an interaction parameter (𝜒_05_, see Appendix 1—table 1). We have not tested the entire parameter space of the dynamic model, because it is more complex and elaborated that our equilibrium model (in the revised manuscript we emphasize the fact that we use the equilibrium model to obtain simple semi-quantitative and analytical estimations of the effects of the different model parameters). In spite of this, our results show that the effect of a functional TANGO1 (which forms a ring capping the COPII patch), as compared to an inert TANGO1 (no interactions), is to increase the region of the parameter space where incomplete, open-necked structures are found (Figure 5B). This is qualitatively similar to our predictions from the equilibrium model (Figure 5A), which we believe further supports our view that a possible function of the TANGO1-COPII interaction is in “stabilizing” structures with wide-open necks. This, we speculate, is used by TANGO1 to facilitate the packaging of procollagens into the nascent transport intermediate.

In summary, our data shows that although not strictly necessary, the linactancy of TANGO plays an important role in large carrier formation by creating wider necks and making the carrier growth mechanisms more robust. Importantly, in our previous manuscript (Raote et al., 2018), we experimentally showed that the mutant of TANGO1 with a reduced affinity for COPII components (TANGO1-∆PRD mutant), which can still bind and recruit procollagen to ERES, is defective in procollagen export. This observation suggests that a strong affinity of TANGO1 for COPII molecules (which corresponds in our models as the linactant strength of the filament) is indeed required for efficient procollagen export.

Unfortunately, we are not aware of any in vivo data on the kinetics of procollagen recruitment to export sites. It is known that the overall time spent from synthesis to secretion to the ECM of procollagen I is about 10-30 minutes (see e.g. Weinstock M, Leblond CP. (1974) J. Cell Biol. 60(1):92–127), and that a synchronizable procollagen I takes about 15 minutes from completion of triple helical formation in the ER to arrival at early Golgi cisternae (McCaughey et al., 2018). Moreover, the half-lives of COPII components on the ER membrane are in the order of a few seconds (Forster et al., 2006). It is therefore plausible to suggest that a mechanism to stabilize open nascent carriers exists, and that such mechanism could facilitate the efficient packing of complex cargoes. Taking into consideration our theoretical results on the role of TANGO1 in stabilizing incomplete transporters (Figure 2, Figure 5A,B), we propose that TANGO1 rings can fulfill such a task of arresting the growth of standard COPII carriers and stabilizing open shallow buds for the efficient packaging of procollagens (Figure 8A,B).

We concur with the reviewer on the difference between stabilization and slow-down of the dynamics. With our equilibrium model, we can only obtain the stable and metastable membrane shapes for each set of parameters. We refer to “stabilization of open shallow buds” to the formation of such stable (or metastable) shapes. If one of the parameters (e.g. membrane tension) changes, then the free energy profile of the system is concomitantly altered and new stable/metastable states appear (see e.g. Figure 4A). Depending on the initial state of the system (e.g. globally stable state before changing the parameters), the new stable state could be separated by an energy barrier, which we calculated and is presented in Figure 4C. We simplified the estimation of the transition times based on Arrhenius kinetics and added a cautionary note in the text that our equilibrium model cannot provide dynamic information about the actual transition times between metastable shape. We have also removed any reference to “slowing down” the transitions between local equilibrium states.

All these points above have been clarified and explained in more detail in the revised version of the manuscript.

2) The region in the parameter space, where the proposed model can describe the transport carrier growth, is relatively small, i.e., having other parameters fixed, it spans approximately within 0.026 kT/nm2 < mu_c_^0^ < 0.032 kT/nm2 and 0.001 < σ < 0.003 kT/nm2 (Figure 4A). This raises concerns that the proposed mechanism is not very robust, e.g., if one parameter from the Table 1 changes for ~10% the proposed mechanism could break down completely. E.g. if mu_c_^0^ is 0.034, the carrier could grow spontaneously, without the need for a linactant effect or tension regulation. I would encourage the authors to discuss this aspect of their model.

We agree that the region of the parameter space that allows for a tension regulation-mediated carrier growth seems relatively small. However, similar narrow ranges of parameters have been shown to regulate clathrin-coated vesicle formation (Saleem et al., 2015; Hassinger et al., 2017). Moreover, the shape diagram is much more robust with respect to other parameters, in particular to those related to the TANGO1 filaments (see Figure 6B,C). Indeed, if 𝜇_0_^1^ is too large, growth of a procollagen-containing carrier could grow spontaneously without the requirement of other mechanisms (tension regulation, linactant effect, etc.). However, we know that different modular aspects of TANGO1 interactions with its effectors are individually required for efficient procollagen export from the ER (e.g. PRD domain, which controls linactant strength; cytosolic coiled-coiled domains, which control TANGO1 filament formation and recruitment and fusion of ERGIC membranes) (Raote et al., 2018). We believe these experimental evidences provide context to the relatively narrow range of parameters obtained by our theoretical approach. Finally, although in some parameter regions TANGO1 is not required for bud growth, its presence (or absence) allows for a richer range of COPII-mediated budding behavior, influencing cargo selectivity and export efficiency. We discuss these points in the revised version of the manuscript.

3) While the analysis carried out with model B (dynamic computational model) is very comprehensive and sophisticated (kudos to its authors!), it is not clear if it can be applied to TANGO1/COPII transport carrier growth as presented in this paper. How to compare the results of both models presented? Namely, while the first model assumes that the whole carrier is covered with COPII, and that COPII coat is surrounded by a TANGO1 ring, the second model in intrinsically continuous and its results do not show any phase separation (or capping/no-capping transitions). Also, it is not clear how to compare the (values of) parameters mu_c_^0^ and chi_c_, which are not defined in the same way. Even if these parameters were conceptually similar, can one be sure that the model B is being solved in the same parameter region as Model A (as we saw earlier, model A is very sensitive to the values of mu_c_^0^)? Unsurprisingly, Tozzi, Walani and Arroyo, 2019, show growth of protrusions that are very similar to the ones presented in Figure 7D, but with only one protein species and different parameter values. In addition, the time scales within both models are quite different, Model A assumes average budding transition times in the region of 0.1 min, while Model B shows that this time can be less than 100 ms.

In our dynamic model, which we have renamed in the revised version as model “A” due to the re-ordering of the manuscript, we have not directly observed dissolution of TANGO1 rings after they are initially formed. Although we might find such a behavior through a thorough exploration of the parameter space, we believe this would be too overwhelming in the context of the present study, and is beyond the scope of this paper. However, we did observe that there is nucleation of TANGO1 ring-like structures at the base of COPII coats (phase separation) (see e.g. Figure 2B). The reviewer is right that this model is continuous in the sense that the TANGO1 ring is not a two-dimensional line but more like a diffuse interface with a certain thickness (somehow analogous to the way phase-field or level-set methods deal with interfaces). In spite of this fundamental difference, the results of this model (formation of a narrow ring-like TANGO1 structure around COPII patches) qualitatively agree with the results obtained using the equilibrium model (renamed as model B).

Moreover, it is true that it is not straightforward to quantitatively compare 𝜇_0_^1^ (equilibrium model) with 𝜒_0_ (dynamic model), but given the similarity of behaviors between the two models, we consider that we are qualitatively in the right range. Indeed, since the two models are fundamentally different, we do not think that a quantitative comparison would be more informative. However, we argue that the qualitative match between the two models (see e.g. Figure 5A,B), despite their different implementation of the underlying hypotheses, suggest that the proposed mechanism is robust and possibly more general than any of the two specific models.

In Tozzi et al., 2019, similar pearled shapes are indeed obtained with a single (curvature-inducing) protein component. It is important to note, their explanation required imposition of a cut-off neck radius to obtain such shape, exactly the same way we impose a minimum radius representing the procollagen molecules in the bud. Importantly, the question we ask is completely different. Here we ask what possible mechanism compatible with known TANGO1 characteristics can regulate COPII-mediated procollagen export. Our answer is that in a system with curvature-inducing proteins (here COPII) such as the one considered in Tozzi et al., adding a second species (TANGO1) finely controls the budding mechanism. And that when the neck size is sterically constrained, such as in the presence of procollagen, long transport intermediates can grow. This later point was not rationalized in biological context by Tozzi et al., but rather used as a practical way to obtain a variety of shapes.

Finally, about the different time scales, we would like to highlight that they represent different physical processes. The equilibrium model, by definition, cannot provide any dynamic information on the shape changes, but it allows us to estimate the transition time between local equilibrium states using Arrhenius kinetics (knowing the free energy barriers between those states). This time scale is determined by the height of energy barriers and temperature of the system (fluctuations allowing to overcome the energy barriers). This is in 0.1 min time scale reported for relatively low energy barriers. The dynamic model, however, provides us with the actual deformation dynamics to reach a local equilibrium state from an initial (unstable) state. This time scale is related to membrane bending and protein diffusion, and describe a totally different process compared to the equilibrium model. In this context, our computations show deformation times of the order of 0.1 s. However, our dynamic model does not explicitly incorporate thermal fluctuations, and therefore the exploration of stable shapes from metastable shapes is not possible, unless the system is taken to a different state (by e.g. membrane tension reduction), where the system can naturally evolve to its new stable shape. Bringing back the system to the initial state can show indeed the existence of those different locally stable shapes (see Figure 3, where the transient decrease in tension leads to growth of the carrier).

All these points are now discussed in the revised version of the manuscript. We also adapted the x-axes in Figure 3A,B to highlight the fact that the two time scales (shape changes and in that case, membrane tension changes) are independent to each other.

Reviewer #2:The authors present a theoretical study of how TANGO1 protein controls the transport of large cargo (procollagen) from the endoplasmic reticulum exit sites (ERES) to the Golgi apparatus. This is an interesting problem and has been dealt with experimentally from various angles by the authors (several published in eLife). This work builds on the 2018 eLife paper that revealed how TANGO1 organizes to corral COPII coats, and recruits ERGIC (ER-Golgi intermediate compartment) membranes for procollagen export. Here the authors develop an analytical model that considers COPII assembly into a spherical shell on the membrane and TANGO1 polymerization into a filament that caps the COPII lattice. They then study how various physical parameters (TANGO1 self-interaction, COPII self-interaction, TANGO-COPII interaction, membrane tension, external force) influence the resulting equilibrium membrane shape. In the end they discuss the dynamic growth of the protrusion.The problem is interesting, and the Abstract seems exciting, however, in my opinion the data presented fails to support the main claims of the paper.

Following the reviewer’s comments (see also our specific answers below), we rewrote the Abstract to be more cautious and accurate with our claims.

Main concerns:i) The first claim by the authors is that by capping the COPII lattice TANGO1 prevents premature carrier scission and enables packing the bulky cargo into the carrier. This claim is repeated many times throughout the paper, without any data to support it, since the model does not consider carrier scission or cargo loading. This claim appears mainly speculative.The authors do indeed show that TANGO1 capping stabilizes incomplete buds, but they do not show the relationship between this stabilization and cargo-loading or carrier scission. In fact, in their dynamic model they prevent carrier scission "by hand", by not allowing radii below certain size. They also do not present any experimental evidence that the procollagen transport is necessarily carried by incomplete buds or pearled tubes.Hence the result that the capping stabilizes incomplete buds can also be interpreted in other ways, for instance as inhibitory to the carrier growth since the capping imposes a free energy barrier for the growth (Figure 2). In summary: there is no evidence in the paper that the capping indeed enables packing the bulky cargo.

The reviewer is right that our models do not explicitly consider carrier scission. Regarding procollagen loading, although it is not included explicitly in any of our models, it is included “effectively” as a minimal neck radius in our dynamic model (see Appendix 1).

Taking this into account, our theoretical approaches focused on understanding the growth of a transport intermediate for procollagen export beyond that of a regular COPII-coated spherical vesicle. Our results show that, under certain conditions, TANGO1 forms a ring by binding to the periphery of a COPII lattice (a situation we referred to as *capping*) (see e.g. Figure 6 and Figure 5A,B). Notably, our results reveal that when there is a TANGO1 ring (capping conditions), the equilibrium shape of the membrane is altered in a way that generally favors open shapes with relatively large neck openings. This prevents the formation of fully closed buds (therefore preventing fission) (Figure 2B). In contrast, in the absence of TANGO1, the buds tend to close their necks to very small radii (Figure 2A), that we interpret as a possible route toward fission, although the precise fission mechanism is not modeled here.

We suggest that stabilization of open shallow buds could possibly enable efficient recruitment of procollagen to the export site, which is mediated by TANGO1 itself. Importantly, experimental evidence indicates a role for TANGO1 in efficiently packing procollagen for export from the ERES. Hence, our interpretation of the theoretical results presented here is in full agreement with the reviewer’s comment: TANGO1 rings have an “inhibitory role to the carrier growth”, which we phrased in our initial submission as “prevent premature fission”. Further events are required (such as tension regulation or force application, together with efficient procollagen packing to sterically prevent membrane fission away from the TANGO1 ring) for the elongation of the otherwise stable shallow bud into a large transport intermediate.

We reason that a possible source for such tension regulation would be mediated by TANGO1. TANGO1 family of proteins, being multi-domain, multi-functional proteins, are required to recruit membranes (ERGIC/early Golgi cisternae) through their interaction with the NRZ tethering complex. We hypothesize, based on estimates of ER and Golgi membrane tension, that these fusion events could produce a transient decrease in the ER membrane tension, which then would alter the stable shapes of the system and as a result can induce growth of the carrier (Figure 4B). In fact, we propose that this is the reason why in the absence of the tethering factors (NRZ proteins) or in cells expressing a deletion mutant of TANGO1 that is unable to recruit ERGIC membranes, procollagen export from the ER is inhibited (Raote et al., 2018).

Moreover, in our dynamic model, we introduced a steric interaction preventing neck closure (and therefore fission) mimicking the presence of procollagen at those necks. In the absence of functional TANGO1, which is the receptor molecule that presents procollagen to COPII coats (Saito et al., 2009), this interaction does not occur, and therefore we believe that our model’s assumptions are justified based on experimental grounds. This is explained in more detail in the revised version of the manuscript.

In sum, we agree that our interpretation is based on the combination of our experimental and theoretical findings. Therefore, following the reviewer’s comment, we have reorganized the text to make these points clear at the outset. Former Figure 1 is now split into two figures, leaving the second part (new Figure 8) as a summary of our working model. We are also more careful now with our explanations to clearly distinguish between the models’ results and our interpretation of these results within the experimental context of the process. We would also like to direct the reviewer to our detailed response to reviewer 1, point 1 on similar issues.

ii) Related to the previous comment, the main assumption of the paper is that the procollagen transport is facilitated by incomplete COPII buds. However, the authors do not really present any evidence for that. For instance, if the carrier has a spherocylindrical shape (which has been reported for COPII in vitro) the whole conclusions on the role of capping would be different.

We would like to argue that one has to be cautious about this, since COPII in vitro has no clear bearing on COPII in vivo and especially to procollagen export. In addition, as to date and to the best of our knowledge, there is no clear evidence on the shape and COPII lattice coverage of the structure responsible for procollagen export out of the ER. Two papers that claim the involvement of COPII mega containers have been challenged by new data (the work of Leikin and colleagues). For these reasons, in our dynamic model we do not assume any pre-defined membrane shape (as we do in the equilibrium model). The results from our dynamic model do not show the formation of large spherocylindrical shapes.

Based on the overall comments from the reviewers, we have re-structured and revised the text to present the more general dynamic model early in the text (and hence no shape restriction). We then move to the equilibrium, analytical model (spherical geometry imposed, justified by the results from the dynamic model) to gain insight into the physics of the process.

iii) The second main claim of the paper is that TANGO1 enables the carrier growth by reducing the membrane tension, by recruiting ERGIC. The authors model this effect "by hand", by decreasing the tension in their model and showing that the free energy minimum is shifted to long carrier shapes. In their dynamic model they impose sudden decrease and increase in tension and show that this enables and prevents the carrier growth, respectively. This result is nice, but not unexpected. The authors also do not really show that TANGO1 enables the carrier growth, but rather that the decrease of the membrane tension enables it, be it mediated by TANGO1 or otherwise. The role of membrane tension on the protrusion growth has been studied extensively before (see eg the body of work by Nassoy and Bassereau, for instance EPL 2005), and agrees with the findings presented here. In that sense the physical result that the decrease in the membrane tension enables the protrusion growth is not novel, although it has been put in a context of a different system.

We fully agree with the reviewer. We do not claim that our results on tension regulation of carrier shape, from a purely mechanical perspective, are completely novel, since as the reviewer says, there is a large pool of literature on that respect. In our previous submission, we cited only a few of these earlier manuscripts (including references to Apodaca et al., Kosmalska et al., Wu et al., Hassinger et al., Niu et al., and Saleem et al.), but we agree that we had missed relevant references. We have corrected this error by explaining this point better and adding new references (e.g. Cuvelier et al., 2015; Mercier et al., 2020).

However, as the reviewer also acknowledges, our aim here was to put the tension regulation mechanism for bud/tube growth in the context of procollagen export, and to rationalize the possible role(s) of TANGO1 in modulating this. So, we believe that our results convincingly show that membrane tension regulation is a possible mechanism to create a large transport intermediate for procollagen export at ERES, which, to the best of our knowledge, has not been studied before. We expect, as also has been published by other groups, that membrane tension regulation as a means to control membrane protrusion is more general, but is controlled by different players at different intra-cellular locations (e.g. a very recent manuscript by the Roux lab, Nature Cell Biology 22 (8), 947-959).

Unfortunately, we have no direct experimental evidence of membrane tension reduction/regulation at ERES for procollagen export (as described in the “Proposal for experimental approaches to test the model” section in the Discussion). However, we have a number of experimental results that, we believe, allow us to propose that TANGO1 can possibly mediate such tension reduction (such as recruitment of ERGIC membranes, binding to tethers, interaction with SNARE proteins). We have now rewritten the text to emphasize that tension reduction is the causative effect of protrusion growth, and that TANGO1 is only one of the possible means by which such tension reduction can occur (interestingly, both cells knock-downed for NRZ components and cells expressing the TANGO1 mutant that cannot recruit these tethers accumulate procollagen at the ER).

iv) The paper feels unnecessarily long and repetitive, where the same claims are repeated many times.

We thoroughly revised and streamlined the text to make the main text shorter and try to avoid any repetitive parts.

Overall, this is a nice exploratory paper that makes bold claims that appear not to be supported by data, it is my impression that the authors somewhat try to fit the results into a preconceived mechanism of how TANGO1 enables procollagen transport, as opposed to the mechanism arising from their data. The physical result that the decrease in the membrane tension enables protrusion growth is not surprising (or novel). In my opinion the work does not present a substantial development with respect to the 2018 eLife paper by the same authors.We respectfully disagree here with the reviewer. Our previous manuscript (Raote et al., 2018) was completely experimental, and in there we proposed that *“A tug-of-war between the filament bending and the effect on COPII stabilisation created by the adsorption of TANGO1 filaments around ERES would then dictate whether and how TANGO1 rings are formed. Interestingly, it has been shown that the line tension of the polymerising protein coat can play a key role in controlling the timing and size of clathrin-coated vesicles (Saleem et al., 2015). We thus propose that the stabilising effect of TANGO1 while adsorbing around ERES would serve as a physical mechanism to delay and enlarge the COPII vesicle, commensurate with cargo size”.*

In the current manuscript, we now back these proposed mechanisms with rigorous physical analyses. It is the contextualization of experimental observations within a physical framework and the presentation of their rigorous physical analysis that we believe is novel and represent a substantial development in our quest to understand the mechanisms of procollagen export from the ER.

Reviewer #3:In this manuscript, the authors develop a theoretical model to analyse a scenario for the formation of large membrane carrier able to export procollagen fibres from ERES. COPII coat components polymerise into a spherical cap, which is prevented from maturation and budding by TANGO1 proteins assembling into a ring at the coat edge. TANGO1 also promotes the fusion of ERGIC vesicles which lower membrane tension locally and allow for the longer COPII-covered carriers able to incorporate procollagen.I appreciate some aspect of the model, primarily the analysis of the self-organisation of two types of membrane proteins into non-trivial structure. The fairly thorough study of the different possible range of behaviour, including a careful account of the relevance of different parameters, is valuable.On the other hand, other aspects of the manuscript seem dubious to me and rely on what I believe to be incompatible assumptions. This is true for the tension regulation aspect, which is unfortunately a point that is strongly emphasise in the manuscript. Another weak point is the way the interaction with the procollagen fibres is treated.Another problem I have with the manuscript is the constant mixing between rigorous results obtained from the model and speculations which are not supported by the model. Therefore, I think that this paper present interesting results regarding the self-organisation of coatomers and ring-forming proteins, which are certainly relevant for the situation under study, but that it requires important rewriting to put the result in their proper context and to separate facts from conjectures.1) The deformation of membrane by aggregation of proteins inducing membrane curvature has been studied in great details, and the role of the different physical parameters in the process, including membrane tension, is well documented. The originality of the present model is the involvement of a second class of proteins that stabilise the coat (e.g. disfavor the formation of a full spherical coat) due to attractive interaction with coat proteins AND the fact that these proteins tend to form semi-flexible filaments. The problem is treated at thermodynamic equilibrium, through energy minimisation. As such, it assumes ergodicity; that the entire space of configuration can be explored, and cannot investigate situations where the system is kinetically trapped into a particular state. This limitation is insufficiently discuss in the text, especially in the Introduction. Furthermore, the possibility for kinetic trapping is often explicitly mentioned without a word of caution, which could confused an uninformed reader that this possibility is accounted for by the model. It is not, and this must be stated explicitly.

We thank the reviewer for the comment. We completely agree that our analytical model is an equilibrium model and therefore ergodicity is assumed. To study the system with this equilibrium model, we performed direct global minimization of the free energy of the system as a function of the free parameters of the model (the shape parameter, 𝜂, and the capping fraction, 𝜔). By computing the free energy of the system for a wide range of these parameters (see e.g. Figure 4A), we have access not only to the globally stable configuration of the system, but also to locally metastable states. This further allows us to estimate the free energy barriers separating the metastable states from the globally stable states. With these results we qualitatively estimate average time scales for shape transitions between locally stable states, based on Arrhenius kinetics (that is, shape changes mediated by thermal fluctuations).

We now stress that this is just an estimation based on an equilibrium model, so the actual dynamics of the membrane shape between two locally stable shapes are beyond the reach of such a model. In addition, following the reviewer’s suggestions, we now explicitly state in the revised version the limitations of our model in terms of the kinetics of shape transitions.

2) Thermodynamic equilibrium implies that proteins in the coat and in the ring can freely exchange with a reservoir, or that the time scales associated with this exchange are much faster than other time scales, such as the one for tension variation. The chemical potential of COPII monomer is assumed fixed, regardless of the coat size. This means that there exists a reservoir of monomer at fixed chemical potential with which the coat can freely exchange component. I see two possibilities. The reservoir is the flat membrane, which is itself in equilibrium with coat component in the cytosol, or the reservoir is the cytosol itself, which requires that coat component bind directly from the cytosol onto the growing coat. The chemical potential \mu_c_ is said to include binding energy, without specifying if it is the binding of one monomer to the membrane or the binding between two monomers. I assume it is the latter which suggests that the reservoir is indeed the rest of the membrane. This makes sense to me, but it should be clearly discussed, including the fact that this model only makes sense if coatomers can be freely exchange between the growing coat and the rest of the membrane. By the way, the binding energy \mu_c_^0^ is expressed as an energy per unit area. This is not optimal. It should be expressed as an energy, so it can be compared with the thermal energy kT.

We thank the reviewer for pointing out that this part was not clear. COPII coat polymerization is a multi-step process that involves many reactions, including the initiation by GDP to GTP exchange in Sar1 that anchors it into the membrane, the posterior recruitment of the inner layer of the COPII coat, etc. In our models, for the sake of simplicity, we described all the subcomponents of the COPII inner and outer coats as a single effective species (a “COPII unit”). For our purposes, we think that a more detailed description would only add complexity and extra free parameters without providing relevant biophysical insights. This is explicitly stated in the revised version with a new section with the main hypotheses and limitations of our model formulations.

Following this, and regarding the nature of the COPII reservoir, the reviewer is indeed correct. In general terms, one would consider a multi-step process by which *(i)* free soluble cytosolic “COPII units” could bind to/unbind from the membrane; and *(ii)* membrane-bound (and also soluble) “COPII units” would undergo a COPII-COPII binding reaction (polymerization). However, this would add extra complexity to our modeling strategy. To simplify this, we followed the approach proposed by Saleem et al., 2015, to investigate the mechanics of clathrin-coated vesicle formation. In there, the authors defined the chemical potential of clathrin as the free energy gain of polymerization of a clathrin unit area. That polymerization energy, 𝜇_0_^1^, included the contributions of the interactions between the coat and the membrane (𝜇_0KG_) and between the coat subunits (actual polymerization interactions, 𝜇_0K0_). In their experimental measurements, the authors showed that the direct coat-coat interactions are the major contributors to the assembly of the coat. Based on this information, we assumed that a similar situation is occurring for COPII polymerization: that the binding energy 𝜇_0_^1^ represents the polymerization energy due to COPII-COPII binding, and therefore the reservoir of COPII is the rest of the membrane (as is the case also for our dynamic model). This is now discussed in the revised version of our manuscript.

Finally, in the revised version, we have included an estimation of the binding energy of COPII polymerization per molecule. To estimate the binding energy in units of 𝑘_*_𝑇 for the binding of a single COPII unit, we used the characteristic size of the Sec13-31 heterotetramers, ~30𝑛𝑚 (see e.g. Stagg et al. Cell 2008; Zanetti et al., 2013), which gives a characteristic area of a COPII unit, 𝑎_1_~(30 𝑛𝑚)^2^. This gives that 𝜇_c_^0^ 𝑎_0_~22 𝑘*_B_*𝑇.

3) The equilibrium treatment becomes particularly problematic when the situation involves diffusion barrier, which is the main argument put forward by the author in support of their claim that TANGO1 is a regulator of membrane tension, allowing to the elongation of membrane carrier. For this two happen, at least three conditions must be satisfied: i) lipids must be prevented from diffusing across the TANGO1 ring, ii) ERGIC vesicles must be allowed to fuse within the area delimited by the TANGO1 ring, and iii) coatomers must be able to freely exchange with their reservoir, i.e. to cross the diffusion barrier. It seems rather unlikely that condition 1 is strictly satisfied. the TANGO1 ring could slow down lipid diffusion, but probably not abolish it. This is indeed the conclusion of Raote et al., 2020, so the question of of tension regulation becomes a competition between time scales; the time scale of tension equilibration vs. the time scale of coat formation. It is not discussed as such but it should.There is an even bigger problem. There is a clear conflict between conditions (i) and (iii). It is not reasonable to claim that lipids are trapped but not coatomers. Furthermore, the fusion of ERGIC vesicles, even if one assumes that it occurs inside the TANGO1 ring, which seems very unlikely considering that this part of the membrane is crowded with COPII presumptuous, would locally and transiently reduce tension because membrane area of a completely different composition than the initial coat would be added to the region inside the ring. None of this is discussed. Probably because it is a difficult problem, but the way it is handled here is not at all satisfactory. The bottom line is that the claim of tension regulation is presumptuous and not acceptable in the current form.

We apologize for the lack of clarity in our previous version. TANGO1 acting as a diffusion barrier (Raote et al., 2020) is not the main argument to support the idea of TANGO1 acting as a means to regulate membrane tension. Actually, our results (both models) hold true regardless of how membrane tension is regulated (see also our detailed responses to reviewer #2). In the models presented in this paper we do not explicitly include the presence of a (partial) diffusion barrier. This is now clearly stated in the revised version, where we also discuss our current model and its results in the context of reference Raote et al., 2020.

We know from our previous experimental work that TANGO1 forms rings at the ERES around COPII coats, and that TANGO1 recruits the NRZ tethering complex (by its coiled-coil domain CC1), which then recruits ERGIC/early Golgi membranes (Raote et al., 2018). Based on these observations, here we propose that the recruited membrane fuse to the ERES, providing a possible tension regulation mechanism. Although this hypothetical mechanism relies on the presence of TANGO1 rings at the export sites, it is independent of the role of TANGO1 as a diffusion barrier (Raote et al., 2020). We do not know the exact site where these membranes would fuse, but we do know that it likely occurs in the vicinity of the export site (see e.g. Figure 6A in Raote et al., 2018). This membrane fusion event would lead to a local reduction of membrane tension due to the addition of lipid molecules (see e.g. Sens and Turner, 2006). Tension however tends to dynamically equilibrate after membrane fusion (Shi et al., 2018). Hence, the reviewer is indeed correct that the problem of tension regulation is a problem of time scales: if tension equilibrates much faster than the time the membrane needs to change its shape, then there is no effective tension reduction.

Mechanical equilibration of the membrane shape (𝜏_GQ0R_) can be theoretically estimated to be of the order of milliseconds (see e.g. Campelo et al., 2017; Sens and Rao, 2013) (see also our own results from the dynamic model where shape changes are found within few milliseconds (see e.g. Figure 3)). The diffusive behavior of membrane tension (at the plasma membrane of HeLa cells) has been measured to be associated with a diffusion coefficient of 𝐷_T_ = 0.024 𝜇𝑚^2^/𝑠 (Shi et al., 2018), which gives a characteristic diffusion time of the order of 𝜏_T_~1 𝑠 (using characteristic length scales of ~100–500 𝑛𝑚), which is much larger than 𝜏_GQ0R_. We envision that the TANGO1-induced diffusion barrier at the base of the growing bud can slow down even further the dynamics of tension equilibration. Hence, we think it is reasonable to assume that the time scale of mechanical equilibration of the membrane shapes is much smaller than that of tension equilibration.

This argument together with a discussion on the possible roles of TANGO1 as a diffusion barrier in controlling or modulating the tension regulation mechanisms are now found in the revised version of the manuscript.

4) As far as I can see, procollagen is entirely absent from the model equations. Nevertheless it is present in the Discussion in mostly two ways; as a way to sterically hinder neck closure or as responsible for a normal force that helps elongate the membrane carrier. The first aspect is entirely conjectural and is not included in any way in the model. It should be presented as such in the Discussion. The second is somewhat introduced phenomenologically in the model, but is not properly discussed. For polymerising procollagen fibres to push on the coat, it would need to exert an opposite force on something else. What would that be? The ER membrane opposite to the ERES presumably, but expect if the fiber is perfectly perpendicular to that membrane, the force would also result in a torque that would made the fibre tilt. It is said repeatedly in the text that this force is dispensable, but we are still lacking an argument to explain how the fibre would properly orientate along the carrier axis of symmetry to be properly incorporated. Since this is a model for how to build membrane carrier for bulky cargo, the interaction between the cargo and the carrier must be discussed more thoroughly.

The proposed role of procollagen as a means to sterically prevent neck closure is now presented in the Discussion of the revised manuscript and not in Figure 1 as before.

We now also explain in more detail our hypothesis of how procollagen folding can exert a force to elongate the carrier. We suggest that TANGO1 (as a ring) indirectly anchors a soluble lumenal protein (procollagen) to the membrane. Hence, as the reviewer correctly presumed, the force would act against the membrane through TANGO1. Indeed, if such force was applied to a single TANGO1 molecule, it would induce a torque that would tilt the procollagen and tend to position it parallel to the membrane plane (in which case the normal force applied at the tip of the carrier would vanish). However, we propose that the ring-like organization of TANGO1 could allow for torque compensation because it would act as the structure to which forces are applied. Specifically, a torque would appear for each of the TANGO1 molecules in the ring. When integrating these torques out for the entire TANGO1 ring, the resultant net torques would vanish and therefore such a structure would help align procollagen molecules perpendicular to the TANGO1 ring plane. This is now discussed in the revised manuscript.

5) Regarding the energy minimisation procedure. It is unclear why all energies are expressed in unit of A_p_ (Equations 1-4 etc). A_p_ is the projected area of the carrier, which changes non-linearly with the number of coatomers as the coat grows. It is unclear to me why this area is important, except if one wishes to discuss steric interaction between buds. I would think that the proper thing to do is to compare the energy per coatomer for different coat geometry, rather than to minimise the energy divided by A_c_. As such, I don't think that the minimisation procedure is correct. For instance, looking at Figure 2B, we can see that the "free" energy of the double bud is smaller than that of the single bud so that the former is said to be "globally stable". What should be compared is the chemical potential (energy per COPII monomer) is both situations, which could very well favour single buds for the parameters of Figure 1B, in particular due to the favorable interaction between TANGO1 and COPII monomers.Related to this, I could not figure out what is the difference between A_m_ and A_c_. A_c_ is supposed to be the coat area and A_m_ the membrane area, but in this simplified geometry using spherical caps, I imagine that the entire curved part of the membrane is covered by coat, so why is A_m_ not equal to A_c_?

We apologize for not being clear with this part of the model. For our equilibrium analytical model, we followed the framework proposed by Saleem et al., 2015. Indeed, we consider (as it was also done in Saleem et al.’s manuscript) that there is a densely packed assembly of TANGO1 rings, as experimentally observed in cells (Raote et al., 2017; Raote et al., 2018); and that the amount of TANGO1 and COPII subunits is large enough (non-limiting amounts).

The reason why we express the energies in unit of 𝐴_X_ was not clearly explained in our previous manuscript. In fact, what we did was to compute the free energy per transport intermediate (per bud) and then divide it by the surface area of the initially flat, uncoated membrane. Like this, we obtain the free energy per unit area, 𝑓_0_ (see e.g. Equation 1 in the revised version). The surface area projected by the coat on the initially flat membrane represents the surface area of the flat, unperturbed membrane, and that is the reason why we divide our energies by 𝐴_X_. These explanations are now made clear in the revised text.

Also, there is a subtle difference between 𝐴_G_ and 𝐴_0_, which only arises for structures with 𝜂 > 1/2 (larger than half buds). 𝐴_0_ is the area of the coated surface of the membrane (represented in orange in the new Appendix 2—figure 1B); whereas 𝐴_G_ is the overall area of the membrane in the bud (black surface area in Appendix 2—figure 1B), which includes the curved surface under the coat plus the flat surface of the membrane that connects to the projected area limit (region shaded in light blue in Appendix 2—figure 1B).

6) TANGO1 is repeatedly described as a linactant, but it is not clear that the stabilisation of incomplete buds is due to the linactant effect. The TANGO1 ring is rigid and its caping of the coat edges involve an elastic energy penalty. Isn't this the dominant effect of coat stabilisation, rather than the linactant effect?

The reviewer states correctly that stabilization of incomplete buds is not solely described by the linactant effect of TANGO1, but by a combination of different factors. These include the linactant strength of TANGO1 and also, as the reviewer suggests, the filament bending rigidity and the filament preferred curvature. In fact, our results indicate that it is the capping transition that facilitates the stabilization of open shallow buds. Hence, the role that the linactant strength of TANGO1 plays is in inducing capping. For instance, if TANGO1 would not have any linactant capacity, there would be no capping transition (see also our response to reviewer 1, major comment 1). This is now further explained in the text.

7) The section proposal of experimental approaches to test the model is not very useful. It does not really provide experimental test of the model based on the quantitative findings of the model. The section “ TANGO1 as a regulator of membrane tension homeostasis” in largely speculative and not supported by the model. It is rather presents hypothesis that are a starting point of the model, several of which are dubious (in particular the fact that tension is reduced while membrane composition is not changed, as discussed above). It should be presented as speculation.

Since this manuscript is a purely theoretical paper that presents no new experiments, we thought it was of interest to include such section in the Discussion. We believe that it can present ideas for future experimental works on the topic, following *eLife*’s editorial suggestions on such manuscripts (Shou et al., 2015). Following the reviewer’s comment, we adapted part of the text to provide more clear testable predictions.

Regarding the section on TANGO1 as a regulator of membrane tension homeostasis, we also revised it to a large extent, to be clear about which of our statements are hypothesis, results, and speculations (also following comments from reviewer 2).

[Editors' note: further revisions were suggested prior to acceptance, as described below.]

The reviewers agree that the paper is suitable for publication in eLife after the authors address the following two concerns:1) The narrative of the paper has been much improved and the complex modeling is now presented clearly and concisely. However, the reviewers judge that the link between the modeling and the biological situation remains rather speculative. The reviewers therefore suggest that the authors add a paragraph of open and honest discussion on the speculative link between the model and biology.

We adapted and expanded a paragraph in the Discussion section, where we now state clearly the potential link between our physical hypotheses and the available biological data:

“We have proposed and analyzed a theory by which TANGO1 assembles into a ring at the ERES to facilitate cargo export. […] The new experimental data will undoubtedly improve our understanding of how cells engage to export cargoes based on their size, volume, and the overall cellular needs.”

2) The authors should address the inconsistent use of high / low tension values in the two models. (0.006 / 0.003 kBT/nm2 in the continuous model, and 0.003 / 0.0012 kBT/nm2 in the equilibrium model).

This is now discussed in the revised manuscript:

“It is worth noting that, despite the different modeling and implementation choices taken in these two approaches, the values of the membrane tension at which the predictions agree only differ by a factor two. This difference could certainly be further reduced by undergoing a thorough analysis of the computational model's parameter space. However, given the computational cost of such study and the uncertainty of the experimental estimates for the ER and Golgi membrane tensions, seeking such a quantitative match between two qualitative models would only bring very limited insights.”